# Unified and explainable molecular representation learning for imperfectly annotated data from the hypergraph view

Bowen Wang [1,9], Junyou Li[2,9], Donghao Zhou[1], Lanqing Li [1,2], Jinpeng Li[1], Ercheng Wang[2], Jianye Hao [3] ✉, Liang Shi [4], Chengqiang Lu[5], Jiezhong Qiu[2], Tingjun Hou [6] ✉, Dongsheng Cao [7] ✉, Guangyong Chen[8] ✉ & Pheng Ann Heng [1]

Molecular representation learning (MRL) has shown promise in accelerating drug development by predicting chemical properties. However, imperfectly annotation among datasets pose challenges in model design and explainability. In this work, we formulate molecules and corresponding properties as a hypergraph, extracting three key relationships: among properties, molecule-to-property, and among molecules, and developed a unified and explainable multi-task MRL framework, OmniMol. It integrates a task-related meta-information encoder and a task-routed mixture of experts (t-MoE) backbone to capture correlations among properties and produce task-adaptive outputs. To capture underlying physical principles among molecules, we implement an innovative SE(3)-encoder for physical symmetry, applying equilibrium conformation supervision, recursive geometry updates, and scale-invariant message passing to facilitate learning-based conformational relaxation. OmniMol achieves state-of-the-art performance in properties prediction, reaches top performance in chirality-aware tasks, demonstrates explainability for all three relations, and shows effective performance in practical applications. Our code is available in our https://github.com/bowenwang77/OmniMol public repository.

The drug development process is notoriously expensive and time-consuming due to the rigorous multi-stage clinical trials which are necessary to ensure safety and efficacy. Schlander et al. reveal that the estimated research and development costs to bring a new drug to market range from $161 million to over $4.5 billion[1]. Recent advancements in data-driven artificial intelligence (AI)-based computational approaches[2], particularly molecular representation learning (MRL)[3], have shown remarkable proficiency in predicting quantum-level chemical properties. These advances, driven by pretraining on large datasets and advanced auxiliary geometry supervision[4–7], offer promising, efficient, and less risky alternatives for preclinical screening of drug-like molecules[8], demonstrating high potential for easing the burden on traditional molecular synthesis and wet-lab experimentation, thus accelerating the drug development timeline[9]. Notably, early evaluation of absorption, distribution, metabolism, excretion, toxicity, and physicochemical (ADMET-P) properties can significantly reduce

[1]Department of Computer Science and Engineering, The Chinese University of Hong Kong, Ma Liu Shui, Hong Kong, China. [2]Zhejiang Lab, Hangzhou, China. [3]College of Intelligence and Computing, Tianjin University, Tianjin, China. [4]Chemistry and Biochemistry, University of California, Merced, CA, USA. [5]University of Science and Technology of China, Hefei, China. [6]Zhejiang University, Hangzhou, China. [7]Xiangya School of Pharmaceutical Sciences, Central South University, Changsha, China. [8]Hangzhou Institute of Medicine, Chinese Academy of Science, Hangzhou, China. [9]These authors contributed equally: Bowen Wang, Junyou Li. ✉e-mail: jianye.hao@tju.edu.cn; tingjunhou@zju.edu.cn; oriental-cds@163.com; gychen@link.cuhk.edu.hk

drug research and development costs, mitigate the risk of side effects and toxicities[10]. Although several advanced ADMET prediction models, such as ADMETlab 2.0[11], ADMETSAR 2.0[12], HelixADMET[13], and other specialized models have been developed, ADMET prediction remains challenging. Major limitations include the model's generalizability and robustness[14], along with its capability to comprend and analyse complex knowledge in medicinal chemistry, pharmacokinetics, and toxicology, which are essential for providing valuable insights into pharmaceutical sciences[15].

Current research primarily focuses on physics-informed molecular representations and sophisticated message-passing architectures[4,6,16,17], developed based on well-organized datasets containing a vast number of molecules targeting uniformed properties, such as Open Catalyst 2020 (OC20)[18], PubChemQC project dataset (PCQM4MV2)[19], and QM9[20]. However, in real-world scenarios, datasets of molecules and their properties are often imperfectly annotated. Let $\mathcal{M} = \{m_1, m_2, ..., m_{|\mathcal{M}|}\}$ and $\mathcal{E} = \{e_1, e_2, ..., e_{|\mathcal{E}|}\}$ be the set of all molecules and all properties of interest, correspondingly, with $|\mathcal{M}|$ and $|\mathcal{E}|$ denotes the total number of molecules and properties. Each property $e_i \in \mathcal{E}$ is associated with a subset of molecules $\mathcal{M}_{e_i} \subseteq \mathcal{M}$, indicating that only a portion of the molecules are labeled with property $e_i$. Formally, $\mathcal{M}_{e_i} = \{m_j \in \mathcal{M} : m_j \text{ is labeled with property } e_i\}, \forall e_i \in \mathcal{E}$. An imperfectly annotated dataset is characterized by $\exists e_i \in \mathcal{E}$ such that $\mathcal{M}_{e_i} \subsetneq \mathcal{M}$, i.e., there exists at least one property that is not labeled for all molecules. A representative example is the prediction of ADMET-P properties, which are crucial for evaluating the pharmacokinetic and pharmacodynamic profiles that underpin a molecule's potential as a drug candidate[21,22]. As conceptualized in Fig. 1a, ADMET properties are labeled in a scarce, partial, and imbalanced manner due to the prohibitive cost of experimental evaluation.

Imperfectly annotated data could put obstacles in the way of developing AI models for MRL. Firstly, imperfectly annotated data complicate model design. A straightforward strategy (Fig. 2a) is to train a task-specific model for each property[23–31]. However, different properties often contain shared insights that could synergistically enhance prediction performance. For example, Ames and Carcinogenicity are known to be highly correlated tasks[32,33] with partially shared molecules. Joint supervision of these tasks is crucial, which can enhance the dataset size and help the model learn task relations with a better understanding of general druggability. Consequently, prevalent ADMET-P prediction methods[11,13] adopt a multi-task training paradigm (Fig. 2b), employing a shared molecular feature extractor followed by multiple property-specific heads. For instance, ADMETlab 2.0[11] employs a multi-task graph attention (MGA) framework to simultaneously learn both regression and classification tasks for ADMET predictions. This state-of-the-art method utilizes graph neural networks (GNNs) to capture molecular structure information and applies attention mechanisms to focus on relevant structural features for different properties. While the shared backbone extracts common features among molecules, the separate heads often fail to fully capture property relationships, also encountering training synchronization issues due to partial labeling[34]. In addition, task-specific and multiple-head models require $O(|\mathcal{E}|)$ and sub-$O(|\mathcal{E}|)$ complexity, respectively, with the number of models/heads increasing linearly with the number of properties ($|\mathcal{E}|$).

Moreover, limited by imperfect annotated data, the sub-optimal model designing further hampered model explainability. When

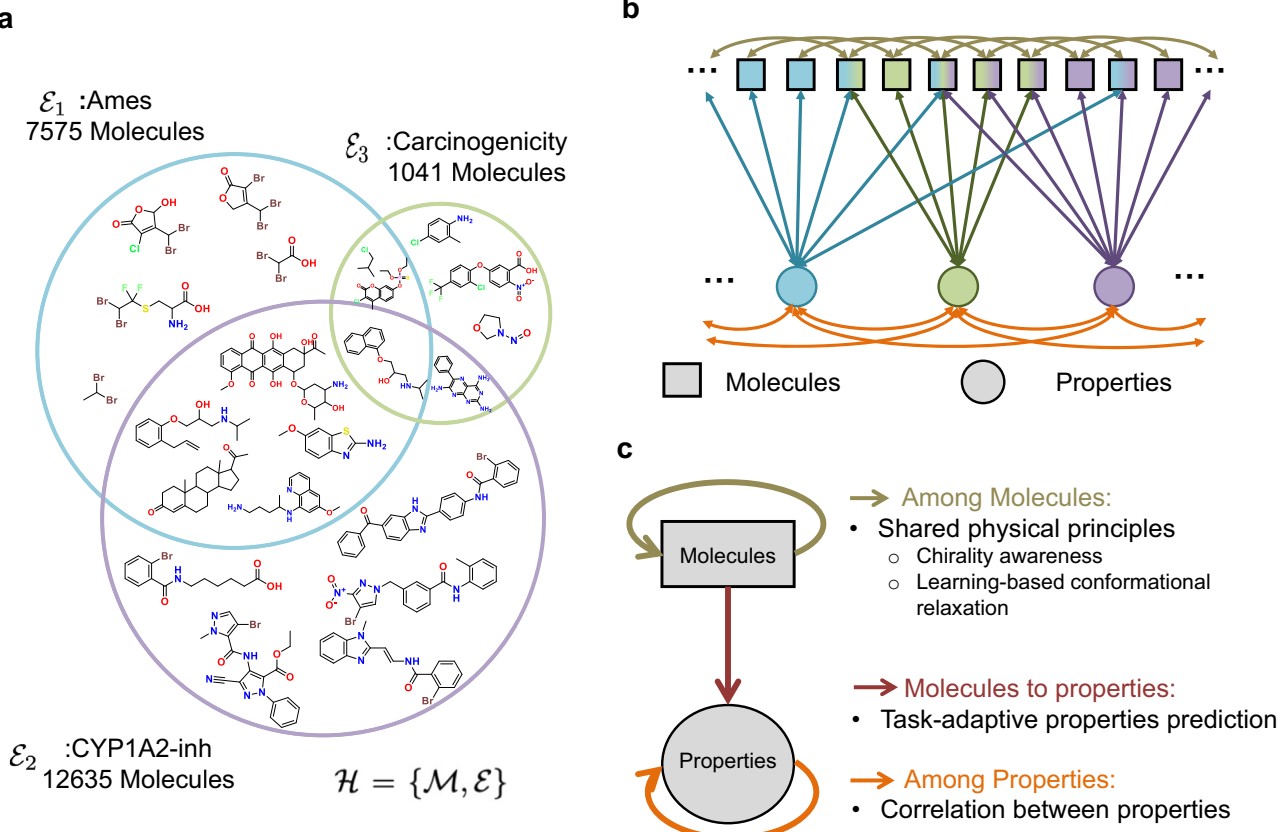

**Fig. 1 | Conceptualization of OmniMol and Hypergraph. a** Hypergraph relation of imperfectly annotated molecules and ADMET-P (absorption, distribution, metabolism, excretion, toxicity, and physicochemical) properties. The hypergraph, denoted as $\mathcal{H}$, is composed of a set of vertices representing molecules ($\mathcal{M}$) and a set of hyperedges representing ADMET-P properties ($\mathcal{E}$). Each large colored circle represents a single property. The overlapping regions illustrate that a single molecule can be associated with multiple properties. **b** The relationship between molecules and properties as a Hypergraph, which is often transformed into a heterogeneous graph that comprises two types of nodes and three types of relationships (**c**).

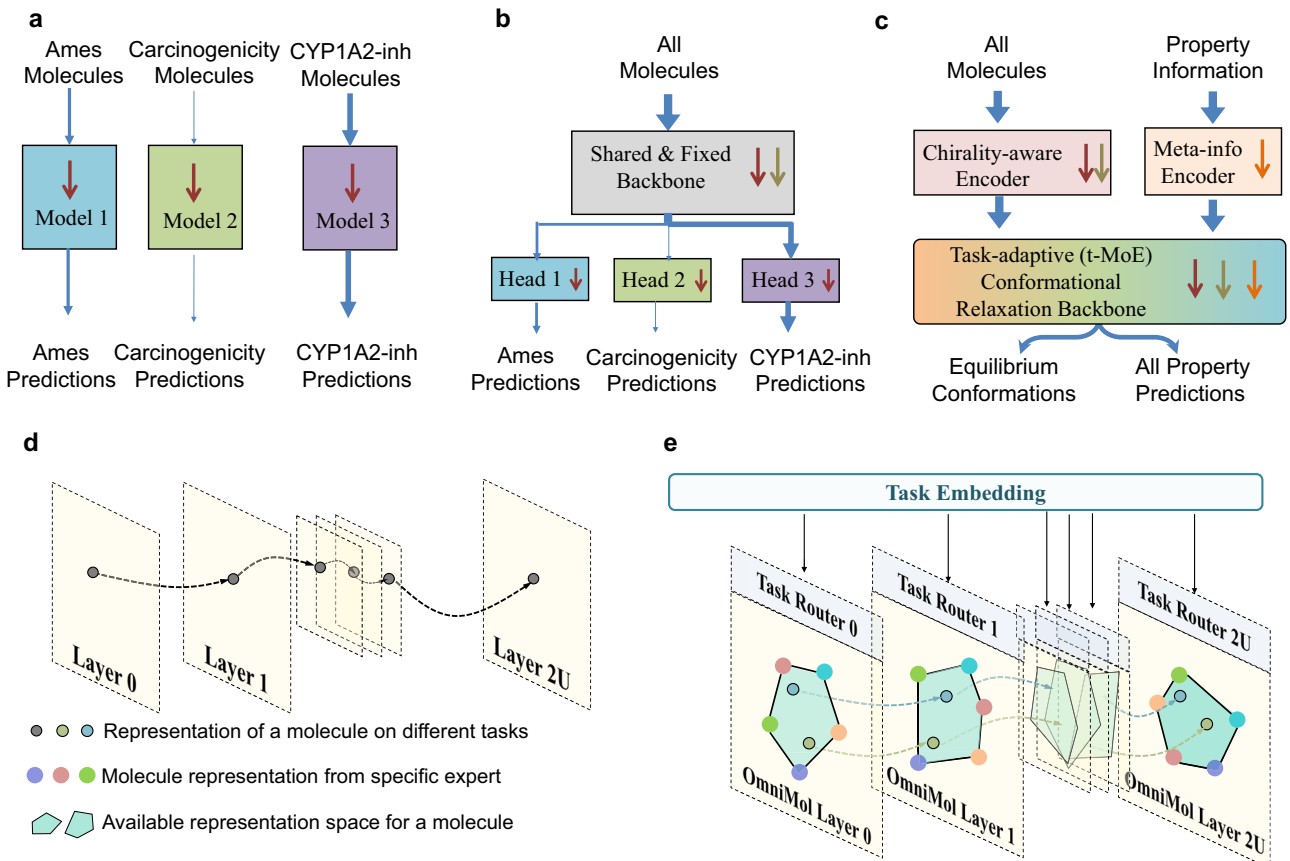

**Fig. 2 | Backbone of OmniMol and its task-routed mixture of experts (t-MoE) architecture. a** Task-specific model for each property. **b** The multi-task training paradigm that employs a shared molecular feature extractor and property-specific heads. **c** OmniMol integrates a specialized encoder and task-routed mixture of experts backbone. **d, e** Comparison of OmniMol and task-agnostic fixed backbone.

When predicting different properties for the same molecule, the task-agnostic fixed backbone will process the representation of the molecule in a fixed way, while the task-adaptive backbone can route the representation of the same molecule to different convex combinations of multiple expert outputs based on task embedding, thereby significantly expanding the available representation space.

conducting practical preclinical data analysis of molecular properties (such as structure-activity relationship study to optimize activity and druggability of lead compounds during the Hit-to-Lead stage), it is essential to understand the rationale behind predictions to ensure the reliability of the results and to derive actionable insights, which are core applications of MRL[35,36]. Besides the highly accurate property prediction performance, a trustworthy deep learning model should also understand fundamental, task-agnostic physical principles shared among all molecule, where large quantity of data is desired. Overcoming the imperfect annotation problem can help merge available molecule-property datasets, drastically increase training data towards the target. At the same time, expect the model to exhibit task-adaptive behavior during predictions, and gain knowledge of correlations among different properties. These aspects have not been sufficiently explored in previous methodological designs due to the imperfect annotation problems.

In summary, a highly unified and explainable multi-task MRL framework is desired to confront imperfect annotation conditions, fully explore dataset potential, and provide reliable predictions. The key lies in carefully designing a data structure that encapsulates the complex many-to-many relation between molecule and property systems. In this work, for the first time, we explicitly encapsulate the available molecular and property data with a hypergraph. Hypergraphs $\mathcal{H} = \{\mathcal{M}, \mathcal{E}\}$ is a generalized graph in which an hyperedge can join any number of nodes. As illustrated in Fig. 1a, we consider the subset of molecules $\mathcal{M}_{e_i} \subseteq \mathcal{M}$ labeled by a property of interest $e_i \in \mathcal{E}$ as a hyperedge. Using topological deep

learning methods[37], this can be further transformed into a heterogeneous graph (Fig. 1b), distinguishing molecules and properties as distinct node types. Fig. 1c highlights the three central types of relations in the graph: among molecules, molecule-to-property, and among properties.

To systematically capture these relationships, building upon a Graphormer[38] structured model, we present OmniMol (Fig. 2c; Fig. 3a–c)[39]. It integrates a specialized encoder to convert task-related meta-information (detailed in Supplementary Table 7) into task embeddings. As detailed in the Methods section and illustrated in Fig. 2d, e, the task embedding is combined with the task-routed mixture of experts (t-MoE) backbone to discern explainable correlations among properties and produce task-adaptive outputs[40]. Our model is able to combine any available molecule-to-property pairs, with the entire end-to-end architecture engaged, bringing better general insight with increased data size. This architecture addresses the issues of imperfect annotation, avoids synchronization difficulties associated with multiple-head models[34], and maintains $O(1)$ complexity independent of the number of tasks.

Further advancements were integrated to ensure OmniMol gains physical principles shared among molecules. We developed a SE(3)-encoder that enables chirality awareness from molecular conformations without expert-crafted features, addressing an important physical symmetry frequently overlooked in existing models[4,17,41–44]. Additionally, we integrate equilibrium conformation supervision[45], recursive geometry updates, and a molecule scale-invariant message-passing strategy, enabling OmniMol to act as a

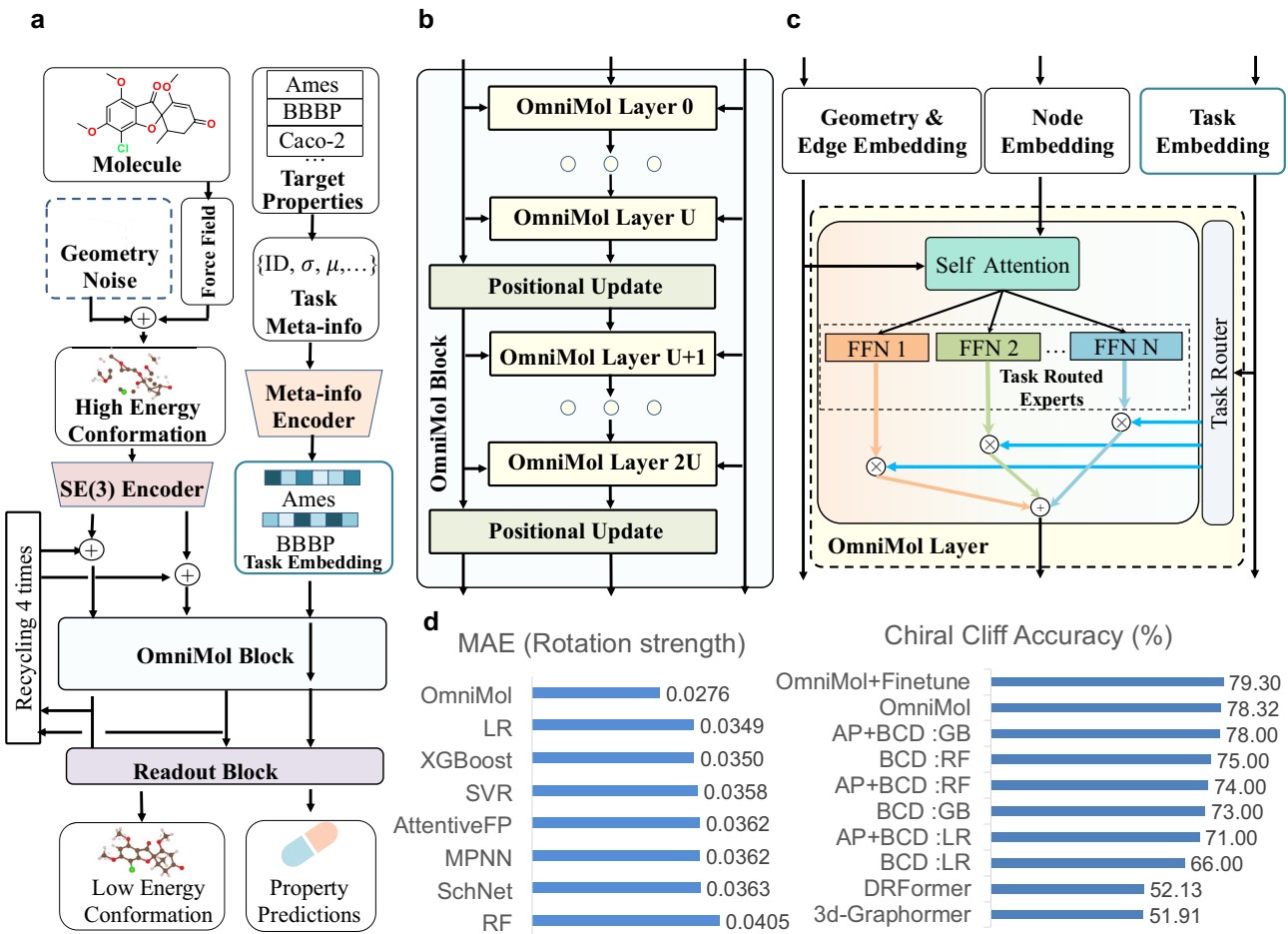

**Fig. 3 | Main architecture of OmniMol and performance on chirality-aware tasks. a** The overall architecture of OmniMol. **b** The core component, OmniMol block, consists of multiple OmniMol Layers, interspersed with the positional update blocks. **c** Each OmniMol layer incorporates a task-routed Mixture of Experts (MoE) layer, facilitating task adaptivity. **d** In chirality-aware tasks, OmniMol surpassed previous methodologies. Abbreviations and symbols: FFN =

FeedForward Neural Network, MAE = Mean Absolute Error, $\mu$ = mean value, $\sigma$ = standard deviation, BCD = best chiral descriptors, AP = AtomPairs Fingerprints, LR = logistic regression, RF = random forest, GB = gradient boosting, XGBoost = Extreme Gradient Boosting, SVR = support vector regression. Other reference methods: SchNet[42], MPNN[60], AttentiveFP[58], DRFormer[17].

learning-based conformational relaxation technique with a heuristic interatomic potential, further ensuring explainable model behavior.

In this work, we use ADMET-P prediction tasks as our main evaluation target, representing the imperfect annotation problem. Our extensive experiments and analyses, benchmarked against established baselines, confirm that OmniMol demonstrates state-of-the-art performance in 47/52 ADMET-P prediction tasks and top performed in chirality awareness tasks. Evidented in our experiments, our model exhibits explainable behavior across all three relationship types: among molecules, molecule-to-property, and among properties. Moreover, OmniMol demonstrates attention distribution performance on specific ADMET-P tasks comparable to or even superior to that of other specialized models, while aligning well with SAR study results in practical applications. The architecture of OmniMol presents a practical approach for applying multi-task representation learning (MRL) to diverse molecular datasets, potentially advancing our understanding of structure-property relationships. While further research is needed to fully validate its broader applicability, OmniMol shows potential for synergistic training on large-scale datasets for atomic systems and property prediction, contributing to the development of more comprehensive and physically informed predictive models in chemistry and drug discovery.

## Results

### Enhancement of representation power with OmniMol

As representative task for imperfectly annotated data, we take ADMET-P prediction as our main target in the following results, and utilized datasets from ADMETLab 2.0[11], as detailed in Supplementary Table 7, to evaluate the efficacy of OmniMol in predicting molecular properties. These datasets comprised approximately 250 k pairs of molecules and corresponding ADMET-P properties, covering 40 classification tasks and 12 regression tasks. However, analysis revealed that the 250k molecule-to-property pairs only consist of 90 k unique molecules; 64.4% of these molecules are associated with a single property label, whereas a minority (53 molecules) are associated with over 30 property labels. In contrast, current models are predominantly developed for well-structured datasets with abundant molecules targeting uniformed properties. Notable examples include the OC20 dataset[18], which contains over 264 million catalysis systems with fully annotated geometry, system energy and atomic force; the PCQM4MV2 dataset[46], consisting of 3.4 million molecules with DFT-calculated HOMO-LUMO gaps; and the QM9 dataset[47], includes 134 k stable small organic molecules with fully labeled geometries and properties. A systematic investigation presented in ref. 48 also highlights the imbalance in drug discovery datasets with respect to the size and distribution of labels.

In solving the imperfectly annotation problem, especially the scarce, partial, and imbalanced data labeling in ADMET-P property prediction, our model equipped the task meta-information encoder and t-MoE module. Further enhancements upon model architectures have been implemented to capture the underlying shared physical principles. A comprehensive description of our architecture is presented in the Methods section. Compared to task-specific models and multiple head models, our model gains a better understanding of all three crucial relations, gaining the common knowledge among molecules and the correlation among properties to facilitate a better molecule-to-property prediction.

We adhered to exactly the same 8:1:1 random data split protocol established by ADMETLab 2.0[11], employed the area under the ROC curve (ROC-AUC) and accuracy as the evaluation metrics for classification tasks, and utilized the $R^2$ correlation factor and mean absolute error (MAE) as the evaluation metrics for regression tasks. Table 1 presents a comprehensive overview of the performance of OmniMol in druggability prediction. In comparison to the baseline model, ADMETLab 2.0, OmniMol demonstrated superior performance across all metrics. Specifically, for classification tasks, OmniMol achieved an average accuracy improvement of over 6.8%, with a majority of 87.5% (35 out of 40) of tasks exhibiting higher performance on both ROC-AUC and accuracy. In regression tasks, there was an 8.83% relative improvement in $R^2$ and a notable 17.2% average relative reduction in MAE, with all tasks exhibiting enhanced performance. We also compared our results with admetSAR 2.0[12] and HelixADMET[13], with the results documented in Supplementary Table 1, where OmniMol also achieved state-of-the-art performance. This exemplary performance can be attributed to the advanced ability of our model to more precisely capture the fundamental pharmacokinetic and pharmacodynamic principles, thereby pinpointing critical factors that influence the biological functionality of a drug. The overall performance also validated the superiority of our method compared to previous task-specific or multi-head methods, overcoming the imperfect annotation problem. The detailed Applicability Domain (AD) of OmniMol in ADMET-P predictions are provided in the Supplementary Section 20.

## Chirality awareness

Chirality is a foundational physical symmetry in molecules that must be captured to attain explainable physical principles, as evidenced by the disparate biological effects of enantiomers. Despite identical chemical composition and atom connectivity, enantiomers can exhibit significantly different therapeutic and toxicological profiles due to their mirror-image structures[49]. A notable case is thalidomide, where the R-enantiomer serves as an effective sedative, while the S-enantiomer has been linked to severe birth defects[50]. Similarly, only one enantiomeric form of Remdesivir can inhibit the RNA-dependent RNA polymerase (RdRp) of coronaviruses, which further shows potential efficacy against SARS-CoV-2[51]. These instances underscore the importance of chirality in pharmaceutical development, a field that is the focus of considerable interdisciplinary research.

Despite advancements in molecular representation learning that incorporate geometric features, there is a prevalent shortfall in the accurate determination of chirality[44]. For example, DRFormer[17] excels in the OC20-IS2RE dataset, but our evaluation (Fig. 3d) indicates its inability to differentiate between chiral pairs. The limitation of DRFormer arises from encoding only interatomic distances, resulting in E(3)-equivariance in node embeddings and E(3)-invariance at the graph level, which keeps all rotation, reflection, and translation groups but does not discriminate against molecular reflections. To address chirality, a model must ensures SE(3)-equivariance in node embeddings and SE(3)-invariance in property predictions, which demand encode geometric features involving at least four atoms[52]. As we implemented in OmniMol, our proposed chirality-aware encoder

enabling sensitivity to reflections. We demonstrate the quantum-level chirality discernment of OmniMol through three experiments.

We first compiled 131k simplified molecular-input line-entry system (SMILES)[53] strings of molecules with a maximum of 12 heavy atoms that contain at least one chiral center from the PubChem[54] database, forming 45,182 enantiomer pairs. For preliminary validation, we isolated 31,270 pairs with a single chiral center, yielding 62,540 molecules (50% R, 50% S), classifying them as R/S according to CIP rules[55] and treated as a binary classification task. These were randomly divided into 80:20 train/test splits. Our model achieved a 96.78% accuracy in R/S chirality prediction, outperforming DRFormer's 50% random prediction accuracy.

In a more intricate task, we evaluated the proficiency of our model in predicting optical rotatory strength, a more complex property that demands quantum-chemical calculations. Utilizing the complete set of 45,182 enantiomer pairs, we used time-dependent density functional theory (TDDFT) to calculate the rotatory strength of each molecule, serving as our regression target. The range-separated functional CAM-B3LYP and the 6-311++G(d,p) basis set were adopted for the TDDFT calculations using the PySCF program[56,57]. With the standard 80:10:10 train/validation/test splits, the results, as detailed in Fig. 3d, exhibit a 21.03% relative decrease in MAE compared to the top result among baseline models[42,58–60], affirming the adeptness of OmniMol in predicting quantum-level properties from mere molecular configurations.

Lastly, we evaluated our performance in predicting binding affinity variations in chiral pairs. We used a dataset from ref. 49 to analyze the chiral cliff, a measure of drug efficacy variation due to chirality. Adhering to an 8:2 training/test split from previous studies, OmniMol demonstrated a 78.3% accuracy in identifying molecules with significant chiral binding affinity variances, improving to 79.3% upon fine-tuning with the ADMET-P prediction dataset. The performance is on-par with the 78% accuracy of the best chiral descriptor-based method noted in the referenced study, achieved without meticulously designed chiral descriptors. In contrast, 3d-graphormer[38] and DRFormer reached only random prediction accuracy (51.9% and 52.1% respectively). These results validate that OmniMol gains chirality awareness in drug candidate screening.

## Learning-based conformational relaxation

In drug discovery, the accurate representation of molecular dynamics, informed by interatomic potential, is crucial for understanding the complex molecular interactions within binding environments that affect pharmacodynamics and pharmacokinetics, and is an important knowledge shared among molecules to reach model explainability. OmniMol, while inspired by the principles that govern interatomic potentials, diverges from traditional force fields. It functions as a heuristic conformational relaxation algorithm that predicts low-energy molecular conformations without directly modeling the forces or potential energy surface.

As detailed in the Methods section and illustrated in Fig. 3, to incorporate such physical insight into the model, OmniMol begins by introducing 3D Gaussian geometry noise to the equilibrium conformation of a molecule—determined using the Merck Molecular Force Field (MMFF)—to represent a high-energy molecular conformation. It then employs a geometry prediction head in parallel with a graph-level property head to iteratively guide the molecule toward the low-energy equilibrium conformation. The architecture features a recursively explicit geometry update mechanism and a molecular scale-invariant message-passing strategy, enhancing the capability of our model to emulate molecular dynamics during conformational relaxation. A molecule scale-invariant message-passing strategy shared among molecules is further equipped to help ensure a reasonable geometry update for a more reliable molecule-to-property relation.

To elucidate the model behavior when processing each high-energy conformation during message updating, we extracted the

**Table 1 | Performance of OmniMol on predicting ADMET-P property**

| Classification Tasks | | | | | | | |
|---|---|---|---|---|---|---|---|
| Category | Task Name | ROC-AUC | | ACC | | MCC | |
| | | ADMETlab 2.0 | OmniMol | ADMETlab 2.0 | OmniMol | ADMETlab 2.0 | OmniMol |
| Absorption | Pgp-inh | 0.922 | 0.942 | 0.867 | 0.907 | 0.723 | 0.799 |
| | Pgp-sub | 0.840 | 0.907 | 0.768 | 0.840 | 0.538 | 0.631 |
| | HIA | 0.866 | 0.940 | 0.924 | 0.949 | 0.687 | 0.697 |
| | $F_{20\%}$ | 0.833 | 0.933 | 0.750 | 0.880 | 0.414 | 0.617 |
| | $F_{30\%}$ | 0.848 | 0.910 | 0.802 | 0.891 | 0.580 | 0.678 |
| Distribution | BBBP | 0.908 | 0.922 | 0.862 | 0.853 | 0.718 | 0.748 |
| Metabolism | CYP1A2-inh | 0.928 | 0.934 | 0.852 | 0.885 | 0.704 | 0.791 |
| | CYP1A2-sub | 0.737 | 0.976 | 0.649 | 0.892 | 0.298 | 0.618 |
| | CYP2C19-inh | 0.913 | 0.924 | 0.839 | 0.857 | 0.679 | 0.773 |
| | CYP2C19-sub | 0.758 | 0.958 | 0.654 | 0.923 | 0.300 | 0.517 |
| | CYP2C9-inh | 0.919 | 0.919 | 0.841 | 0.872 | 0.671 | 0.735 |
| | CYP2C9-sub | 0.725 | 0.902 | 0.707 | 0.866 | 0.386 | 0.639 |
| | CYP2D6-inh | 0.892 | 0.917 | 0.824 | 0.898 | 0.558 | 0.693 |
| | CYP2D6-sub | 0.847 | 0.903 | 0.775 | 0.865 | 0.553 | 0.743 |
| | CYP3A4-inh | 0.921 | 0.922 | 0.832 | 0.848 | 0.659 | 0.721 |
| | CYP3A4-sub | 0.776 | 0.810 | 0.713 | 0.772 | 0.437 | 0.478 |
| Excretion | $T_{1/2}$ | 0.801 | 0.851 | 0.727 | 0.771 | 0.478 | 0.525 |
| Toxicity | hERG inhibition | 0.943 | 0.942 | 0.889 | 0.887 | 0.778 | 0.830 |
| | Hepatotoxicity | 0.814 | 0.794 | 0.720 | 0.728 | 0.461 | 0.415 |
| | DILI | 0.924 | 0.932 | 0.894 | 0.915 | 0.793 | 0.807 |
| | Ames mutagenicity | 0.902 | 0.907 | 0.807 | 0.845 | 0.606 | 0.707 |
| | Rodent Acute Toxicity | 0.853 | 0.835 | 0.778 | 0.777 | 0.549 | 0.547 |
| | FDAMDD | 0.804 | 0.837 | 0.736 | 0.793 | 0.471 | 0.522 |
| | SkinSen | 0.707 | 0.842 | 0.775 | 0.875 | 0.462 | 0.340 |
| | Carcinogenicity | 0.788 | 0.806 | 0.731 | 0.769 | 0.476 | 0.507 |
| | EC | 0.983 | 0.996 | 0.957 | 0.978 | 0.908 | 0.980 |
| | EI | 0.982 | 0.982 | 0.952 | 0.962 | 0.876 | 0.955 |
| | Respiratory | 0.828 | 0.876 | 0.764 | 0.821 | 0.514 | 0.639 |
| | NR-AR | 0.886 | 0.931 | 0.890 | 0.985 | 0.348 | 0.717 |
| | NR-AR-LBD | 0.915 | 0.934 | 0.936 | 0.983 | 0.472 | 0.720 |
| | NR-AhR | 0.943 | 0.952 | 0.862 | 0.937 | 0.573 | 0.731 |
| | NR-Aromatase | 0.852 | 0.884 | 0.849 | 0.961 | 0.264 | 0.320 |
| | NR-ER | 0.771 | 0.837 | 0.815 | 0.921 | 0.320 | 0.498 |
| | NR-ER-LBD | 0.850 | 0.915 | 0.903 | 0.966 | 0.364 | 0.590 |
| | NR-PPAR-γ | 0.893 | 0.896 | 0.896 | 0.979 | 0.344 | 0.590 |
| | SR-ARE | 0.863 | 0.888 | 0.827 | 0.901 | 0.469 | 0.606 |
| | SR-ATAD5 | 0.874 | 0.867 | 0.919 | 0.975 | 0.361 | 0.365 |
| | SR-HSE | 0.907 | 0.912 | 0.868 | 0.959 | 0.393 | 0.598 |
| | SR-MMP | 0.927 | 0.953 | 0.897 | 0.940 | 0.660 | 0.800 |
| | SR-p53 | 0.881 | 0.911 | 0.841 | 0.960 | 0.365 | 0.599 |
| Mean | | 0.863 | 0.905 | 0.822 | 0.890 | 0.530 | 0.645 |
| Regression Tasks | | | | | | | |
| Catagory | Task Name | $R^2$ | | MAE | | RMSE | |
| | | ADMETlab 2.0 | OmniMol | ADMETlab 2.0 | OmniMol | ADMETlab 2.0 | OmniMol |
| Physiochemicalproperty | LogS | 0.854 | 0.878 | 0.588 | 0.509 | 0.850 | 0.741 |
| | LogD | 0.892 | 0.924 | 0.347 | 0.290 | 0.462 | 0.371 |
| | LogP | 0.957 | 0.964 | 0.256 | 0.223 | 0.357 | 0.309 |
| Absorption | Caco-2 | 0.746 | 0.886 | 0.222 | 0.195 | 0.307 | 0.275 |
| | MDCK | 0.731 | 0.801 | 0.199 | 0.179 | 0.291 | 0.251 |
| Distribution | PPB | 0.733 | 0.856 | 0.083 | 0.058 | 0.135 | 0.094 |
| | VDss | 0.782 | 0.848 | 0.457 | 0.347 | 0.670 | 0.660 |
| | Fu | 0.763 | 0.848 | 0.263 | 0.197 | 0.367 | 0.310 |
| Toxicity | BCF | 0.786 | 0.800 | 0.435 | 0.416 | 0.603 | 0.591 |

**Table 1 (continued) | Performance of OmniMol on predicting ADMET-P property**

| Classification Tasks | | | | | | |
|---|---|---|---|---|---|---|
| $IGC_{50}$ | 0.723 | 0.858 | 0.335 | 0.232 | 0.496 | 0.360 |
| $LC_{50}$ | 0.745 | 0.789 | 0.643 | 0.532 | 0.863 | 0.782 |
| $LC_{50}DM$ | 0.524 | 0.716 | 0.692 | 0.566 | 0.994 | 0.821 |
| Mean | 0.770 | 0.844 | 0.377 | 0.312 | 0.533 | 0.464 |

For more details on the endpoints, please refer to Supplementary Table 6.

explicitly updated intermediate geometry after each recursively recycled OmniMol block shown in Fig. 3a. We visualize and analyze two representative molecules in Fig. 4a, where the blue curve indicates the node-wise mean absolute geometry distance to the equilibrium conformation, and the red curve shows the system energy as computed by MMFF. First, we observe that OmniMol effectively relaxed the high-energy conformations. More importantly, there is a significant contrast between the steady decrease in energy and the non-monotonic behavior of the geometry MAE. This underscores that OmniMol is not merely interpolating between noisy and equilibrium states, although supervised solely by the equilibrium geometry. Instead, it employs learned transformations that reflect the outcomes of the atomic interactions, iteratively directing atoms to form low-energy conformations, even if the result may not be geometrically proximate to the equilibrium state (e.g., from initial conformation to block-1 conformation). This ability to predict energetically favorable conformations, without explicitly calculating interatomic forces, is of particular importance for drug discovery, showing that OmniMol learned a heuristic interatomic potential for conformational relaxation, which is shared among tasks and instances.

### Task-relation-aware adaptive for ADMET-P prediction

In the construction of the molecule-to-property relationship, a particularly noteworthy finding is the ability of the task meta-information encoder in OmniMol to identify intrinsic connections among properties. In the t-MoE framework, OmniMol is designed to allocate specialized experts to distinct tasks based on the task embedding. This approach allows the model to tailor its behavior specifically for each unique property, moving beyond a generic approach and learning optimal model parameters tailored to each task (Fig. 2d, e). For correlated tasks, the model exhibits analogous attention distributions on molecules.

To examine this ability, we conducted a detailed analysis of the task embeddings produced by the task meta-information encoder for each task ($t_i$). These task embeddings, optimized through the training process, can be considered to encapsulate vital insights on property-property relationships. They facilitate the soft assignment of an appropriate set of experts, effectively utilizing the hypergraph connections of molecules and enhancing performance. Employing the uniform manifold approximation and projection (UMAP)[61] algorithm for dimension reduction on these task embeddings ($t_i$), we explored the underlying relationships among all tasks. The results, as depicted in Fig. 4b, c, offer several compelling insights.

Firstly, the six different groups of ADMET-P properties are distinctly segregated into separate clusters in the representation space. Additionally, there is a discernible clustering pattern distinguishing regression tasks from classification tasks. Upon closer examination, we noted that in metabolism tasks, properties related to substrates of cytochrome P450 enzymes (CYPs-sub) are distinctly positioned in the embedding space compared to their inhibitors (CYPs-inh), even without prior knowledge input. More importantly, several closed related tasks such as {DILI and H-HT}, {Ames and Carcinogenicity}, and {EI and EC} are automatically grouped into proximal positions within the embedding space. These findings corroborate our hypothesis that the task meta-information encoder can effectively discern the inherent relationships between various property prediction tasks.

To further examine the behavioral mechanism of our model when tasked with different endpoints, we selected two sets of ADMET-P prediction tasks to conduct a qualitative experiment. Each set consists of four endpoints, where two of them are closely related to each other while showing less relevance to the other two endpoints. The first set includes {Ames, Carcinogenicity, CYP1A2-inh, CYP3A4-inh}, as indicated by the orange range in the UMAP visualization (Fig. 4b) and in the orange box of Fig. 4c. The second set comprises {F_{30%}, F_{20%}, H-HT, DILI}, shown by the blue range and in the blue box. We identified and observed the intersections within the two sets of molecules and visualized the results. Based on the task embedding, the molecule receives varying node-wise attention, which is influenced by the proportion of the mixture of expert modules assigned by the task router.

As illustrated in Fig. 4c, established pharmaceutical knowledge indicates a strong correlation between Ames and Carcinogenicity, as well as a link between CYP1A2-inhibitors and CYP3A4-inhibitors. For Metronidazole (1st row), our model reveals similar node attention patterns for related tasks, even without prior knowledge input. Specifically, the nitro group is a crucial group that may induce gene mutation or cancer[62], while nitrogen-containing heteroaromatic molecules may inhibit CYP450 enzymes[63]. Notably, these patterns differ significantly between the two different task groups, and this differentiation is not merely a random derivation from prior ADMET-P group knowledge. For Procarbazine (2nd row), the hydrazino group shows a detrimental impact on molecular druggability, which prompts OmniMol to allocate high attention across all tasks to this group, reflecting a nuanced understanding of ADMET-P prediction. Similarly, a common pattern emerges in the second set, where OmniMol differentiates attention patterns for distinct tasks (3rd row) and pinpoints crucial substructures affecting druggability (4th row). These findings underscore the capability of OmniMol to discern and utilize inherent task relationships in ADMET-P prediction, while seamlessly adapting to specific tasks and modulating node attention in response to target variables. Concurrently, OmniMol retains fundamental pharmacological principles throughout this adaptive process. In conclusion, OmniMol is a model that assimilates universal knowledge across various tasks and exhibits task-specific adaptability for optimal performance.

### Validation by comparative study and explainable attention distribution

Based on the above, we observed OmniMol's excellent performance in task-related molecular attention distribution. To validate OmniMol's explainability and chemical reasoning capabilities, we analyzed its node-level attention distributions across eight distinct ADMET-P endpoints. Our analysis demonstrates that OmniMol's attention mechanisms align with established pharmaceutical knowledge and generate interpretations comparable to or even superior to specialized single-task models. The model's ability to capture structure-property relationships was evaluated through the node-level attention values applied prior to generating graph-level predictions.

As shown in Fig. 5, we visualized atomic attention distributions for three representative molecules per endpoint, comparing OmniMol's

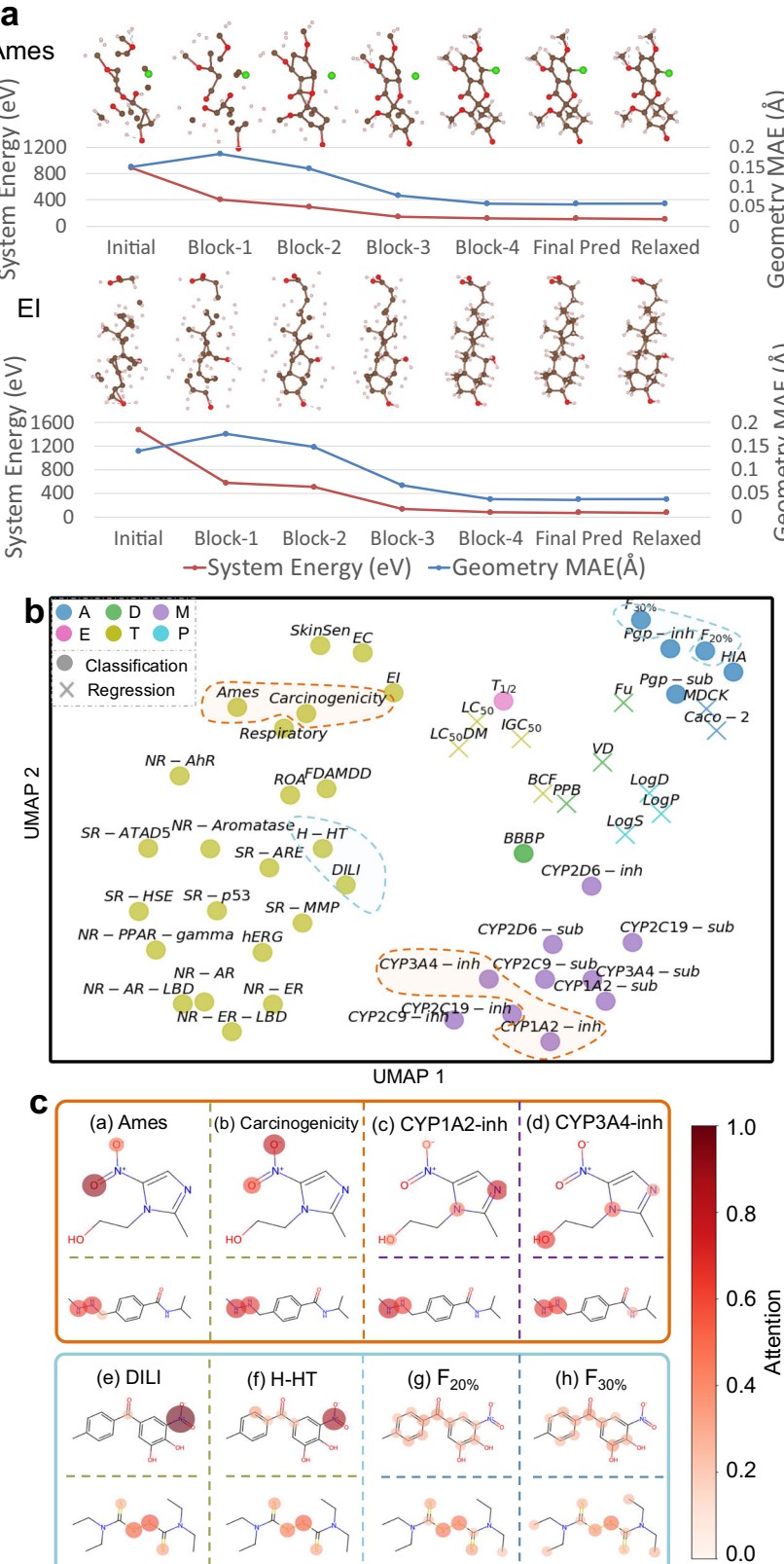

**Fig. 4 | Inspecting the behavior of OmniMol. a** Visualization of intermediate geometry demonstrates that OmniMol behaves as a heuristic interatomic potential that aids molecules in achieving their low-energy conformations. **b** UMAP visualization shows OmniMol exhibits effective awareness of task relationships as it effectively categorizes different ADMET-P properties, with related representative endpoints located in close proximity. Different colors represent different major categories of ADMET-P tasks, such as Absorption or Distribution. The two tasks enclosed by dashed lines are considered pharmacologically related and highlighted as case studies. **c** OmniMol adaptively assigns different attention patterns according to the specific property prediction tasks. Abbreviations: MAE = Mean Absolute Error, UMAP = Uniform Manifold Approximation and Projection, ADMET-P = Absorption (A), Distribution (D), Metabolism (M), Excretion (E), Toxicity (T), and Physicochemical (P).

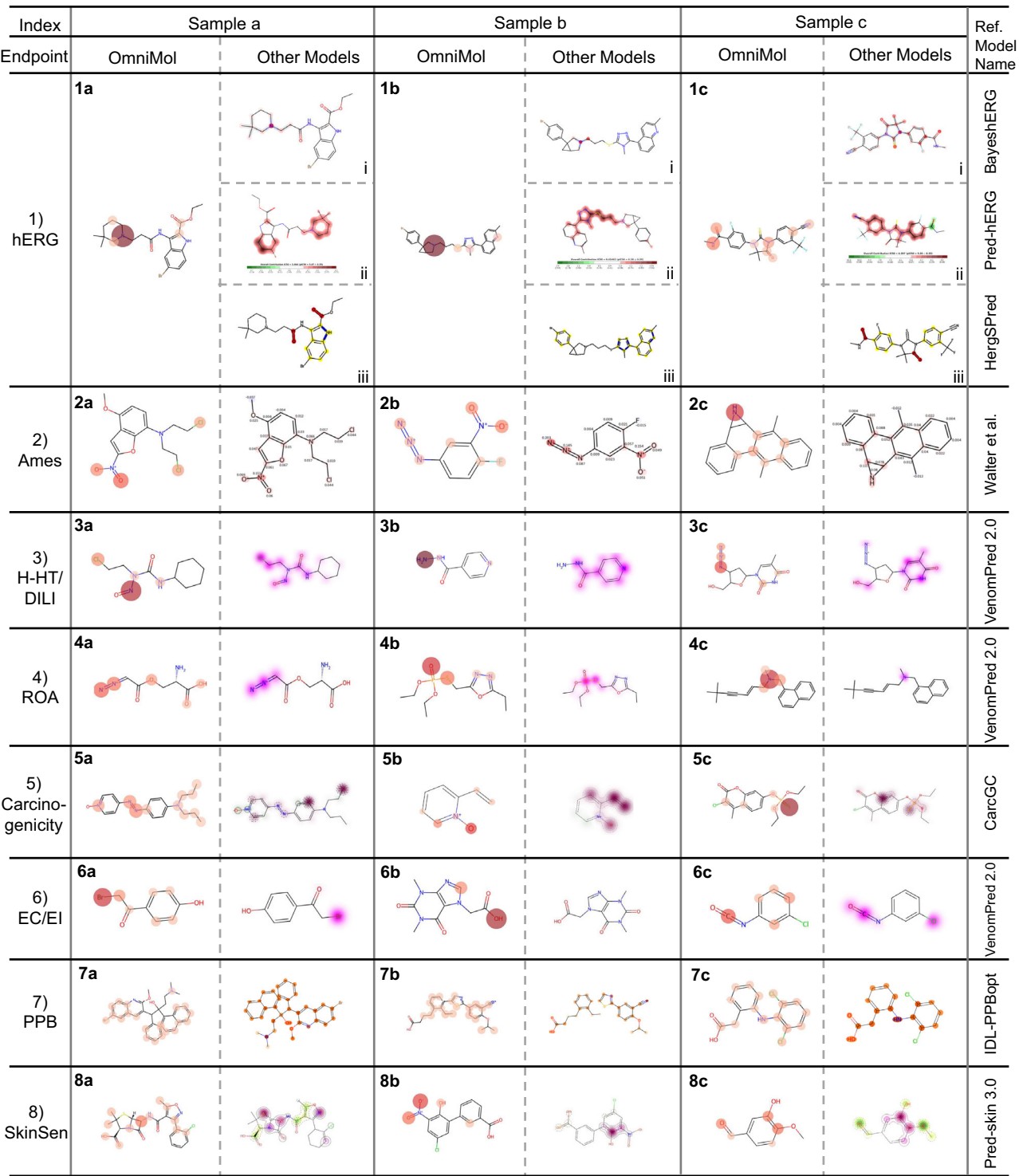

**Fig. 5 | Atomic attention distributions in OmniMol compared to specialized models.** For each endpoint, three representative molecules demonstrate Omni-Mol's ability to identify task-relevant structural features. Color intensity represents attention values, with darker colors indicating higher attention values. Reference methods: BayeshERG[65], Pred-hERG[75], HergSPred[28], Walter et al.[76], VenomPred2.0[77], CarcGC[78], IDL-PPBopt[79], Pred-skin 3.0[80]. Abbreviations: hERG = human Ether-á-go-go-Related Gene, this refers to hERG inhibition, Ames = Ames mutagenicity test, H-HT = Human Hepatotoxicity, DILI = Drug-Induced Liver Injury, ROA = Rat Oral Acute Toxicity, EC = Eye Corrosion, EI = Eye Irritation, PPB = Plasma Protein Binding, SkinSen = Skin Sensitization. For more details on the endpoints, please refer to Supplementary Table 6.

attention patterns with those of established specialized models. For instance, in hERG inhibition prediction, OmniMol effectively identifies tertiary amine, amino groups, and polar groups, which are structural features known to influence hERG inhibition[64]. This aligns closely with specialized models like BayeshERG[65], demonstrating very similar

attention patterns. Similarly, for acute toxicity assessment, both OmniMol and specialized models like VenomPred 2.0 correctly emphasize azo and phosphate ester groups, which are established toxicophores[66]. Through comparative analysis of attention distribution patterns, three distinct levels of correspondence between

OmniMol and specialized models were defined: high similarity, moderate similarity, and partial similarity. Notably, the majority of comparisons exhibited high to moderate similarity, indicating that OmniMol successfully identifies chemically relevant structural features without explicit feature engineering. The whole analysis of attention patterns for all endpoints, including detailed comparisons with specialized models and chemical reasoning, is provided in Supplementary Section 18.

Importantly, a key distinction of OmniMol is its ability to operate directly on 3D conformations that generated from SMILES strings, unlike most specialized models that rely on pre-computed molecular descriptors (e.g., fingerprints, substructures, or physicochemical properties). Through learned embeddings and attention mechanisms, OmniMol autonomously identifies task-relevant functional groups. The node-level attention architecture enables OmniMol to focus on critical functional groups essential for specific properties while avoiding excessive attention to other molecular fragments, addressing the limitations of conventional sum-aggregated graph-level representations that may struggle with varying molecular sizes and complexities. In contrast, while models utilizing pre-computed molecular descriptors can effectively encode established structural patterns and rules, they may be constrained in capturing complex ADMET phenomena and adapting to different chemical space and scenarios[67], thereby limiting their practical applications. OmniMol's SMILES-based approach combines adherence to established chemical principles with the flexibility to learn new structure-property relationships, positioning it as a robust framework for comprehensive ADMET-P prediction.

### Practical applicability demonstrated through real-world SAR studies

Encouraged by OmniMol's convincing performance as described above, we conducted comprehensive structure-activity relationship (SAR) studies on four pivotal ADMET properties to validate the model's practical utility in molecular optimization. As presented in Fig. 6, to rigorously test generalization capability, we selected multiple compounds not included in our dataset, except for compounds **11a** and **12b**, from published SAR studies[68–70]. Interestingly, across all four endpoints, OmniMol's performance in attention distribution aligns with the results of SAR studies, even for subtle structural changes. In other words, OmniMol assigns high attention values to endpoint-revelant groups in the pre-optimized molecules, while the attention on modified groups is reduced or completely eliminated in the corresponding optimized molecules.

Specifically, in hERG inhibition cases, OmniMol accurately detected key sites involving basic centers and molecular rigidity. In the pre-optimized molecules (**10a** and **11a**), OmniMol assigned high attention values to tertiary amines at ring junctions, which are known to influence hERG inhibition[64]. While the post-optimized molecules (**10b** and **11b**) showed a significant reduction in attention or complete disappearance at the corresponding sites, aligning with established strategies for mitigating hERG inhibition. In contrast, BayeshERG and other models seem to be rigid, as their attention distribution is almost identical before and after optimization (Supplementary Fig. 3). Similarly, OmniMol detected subtal atomic-level changes that affect efflux ratio (ER) of MDCK-MDR1, which is a key standard for identifying whether a molecule is P-glycoprotein substrate[71].

A detailed analysis of all optimization cases, including comparative performance against specialized models and comprehensive structure-property relationships for each endpoint, is provided in the Spplementary Section 19. Notably, OmniMol demonstrated adaptability to diverse structural modifications while maintaining consistent chemical reasoning across different property optimization scenarios. These results validate OmniMol's potential as a practical tool for ADMET property prediction and optimization in drug development, as OmniMol not only aligns with established pharmaceutical knowledge,

but also integrates the ability to identify subtle structural changes in practical SAR studies.

## Discussion

In this study, we have introduced OmniMol, a practical molecular representation learning framework that leverages a hypergraph perspective to capture the complex relationships among molecules, molecule-to-property, and among properties in imperfectly annotated datasets, which are prevalent in real-world scenarios. Our use of ADMET-P prediction tasks demonstrates the effectiveness of OmniMol in addressing the challenges posed by sparse, partial, and imbalanced annotations. Our extensive experiments and analyses confirm OmniMol's outstanding performance in ADMET-P predictions, robust chirality sensitivity, and high interpretability across all three relationship types.

The significance of OmniMol extends beyond ADMET-P assessment for drug development. The ability to combine available molecule-to-property pairs and engage the entire end-to-end architecture brings better general insight with increased data size. This lays the groundwork for applying generalized molecular representation learning across diverse datasets, thereby capturing fundamental physical mechanisms.

The visualization of meta-embedding learned with UMAP demonstrates that OmniMol learns meaningful task representations beyond the original dataset statistics. The learned embeddings capture both explicit categorical relationships (ADMET-P groupings) and implicit biological connections. Highly correlated endpoints are distributed in proximal space, such as H-HT and DILI, while maintaining a structured organization that reflects the fundamental nature of the prediction tasks (regression vs. classification). This provides strong evidence that the meta-embedding process effectively discovers and encodes relevant task relationships, enhancing the model's ability to transfer knowledge across related tasks.

OmniMol's performance in ADMET-P property prediction tasks underscores its ability to generate meaningful and interpretable attention distributions. Comparative studies with existing specialized models across eight ADMET-P prediction tasks show that OmniMol achieves competitive or superior performance. Unlike most specialized models that rely on predefined physicochemical descriptors and substructure fingerprints to capture known pharmaceutical patterns, The input for OmniMol stems from SMILES strings and involve 3D conformation information, with hypergraph topological connection involved among different datasets, enabling it to comprehend pharmaceutical patterns and knowledge. This approach enhances the flexibility and adaptability of the model, allowing it to excel in real-world applications. Notably, OmniMol accurately identifies critical chemical groups and substructures influencing corresponding properties, aligning strongly with established pharmaceutical knowledge.

The detailed analysis of multiple SAR optimization cases across four ADMET-P tasks highlights OmniMol's ability to assign high attention values to suboptimal groups in pre-optimized molecules while reducing or eliminating attention to these corresponding sites in post-optimized molecules. This result is particularly remarkable given that these examples are almost entirely different from those in OmniMol. Even when the molecular changes before and after optimization are minimal, OmniMol exhibits strong adaptability, practicality, and generalization capabilities. These strengths enable OmniMol to surpass the limitations of fixed-group recognition, establishing it as a highly flexible and effective tool for molecular property prediction and optimization. Its potential extends to broader and more practical applications, such as the highly challenging and resource-intensive SAR optimization phase in preclinical studies.

Despite its strengths, OmniMol have some limitations, particularly with respect to the domain of supported data. Given that the training data for the classification tasks consist of binary values, OmniMol can't

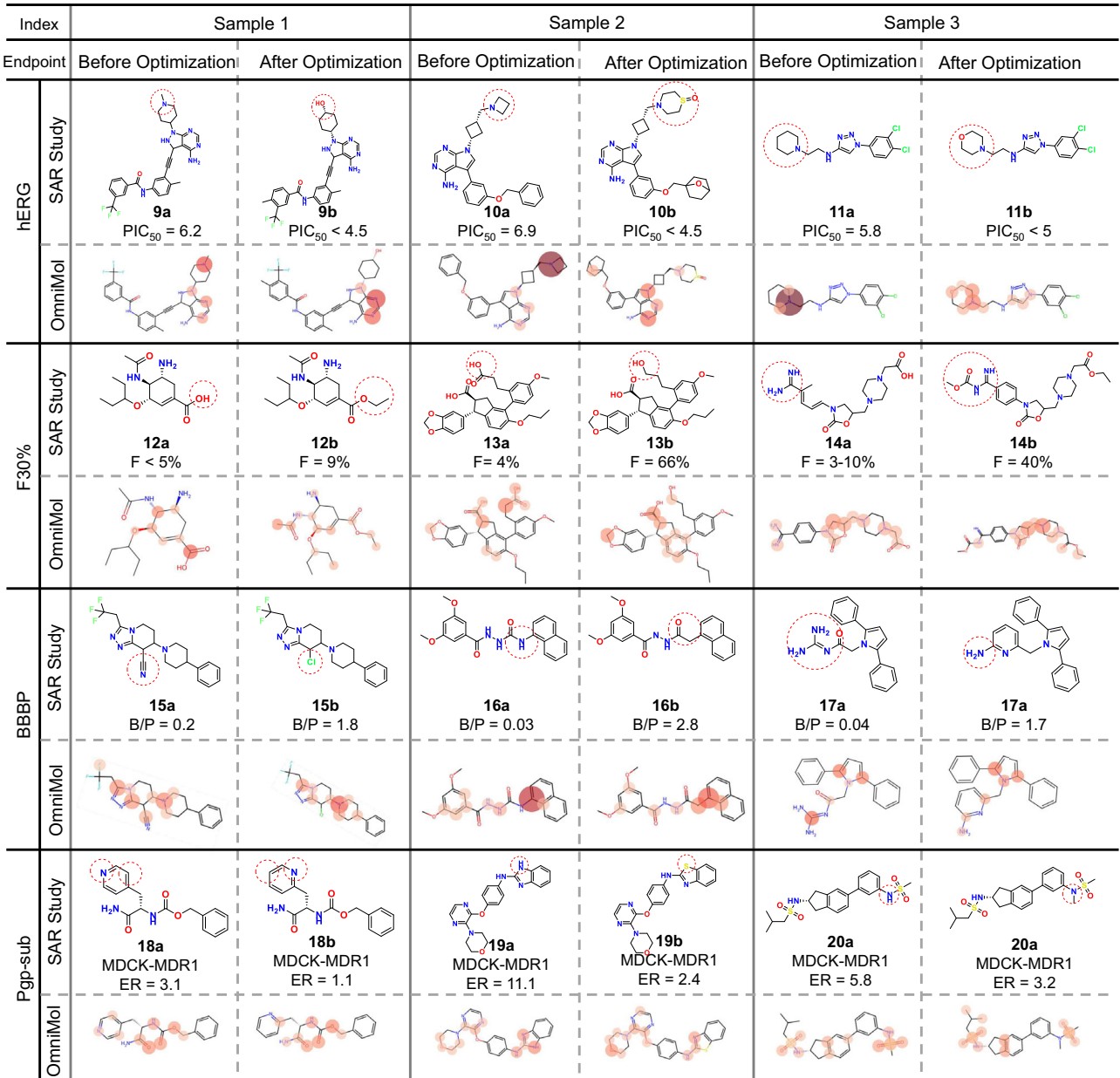

**Fig. 6 | Structure-activity relationship (SAR) analysis of OmniMol in real-world case study.** OmniMol demonstrates the capability to identify key structural features relevant for optimization across four critical ADMET-P properties. Attention weights are represented by color intensity, with darker colors indicating higher attention values. Red circles highlight key modification sites. Abbreviations and symbols: pIC$_{50}$ = the negative log of the IC$_{50}$ value when converted to molar, F = oral bioavailability, B/P = Brain-to-Plasma Concentration Ratio, MDCK-MDR1 = Madine-Darby Canine Kidney - Multidrug Resistance Protein 1, ER = efflux ratio, hERG = human Ether-á-go-go-Related Gene, this refers to hERG inhibition, F$_{30\%}$ = oral bioavailability at 30%, BBBP = Blood-Brain Barrier Permeability, Pgp-sub = P-glycoprotein substrate, ADMET-P = absorption, distribution, metabolism, excretion, toxicity, and physicochemical.

accurately predict the actual experimental values on these tasks. It is important to note that our work is retrospective, and while rigorous validations have demonstrated its utility, future studies should focus on incorporating prospective experimental scenarios to further establish its practical value. Moreover, currently, the framework is designed to handle datasets consisting of small drug-like molecules. Expanding on our empirical findings, future work could extend OmniMol's applicability to more diverse and complex datasets. For example, OmniMol could be adapted to predict properties such as HOMO-LUMO gaps in small molecules[46], adsorption energies in catalytic systems[18], and force field predictions for biological molecules[72]. Furthermore, the framework could potentially be scaled to macro-scale systems, including industrial simulations such as car or airfoil

design[73]. By leveraging these additional physical properties, OmniMol could evolve into a comprehensive "world model" that supports both simulation-driven ADMET-P assessments and serves as a physics-informed foundation for other AI-driven generative techniques. Such advancements would significantly contribute to the development of more comprehensive, physically feasible, and effective drug candidates.

In summary, OmniMol represents a significant advancement in molecular property prediction, with its robust adaptability, interpretability, and generalization capabilities. It not only demonstrates reliable accuracy in traditional ADMET-P prediction tasks but also provides a framework for broader applications in molecular and material sciences. Future extensions of OmniMol could further

enhance its impact, transforming it into a versatile tool for both research and practical applications in drug discovery and beyond.

## Methods

The conceptual framework of OmniMol is depicted in Fig. 2, with an in-depth workflow outlined in Fig. 3a. The initial input for each candidate includes the SMILES[53] strings of molecules along with pertinent task meta-information relevant to the target task at hand. The processing of task meta-information through a dedicated encoder generates a task embedding, as the guidance for our task-adaptive message passing. The SMILES strings of drug candidates are first converted into high-energy conformations. These conformations are further encoded by an SE(3)-encoder to ensure a chirality-aware initial molecular representation, capturing detailed geometric, edge, and node features. This comprehensive integration of data is subsequently channeled through 4 times recycling of the OmniMol Block, which collaborates with a final readout block to refine the representation of molecule towards its low-energy conformation, thereby aiding accurate predictions of its molecular properties. The meta-information of each task is also used in the data loader and loss function for dealing with the imbalanced data labeling problem. Detailed discussions on the OmniMol architecture are presented in the following sections. Assessments of the critical components of the model and their contributions to its overall effectiveness are presented in Supplementary Table 4.

### Task-adaptive learning with mixture-of-experts architecture

A key innovation of OmniMol is to balance the broad applicability among properties and fully utilize all available datasets for common knowledge among molecules. This calls for a model with task-specific adaptability. As illustrated in Fig. 3, this balance is achieved through the implementation of the proposed task-routed mixture-of-experts (t-MoE) architecture. The framework draws an analogy to a multi-disciplinary team of experts, where each member contributes specialized knowledge towards a comprehensive assessment. Similarly, OmniMol utilizes task embeddings that guide the focus of the model, which encapsulates vital meta-information for tasks for nuanced decision-making.

At the core of each OmniMol layer, a task-agnostic self-attention mechanism is employed to learn universal parameters that are broadly applicable to all ADMET-P prediction tasks. $N_\mathbf{M}$ denotes the number of parallel feed-forward networks (FFNs), each representing an "expert" with unique parameters, are integrated within the layer. Task meta-information $t_{\mathrm{meta}} \in \mathbb{R}^{d_{\mathrm{meta}}}$ will be associated along with each molecule, includes the following information, and is detailed in Supplementary Table 7:

- ID: Each task is associated with an integer index.
- Statistics: For classification tasks, the input includes the proportion of positive samples $\eta$ within the training dataset. For regression tasks, it incorporates the mean $\mu$ and standard deviation $\sigma$ of the target property from the training set.
- Group: An indicator specifying the belonging relation of a task with its supergroup. In the ADMET-P prediction context, it is an indicator of the task as absorption, distribution, metabolism, excretion, toxicity, or physicochemical property prediction.
- Type: An identifier of the task as either regression or classification.

Subsequently, $t_{\mathrm{meta}}$ is transformed by a learnable task meta-information encoder $f_{\mathrm{emb}}(\cdot)$ into the task embedding $t_{\mathrm{emb}}$, which can be expressed as:

$$t_{\mathrm{emb}} = f_{\mathrm{emb}}(t_{\mathrm{meta}}) \in \mathbb{R}^{d_{\mathrm{emb}}}, \tag{1}$$

where $f_{\mathrm{emb}} : \mathbb{R}^{d_{\mathrm{meta}}} \to \mathbb{R}^{d_{\mathrm{emb}}}$ and $d_{\mathrm{meta}}$ and $d_{\mathrm{emb}}$ is the dimension of task meta-information and the resultant task embedding, respectively. Further, the task embedding is processed through the task router

$f_{\mathrm{router}}(\cdot)$ (a feed-forward network) that outputs gating weights with dimensions matching the expert number $N_\mathbf{M}$. After a softmax operation, this tensor allocates attention intensity for all experts, effectively directing the focus of our model. Specifically, given $t_{\mathrm{meta}} \in \mathbb{R}^{d_{\mathrm{meta}}}$, it undergoes encoding via a task encoder $f_{\mathrm{Task}} : \mathbb{R}^{d_{\mathrm{meta}}} \to \mathbb{R}^{N_\mathbf{M}}$, leading to:

$$\alpha = \mathrm{Softmax}(f_{\mathrm{router}}(t_{\mathrm{meta}})), \tag{2}$$

where $\alpha \in \mathbb{R}^{N_\mathbf{M}}$ represents the gating weights for experts. Then, node features $X^l$, which are produced by the self-attention layer, are then fed into t-MoE, and are updated as:

$$X^{l+1} = \sum_{i=0}^{N_\mathbf{M}} \alpha_i \cdot \mathrm{FFN}_i(X^l), \tag{3}$$

where $\mathrm{FFN}_i(\cdot)$ denotes $i$-th experts. This innovative architecture, illustrated in Fig. 1c, enables the model to navigate a representation space defined by the convex hull of the outputs of $N_\mathbf{M}$ experts. Through this, the model can adaptively select the most relevant expert configurations for any given task during inference, effectively broadening the representation space to encompass an infinite combination of expert insights. Moreover, by optimizing the task embedding $t_{\mathrm{emb}}$, the model intrinsically captures task relevance, allowing for the seamless integration of the most appropriate expert knowledge. This task-adaptive learning mechanism significantly enhances task-specific performance by facilitating the nuanced selection and combination of expert knowledge based on the characteristics of each task.

### Building conformational relaxation model by iterative geometry update

This subsection focuses on a pivotal component of the OmniMol framework: the enhancement of generalizability via building a conformational relaxation model. The premise is that the physical principles for interatomic potential are localized and invariant characteristics across different molecules irrespective of their molecular scale or targeted property. Learning a heuristic interatomic potential for conformational relaxation serves as the cornerstone for our model's ability to generalize across molecules.

The learning process commences with the molecular SMILES representation, followed by conformation optimization using the Merck Molecular Force Field (MMFF). We denote the molecular system by $\mathcal{V} = \{v_i, i \in [1, |\mathcal{V}|]\}$, where each node $v_i$ symbolizes an atom, and $|\mathcal{V}|$ represents the total number of nodes. The $\mathcal{P} \in \mathbb{R}^{|\mathcal{V}| \times 3}$ corresponds to the spatial coordinates of all nodes, such that each row $\{\mathbf{p}_i \in \mathbb{R}^3, i \in [1, |\mathcal{V}|]\}$.

The goal of optimization is to ascertain a local minimum conformation, $\mathcal{P}^\star$, characterizing an equilibrium state of minimal potential energy with atoms experiencing negligible forces. A perturbation, $\tilde{\mathcal{P}}$, is applied to $\mathcal{P}^\star$ to simulate high-energy conformations, resulting in a non-equilibrium state $\mathcal{P}^0 = \mathcal{P}^\star + \tilde{\mathcal{P}}$.

Addressing the non-linear dynamics of force-displacement relationships, particularly for significant perturbations as depicted in Fig. 3a, b, our model integrates an intermediate positional update mechanism via model recycling. Let $S$ denote the count of Positional Update blocks used. The strategy of layering OmniMol components aims to iteratively refine the molecular conformation with $\Delta\mathcal{P}^x = \mathcal{P}^x - \mathcal{P}^{x-1}, x \in [1, S]$ through learning an interatomic potential $\mathbf{f}_{\mathrm{ForceField}}$. The predicted equilibrium state $\mathcal{P}^S$ is supervised using the L1 loss function:

$$\mathcal{L}_{\mathrm{ForceField}} = \mathrm{L}_1(\mathcal{P}^S - \mathcal{P}^{S-1}, \mathcal{P}^\star - \mathcal{P}^{S-1}) = \| \mathcal{P}^S - \mathcal{P}^{S-1} - (\mathcal{P}^\star - \mathcal{P}^{S-1}) \|_1, \tag{4}$$

where $\mathrm{L}_1(\cdot, \cdot)$ denotes the L1 loss function and $\| \cdot \|_1$ represents the L1 norm. This loss function facilitates the training of our model by

comparing the last geometry update $\mathcal{P}^S - \mathcal{P}^{S-1}$ against the equilibrium geometry, using the penultimate prediction $\mathcal{P}^{S-1}$ as a reference.

This methodology offers several benefits. It aligns with Hooke's Law for minor perturbations, indicating a linear force-displacement relationship, and diverges from the direct prediction of equilibrium from high-energy states. By dividing the geometry deviation into smaller increments and focusing on the final update, this approach minimizes the discrepancy between model supervision on geometry displacement and interatomic potential. The loss function penalizes cumulative positional inaccuracies throughout the iterative updates with $\mathbf{f}_{\mathrm{ForceField}}$, ensuring the refined accuracy of the learned potential across all iterations. This multi-step update process enables the model to accurately navigate the complex path to equilibrium, offering a nuanced approximation beyond simple one-step methods. Consequently, this approach not only simulates MMFF by introducing geometric perturbations but also refines the learning process through successive heuristic interatomic positional updates, thereby enhancing the generalization capabilities.

### Addressing molecule scale imbalance-induced bias

The challenge of building a reliable message-passing strategy for the molecule-to-property relation also lies in the considerable variation of atom counts across molecules. In our example of ADMET-P prediction, atomic counts range from around 10 to over 500 atoms. This variation introduces biases affecting both graph-level and node-level predictions, necessitating tailored strategies to mitigate the impacts of molecular scale imbalance-induced bias.

For graph-level predictions, conventional energy prediction models use a sum-based aggregation across nodes, reflecting the physical principle that the total potential energy of a molecule depends on its atomic count. However, in drug discovery, the emphasis often falls on specific functional groups within the molecule, with the remaining structure playing a secondary role. To address this, as illustrated in the right part of Supplementary Fig. 1, OmniMol employs a node-level softmax attention mechanism in the readout block to standardize graph-level outputs, enabling the model to focus on relevant nodes. This facilitates the automatic identification and prioritization of critical substructures and functional groups, as demonstrated by our experimental results.

Node-wise predictions towards equilibrium geometry are vital for the enhanced capabilities of our model. However, the variance in atom numbers across molecules can bias these predictions. This is a known challenge in graph neural networks, including some variants of Equivariant Graph Neural Networks (EGNNs)[74]. In particular, models using sum-based aggregation over neighbors might become skewed by atom count differences. Such models typically generate a scalar for each directed edge using edge-wise embeddings, which scales the interatomic directions before aggregation for node-wise geometry predictions. This methodology risks biasing predictions towards larger molecules, thereby challenging the generalizability of the learned interatomic potential across varying particle systems.

As decomposed in Fig. 7, OmniMol mitigates these challenges using the DR-Label strategy[17] for message aggregation from edges to nodes, which stabilizes E(3)-equivariant node-wise geometry predictions.

We define the following notations for our geometric update framework:

- $\mathbf{p}_i^{x-1} \in \mathbb{R}^3$ denotes the position vector (column vector) of atom $v_i$ at state $x-1$
- $\mathbf{d}_{ij}^{x-1} \in \mathbb{R}^3$ represents the unit direction vector from atom $v_j$ to $v_i$:

$$\mathbf{d}_{ij}^{x-1} = \frac{\mathbf{p}_i^{x-1} - \mathbf{p}_j^{x-1}}{||\mathbf{p}_i^{x-1} - \mathbf{p}_j^{x-1}||} \tag{5}$$

- $\Delta\mathbf{p}_i^x \in \mathbb{R}^3$ represents the geometry update vector for atom $v_i$ from state $x-1$ to $x$

During model supervision, we decompose the node-level supervision signal $\Delta\mathbf{p}_i^\star \in \mathbb{R}^3 = \mathbf{p}_i^\star - \mathbf{p}_i^{S-1}$ into edge-level magnitudes. For each edge $(i,j)$, we compute:

$$u_{ij} = (\mathbf{d}_{ij}^{S-1})^\top \Delta\mathbf{p}_i^\star \in \mathbb{R}, \tag{6}$$

where $(\mathbf{d}_{ij}^{S-1})^\top$ is a $1 \times 3$ row vector. This scalar projection $u_{ij}$ has two important properties:

- It is independent of the neighborhood size $|\mathcal{N}_i|$
- It is invariant to the graph construction algorithm[17]

During forward propagation, our model generates edge-wise scalar predictions $\{m_{ij}^{x-1}\}$. Each $m_{ij}^{x-1}$ represents the predicted projection magnitude of the geometry update along direction $\mathbf{d}_{ij}^{x-1}$. The corresponding projection vector for each edge is:

$$\mathbf{v}_{ij} = m_{ij}^{x-1}\mathbf{d}_{ij}^{x-1} \in \mathbb{R}^3. \tag{7}$$

To reconstruct the full geometry update $\Delta\mathbf{p}_i^x$ from these projections, we solve for the center point $\mathbf{c}_i$ of a sphere that passes through the origin and minimizes:

$$\min_{\mathbf{c}_i \in \mathbb{R}^3} \sum_{j \in \mathcal{N}_i} (||\mathbf{v}_{ij} - \mathbf{c}_i||^2 - ||\mathbf{c}_i||^2)^2 \tag{8}$$

This optimization problem has an analytical solution. Let us define:

$$\mathbf{b}_i = \frac{1}{|\mathcal{N}_i|}\sum_{j \in \mathcal{N}_i}(\mathbf{v}_{ij}^\top \mathbf{v}_{ij})\mathbf{v}_{ij} \in \mathbb{R}^3 \tag{9}$$

and

$$A_i = \frac{1}{|\mathcal{N}_i|}\sum_{j \in \mathcal{N}_i}\mathbf{v}_{ij}\mathbf{v}_{ij}^\top \in \mathbb{R}^{3 \times 3}, \tag{10}$$

where $A_i$ is a $3 \times 3$ symmetric positive definite matrix when the neighborhood contains at least three non-coplanar atoms (guaranteed by our graph construction). The matrix $A_i$ captures the spatial distribution of the projected vectors, while $\mathbf{b}_i$ represents their weighted average.

The optimal solution is then given by:

$$\mathbf{c}_i = (2A_i)^{-1}\mathbf{b}_i. \tag{11}$$

Finally, the geometry update vector is computed as:

$$\Delta\mathbf{p}_i^x = 2\mathbf{c}_i = 2(2A_i)^{-1}\mathbf{b}_i. \tag{12}$$

This solution ensures that the reconstructed geometry update best matches all edge-wise projections while maintaining physical consistency.

This methodology enables molecule-scale independent node-wise prediction and introduces additional supervision signals to decompose node-wise labels into edge-level information. Combined with the iterative positional update, OmniMol achieves physical explainability, addressing the challenges posed by varying molecular sizes. The method also fosters the learning of transferable knowledge across different molecular structures.

### Constructing SE(3) group encoder for chirality-aware molecular representation

Achieving chirality awareness in molecular representations is essential for accurate molecular property prediction, especially for ADMET-P and druggability. The message passing schema in DRFormer makes it

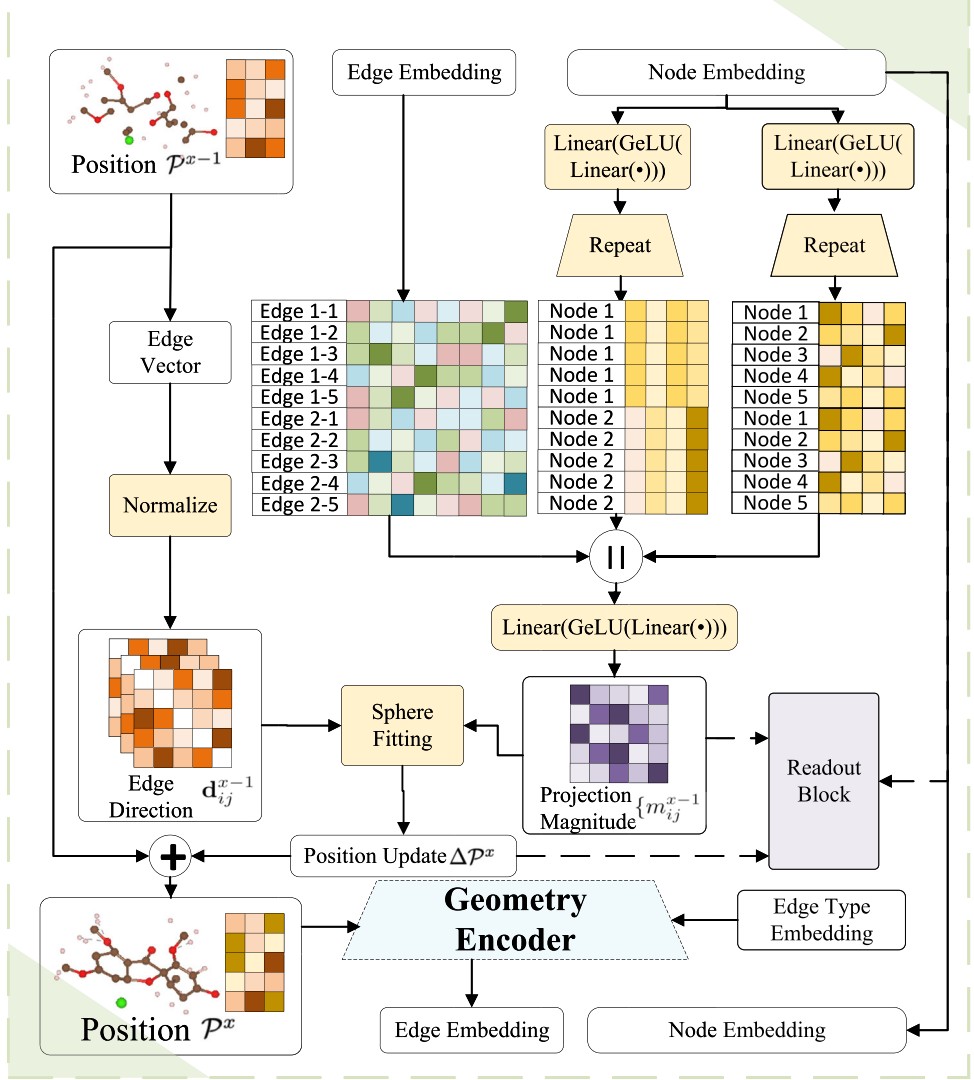

**Fig. 7 | Decomposed view of the positional update block of OmniMol.** It refines atomic coordinates by predicting the position update $\Delta\mathcal{P}^x$ from a previous state $\mathcal{P}^{x-1}$ to a new state $\mathcal{P}^x$. The update is guided by predicted projection magnitudes $m_{ij}^{x-1}$ along unit direction vectors $\mathbf{d}_{ij}^{x-1}$. Linear = Linear Projection, GeLU = Gaussian Error Linear Unit, the Geometry Encoder is detailed in Fig. 8.

E(3)-equivariant in node-wise prediction and E(3)-invariant in graph-level property prediction. To address this, we have developed an SE(3) group encoder that upholds the SE(3) group properties, facilitating efficient chirality awareness in our model, both in theory and through experimental validation.

Our methodology is showcased in Fig. 8. Consider a molecular structure comprising five atoms, represented as $\{v_i, v_j, v_a, v_b, v_c\}$, as illustrated in Fig. 8b. To ensure SE(3)-invariant properties, for each edge $e_{ij}$, we compute a torsion angle $\psi_{ij}$ that remains constant under rotation and translation but changes under reflection. We begin by calculating permutation invariant vectors $\mathbf{c}_{ij}$ and $\mathbf{c}_{ji}$, associated with the directions of edges $e_{ij}$ and $e_{ji}$:

$$\mathbf{c}_{ij} = \sum_{k \in \mathcal{N}_i \setminus j} \mathbf{d}_{ij} \times \mathbf{d}_{ik}, \tag{13}$$

$$\mathbf{c}_{ji} = \sum_{k \in \mathcal{N}_j \setminus i} \mathbf{d}_{ji} \times \mathbf{d}_{jk}. \tag{14}$$

Given $\mathbf{c}_{ij}$ and $\mathbf{c}_{ji}$ are orthogonal to $\mathbf{d}_{ij}$, $\mathbf{c}_{ij} \times \mathbf{c}_{ji}$ is parallel to $\mathbf{d}_{ij}$. We then compute the chirality-aware torsion angle $\psi_{ij}$ using the inner product

$< \mathbf{a}, \mathbf{b} >$ between vectors $\mathbf{a}$ and $\mathbf{b}$:

$$\psi_{ij} = \operatorname{asin}\left(\left\langle \frac{\mathbf{c}_{ij} \times \mathbf{c}_{ji}}{|\mathbf{c}_{ij}| \cdot |\mathbf{c}_{ji}|}, \mathbf{d}_{ij} \right\rangle\right). \tag{15}$$

Importantly, $\psi_{ij}$ is sensitive to reflection operations $\mathbf{f}_{ref}$ towards the molecular system, such that:

$$\left\langle \frac{\mathbf{c}_{ij} \times \mathbf{c}_{ji}}{|\mathbf{c}_{ij}| \cdot |\mathbf{c}_{ji}|}, \mathbf{d}_{ij} \right\rangle = -\left\langle \frac{\tilde{\mathbf{c}}_{ij} \times \tilde{\mathbf{c}}_{ji}}{|\tilde{\mathbf{c}}_{ij}| \cdot |\tilde{\mathbf{c}}_{ji}|}, \tilde{\mathbf{d}}_{ij} \right\rangle, \tag{16}$$

where $\tilde{\mathbf{c}}_{ij}, \tilde{\mathbf{c}}_{ji}, \tilde{\mathbf{d}}_{ij}$ are the vectors corresponding to $\mathbf{c}_{ij}, \mathbf{c}_{ji}, \mathbf{d}_{ij}$ in the reflected molecular system.

The torsion angle $\psi_{ij}$ is then encoded using circular harmonics for edge embedding, defined for a circular harmonic of degree $l$ and order $m$ as $\mathbf{CH}_l^{(\omega)} : S \to \mathbb{R}$:

$$\mathbf{CH}_l^{(\omega)}(\psi) = \begin{cases} \sin(\omega\psi) & \text{if } l = 1, \\ \cos(\omega\psi) & \text{if } l = -1. \end{cases} \tag{17}$$

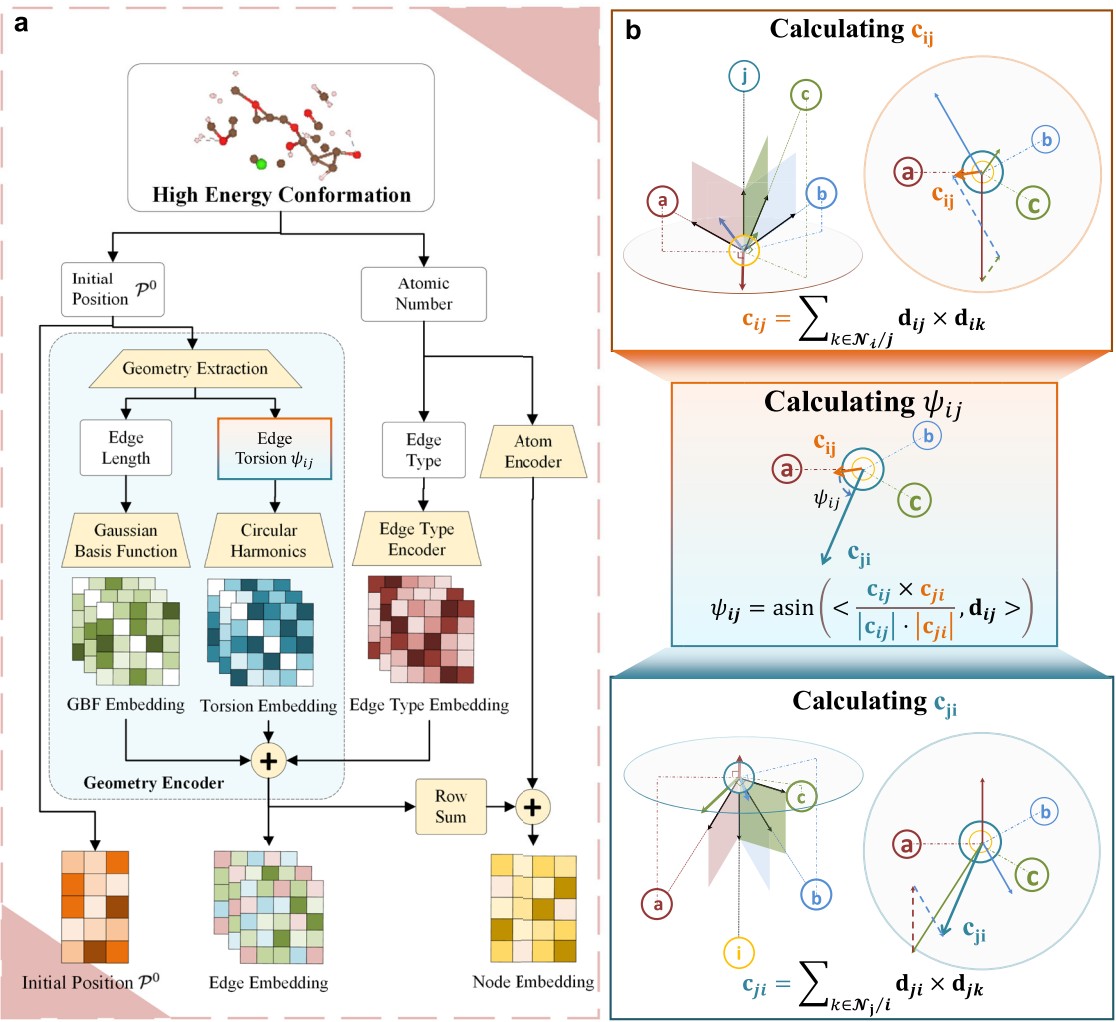

**Fig. 8 | Decomposed view of the SE(3) encoder component of OmniMol. a** The framework of the SE(3) encoder. It illustrates how atomic positions $\mathcal{P}$ are utilized to derive edge length and torsion embeddings. **b** The chirality-aware torsion calculation. A torsion angle $\psi_{ij}$ is computed for each edge using intermediate vectors ($\mathbf{c}_{ij}$, $\mathbf{c}_{ji}$) that are derived from the unit direction vectors of adjacent edges (e.g., $\mathbf{d}_{ij}$, $\mathbf{d}_{ik}$).

Where $l \in \{-1, 1\}$ and $\omega \in \{-\Omega, -\Omega + 1, \ldots, \Omega - 1, \Omega\}$, $\Omega$ being a fixed maximal order. This encoding is chirality aware because the $l = 1$ part encoding will be negated when the molecule is reflected, therefore enriching the molecular representation with detailed spatial information crucial for accurate property prediction. A detailed proof of Eq. (16) and (17) is available in the supplementary material.

### Integration of diverse property prediction tasks

In complement to the task meta-information encoder and t-MoE, dataset diversity is another major barrier for a mixed training. Here we introduce our efforts on three critical areas in confronting the challenge brought by dataset diversity.

Dataset size imbalances: The disparity in dataset sizes, sometimes spanning orders of magnitude, necessitates a balanced training approach. We employ a task-balancing dataloader that adjusts sample frequencies to address these imbalances. The average dataset size is computed as $\bar{S} = \frac{\sum_{i=1}^{T} S_i}{T}$, where $T$ is the number of tasks and $S_i$ the size of dataset $i$. Each task $i$ receives an adjusted batch size $\tilde{S} = (1 - \theta)S_i + \theta\bar{S}$, moderated by a balancing factor $\theta \in [0, 1]$.

Scope of each dataset: Addressing differences between classification and regression tasks includes managing label imbalance in classification and varying statistical properties in regression. For regression tasks, target values are normalized using z-score

normalization:

$$\tilde{y}_{\text{reg}} = \frac{y_{\text{reg}} - \mu_{\text{reg}}}{\sigma_{\text{reg}}}, \tag{18}$$

where $\tilde{y}_{\text{reg}}$ is the normalized target, $y_{\text{reg}}$ the original target, $\mu_{\text{reg}}$ the mean, and $\sigma_{\text{reg}}$ the standard deviation of the regression targets. The L1 loss function is used to calculate the regression loss, $\mathcal{L}_{\text{reg}}$:

$$\mathcal{L}_{\text{reg}}(\tilde{y}_{\text{reg}}, p_{\text{reg}}) = \| \tilde{y}_{\text{reg}} - p_{\text{reg}} \|_1. \tag{19}$$

For classification tasks, a reweighting scheme is utilized to address label imbalance. The loss function for a binary classification task, with $y_{\text{cls}} \in \{0, 1\}$ as the true label and $p_{\text{cls}} \in [0, 1]$ as the predicted probability for the positive class, is weighted based on the positive label ratio $t_+$:

$$\mathcal{L}_{\text{cls}}(y_{\text{cls}}, p_{\text{cls}}) = -\left( \frac{y_{\text{cls}}}{2t_+} \cdot \log(p_{\text{cls}}) + \frac{1 - y_{\text{cls}}}{2(1 - t_+)} \cdot \log(1 - p_{\text{cls}}) \right). \tag{20}$$

Simultaneous supervision of regression and classification tasks: The model differentiates graph-level embeddings for each task type, combining their losses into a total loss $\mathcal{L}_{\text{total}}$, balanced by a

coefficient $\gamma$:

$$\mathcal{L}_{\text{total}} = \mathcal{L}_{\text{reg}} + \gamma \cdot \mathcal{L}_{\text{cls}}. \qquad (21)$$

This formula ensures that $\mathcal{L}_{\text{total}}$, representing the overall loss, effectively integrates both regression ($\mathcal{L}_{\text{reg}}$) and classification ($\mathcal{L}_{\text{cls}}$) losses, with $\gamma$ adjusting their relative importance.

This strategic approach ensures balanced and effective training, adeptly navigating the complexities inherent in property prediction tasks, thereby enhancing model performance across a spectrum of datasets.

## Data availability

The HelixADMET dataset used in this study is publicly accessible at the PaddleHelix repository[13](https://paddlehelix.bd.bcebos.com/HelixADMET_open_data/HelixADMET_open_data.tgz). The chiral cliff dataset is available within the supplementary information of the referenced publication by[49] (https://doi.org/10.1002/cmdc.201700798). The source data generated in this study for figures and tables are provided with this paper. A portion of the ADMETlab 2.0 dataset used in this study is subject to access restrictions due to its inclusion of proprietary data from collaborating companies, which are protected by non-disclosure agreements and commercial restrictions. Researchers may request further details regarding the ADMETlab 2.0 database and its application in this study by contacting Professor Dongsheng Cao at oriental-cds@163.com. To ensure the reproducibility of our methodology despite this restriction, we have provided a representative subset (the training and testing subsets of LogS dataset from ADMETLab 2.0) and a complete workflow to train and validate a "twin model". This dataset is available as part of the Source Data. A detailed description of the procedure and the corresponding results are provided in Supplementary (see Supplementary 22, "Reproducibility Example for LogS Prediction"). The Optical Rotation Data used in this study are available under restricted access. The reason for this restriction is that the data are part of an ongoing research project, and a premature public release could compromise the novelty of future publications. Access can be obtained by contacting the corresponding author, Guangyong Chen, at gychen@link.cuhk.edu.hk. Requests should include a brief description of the intended non-commercial research use. We will endeavor to respond to access requests within 2-4 weeks. Once granted, access to the data will be provided for the duration of the proposed research project under a data use agreement. Source data are provided with this paper.

## Code availability

The code, pretrained model weights, and the Python environment used in this study are publicly available in the OmniMol repository on GitHub at https://github.com/bowenwang77/OmniMol. The specific version of the code used in this paper has been deposited in Zenodo and is cited as ref. 39.

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

## Acknowledgements

This work was funded by the National Natural Science Foundation of China (Grant Nos. 62422605, 92370132 to J.H.), the "Pioneer" and "Leading Goose" R&D Program of Zhejiang (2024SSYS0007 to J..L..), the Zhejiang Province Vanguard Goose-Leading Initiative (No. 2025C01114 to G.C.) and National Natural Science Foundation of China (Grant Nos. 62376254, 32341017, 32341018 to G.C.), the Hong Kong Innovation and Technology Fund (Project Reference Number: ITS/241/21 to P.A.H.), and the National Natural Science Foundation of China (Grant Nos. 22377111 to E.W.).

## Author contributions

B.W. conceived and developed the OmniMol architecture and designed the ADMET and chirality-awareness prediction experiments. JY.L. designed and conducted the model explainability experiments and SAR studies. B.W. and JY.L. jointly wrote the initial manuscript draft. D.Z. contributed to the development of the t-MoE architecture and figure preparation. L.L. formulated the hypergraph-based approach for addressing the imperfect annotation problem. JP.L. designed the data balancing strategy. E.W., J.Q., and J.H. contributed to the theoretical foundation and interpretability of the model. G.C., L.S., and C.L. prepared datasets for rotatory strength prediction. T.H. and D.C. jointly prepared the ADMET property prediction datasets. J.H., T.H., D.C., and G.C. coordinated computational resources. G.C. and P.A.H. supervised the project throughout conceptualization, implementation, experimentation, and manuscript preparation. All authors contributed to manuscript revision and approved the final version.

## Competing interests

The authors declare no competing interests.
