## [Transparent Peer Review file · Nature Communications]

Unified and Explainable Molecular Representation Learning for Imperfectly Annotated Data from the Hypergraph View

Corresponding Author: Professor Guangyong Chen

Version 0:

Reviewer comments:

Reviewer #1

(Remarks to the Author)

The authors presented a manuscript describing a novel framework to perform multi-task learning/predictions of compounds and related properties. In addition, the method estimates a low-energy conformer without physics-based calculations (e.g. molecular mechanics) and correctly classifies the enantiomeric enumeration of optical isomers. Lastly, the authors compare trained models with the state-of-the-art models from ADMETLab 2.0. The manuscript is well-written and I have some suggestions to improve clarity and the overall scientific quality of the work. I suggest that the paper could be accepted after major revision. Following, the authors will find point-by-point comments:

- 1) Lines 84-85: Since "imperfectly annotated dataset" is the core of the work, the authors should provide a formal definition and give examples to illustrate it. It is hard to understand properly the context and the importance of this concept for non-modellers scientists.
- 2) Figure 1D: What do the arrows' thickness differences mean?
- 3) Lines 268-269: Please, define OC20, PCQM4MV2, and QM9 (write the full names, give more context, etc) and provide links for the datasets.
- 4) Lines 283-285: The authors stated that they adopted the same protocol from ADMET-Lab 2.0 to validate their models and that's completely fine to keep comparability between platforms. However, the individual datasets should be characterized according to data balance and other metrics as MCC should be included to avoid biased evaluation of quality metrics.
- 5) Lines 285-286: In this same sense, regression models should be evaluated according to metrics present in Gramatica and Sangion, 2016 (<https://doi.org/10.1021/acs.jcim.6b00088>) work such as r2m, q2F1, q2F2, q2F3, CCC among others.
- 6) Table 1: Provide the meaning for each acronym (e.g. HIA, EC, EI, HILL, etc).
- 7) Lines 311-312: Please, state the molecular target of Remdesivir instead of the disease.
- 8) Lines 532-537: In the following sentence "Additionally, the node-level attention also prevents the model from being biased towards the main part of the molecule or being influenced by the number of nodes (molecule weight) of the molecule, an issue that might occur with conventional sum-aggregated graph-level readout.", what did the authors mean with "the main part of the molecule"? Please provide examples to illustrate this problem.
- 9) Table 2: Please, provide a direct comparison of presented results with maps of atomic importance from other works. For instance, you can use other hERG inhibition software/servers and provide maps for the same molecules in terms of comparison. It would be interesting if you provide a comparison with as much software as possible and for the most number of endpoints as possible. Lastly, it would be important to compare with SAR studies. In this sense, you can introduce a layer of "validation" by comparing physicochemical interpretation for predictions.
- 10) Lines 644-668: The discussion is too shallow and must be drastically improved. For example, the requested comparison related to Table 2 explicability is useful. In the same sense, comparison with other models for the same endpoint could be

interesting as well (please, describe the differences in dataset size, diversity, predictability metrics, etc).

Additional requests:

11) Authors must characterize the datasets with cheminformatics tools (calculation of fingerprints, similarity, physicochemical descriptors, etc).

12) The applicability domain of reported models should be reported and discussed.

13) X/Y-Randomization tests should be carried out to prove that models were not obtained by chance.

14) I strongly suggest to the authors that the datasets and other raw data (predictions, final conformers of compounds, etc) that they judge useful for the readership should be available for download.

(Remarks on code availability)

A password protects the link and I would not be able to see it.

Reviewer #2

(Remarks to the Author)

The submission on OmniMol contains a number of interesting ML ideas. However, I do not believe this submission is appropriate for Nature Communication for both scope and technical reasons.

Major Points:

1. The work is a pure ML retrospective study almost entirely on one dataset collection where improvements on existing methods is small. This scope is more appropriate to an ML or cheminformatics journal until more convincing prospective work can be done.

2. The comparisons to existing methods are weak.

a. I find it misleading that only one baseline was included in the main text where in fact other baselines seem to perform better. (see point 2b as well)

b. Lack of any statistical rigor in the comparison of methods across multiple datasets. I recommend you look at methods such as [1] (later modified by [2]) or Pat Walter's writing [3, 4].

Notably, I suspect HelixADMET and OmniMol would come out as equivalent in a more statistically rigorous comparison as they have equal number of datasets where they outperform each other. I suspect your mean AUC difference is driven entirely by Rodent Acute Toxicity which is a notable outlier in the entire set of results.

3. The geometry update process is much discussed and celebrated in the paper (such as line 794 "serves as the cornerstone"), but there is no evidence that it is useful.

a. Many methods already use a 3D conformers (or multiple conformers) as part of an ML model. What is novel here is the AlphaFold-like iterative geometry updating but from a corrupted true geometry. As far as I can tell from Section A.2, only the entire geometry module was removed, not specifically this unusual corrupted geometry part. The natural and simpler method is to just use the SE(3) encoder on the derived geometry (see technical point below on the source of the conformation).

b. The loss function is unusual with only supervision on the last step and is in contrast to e.g. diffusion models. I would not consider it essential to explore this for a first publication, but I suggest you explore alternatives.

4. The task embedding work is interesting. There is a long history of multi-task models for cheminformatics e.g. [5, 6, 7] but I don't know of ones that include dataset statistics as an input description of the task. The paper examines the embedding, but does not provide any evidence that anything is learned. In order to make the claim that something is learned, you need to compare the learned to the original features space. You could perhaps do this comparison with dimensionality reduction of the original feature space compared to the dimensionality of the learned embedding and demonstrate in some way that the learned embedding is better.

5. Section A.5: Judging by the style of writing and phrases such as "as demonstrated in your example" (where no example is in the text), I believe an LLM was used to generate this proof. As the proof is a scientific task and not just copy-editing, my understanding is that such a use needs to be acknowledged. Further, the proof you need is some statement about your final encoding values CH_{ell}^m and how they change with respect to reflections not simply the trivial cross product relationship covered in A.5

Smaller Technical Points

6. I am unclear if you used the exact same data splits as used in AdmetLab2.0. If they are available, you should. I note that they report "validation and test sets by a ratio of 8:1:1, and stratified sampling was used when partitioning the data for classification to keep the ratio of the positive and negative

instances in the three subsets balanced.” [8] and I see no mention of stratification in your paper. If they did stratification and you did not, it weakens the comparisons.

7. Line 492. It's typical to do an alignment operation to find the minimum MAE or RMSE. If you didn't do that here, you probably should.

8. Line 796. The generation of appropriate conformers is actually quite a tricky subject. See the documentation of Omega for example [9]. A single conformer almost certainly does not explain all properties of a molecule. At a minimum, the procedure you used to estimate the lowest energy conformer needs to be described. More broadly for future work you may want to consider conformers more holistically.

9. Line 854. You are missing a citation to what EGNN you are referring to. There are many variants of equivariant networks out there.

10. I struggled to work through the math at the end of this section and it should be made more clear. You are using non-standard notation with i, j not referring to rows and cols of matrices, making it difficult to understand which vectors are row vector and which are column vectors. Also, whether 'diag' produces a vector or a matrix is unclear. Some intuition for the purpose of the inverse of A_i could also help.

11. Line 967 refers to w_+ and w_- that are neither defined nor used.

12. Citation 1 is good, but there are more recent studies you may be interested in, such as [10,11]

[1] Demšar. Statistical comparisons of classifiers over multiple data sets. *J. Mach. Learn. Res.* 7, 1–30 (2006)

[2] Benavoli, Corani & Mangili. Should we really use post-hoc tests based on mean-ranks? *J. Mach. Learn. Res.* 17, 1–10 (2016)

[3] <https://practicalcheminformatics.blogspot.com/2019/02/some-thoughts-on-evaluating-predictive.html>

[4] <https://practicalcheminformatics.blogspot.com/2020/05/some-thoughts-on-comparing.html>

[5] Martin, E. J., Polyakov, V. R., Tian, L. & Perez, R. C. Profile-QSAR 2.0: Kinase Virtual Screening Accuracy Comparable to Four-Concentration IC50s for Realistically Novel Compounds. *J. Chem. Inf. Model.* 57, 2077–2088 (2017)

[6] Ramsundar, B. et al. Massively Multitask Networks for Drug Discovery. *arXiv [stat.ML]* (2015)

[7] Irwin, B. W. J.; Whitehead, T. M.; Rowland, S.; Mahmoud, S. Y.; Conduit, G. J.; Segall, M. D. Deep Imputation on Large-scale Drug Discovery Data. *Applied AI Letters* 2021, 2 (3). <https://doi.org/10.1002/ail2.31>.

[8] Xiong, G. et al. ADMETlab 2.0: an integrated online platform for accurate and comprehensive predictions of ADMET properties. *Nucleic Acids Res.* 49, W5–W14 (2021)

[9] <https://www.eyesopen.com/omega>

[10] Schlender, M., Hernandez-Villafuerte, K., Cheng, C.-Y., Mestre-Ferrandiz, J. & Baumann, M. How Much Does It Cost to Research and Develop a New Drug? A Systematic Review and Assessment. *Pharmacoeconomics* (2021)

doi:10.1007/s40273-021-01065-y

[11] DiMasi, J. A., Grabowski, H. G. & Hansen, R. W. Innovation in the pharmaceutical industry: New estimates of R&D costs. *J. Health Econ.* 47, 20–33 (2016)

(Remarks on code availability)

Given the other issues, I did not review code. If this comes back for a revision, I would review code at that time.

Reviewer #3

(Remarks to the Author)

Unified and Explainable Molecular Representation Learning for Imperfectly Annotated Data from the Hypergraph View

General:

Method in principle very interesting and relevant!

However the paper incorporates multiple ideas within a single method, but the evaluation of these ideas lacks depth. The evaluation should be more comprehensive and clearly demonstrate which methodological ideas or modules impact prediction performance. This would greatly enhance the value of the study. Currently, the results are not sufficiently differentiated. The introduction could be more comprehensive, and the current state-of-the-art approaches should be mentioned and described. The authors should explicitly indicate what is new in their work and how they have built upon existing methods. A clear comparison to other methods is missing, and the results should systematically demonstrate the advantages of their method compared to alternative approaches. The effects and influence factors should be analyzed separately.

- Additionally, the reported results should include standard deviations.

- It is customary to compare a new method to existing methods, such as Chemprop, Random Forest, KNN, etc. However, in this study, there seems to be only a comparison to a single method, which is also not directly named.

- It is also common practice to compare against baselines using different challenging splits, such as random splits, scaffold splits, or other cluster splits. In this study, it appears that only one split was used.

- The discussion section is very brief and does not adequately highlight the limitations of the approach.

P2 173-79 – more literature information should be included

- Very optimistic description although relevant endpoints are still very challenging

P2 | 93 - please differentiate more between the general problem and a concrete use case

-very short description of a complex relationships

P4 | 168 -“practical clinical data analysis” -> which clinical data and what kind of analysis?

P4 | 173 -“We” typo

P4 | 209-220 - This paragraph should talk about the concrete results that came out of this study, for example, the improvement in prediction performance compared to strong baseline methods. Instead, this paragraph talks about the hypothetical benefits of the method's ideas, making it unclear if the method has an actual benefit.

P4 | 214 -“OmniMol lays the groundwork for applying generalized MRL across diverse datasets, thereby capturing fundamental physical mechanisms.” -> very strong statement which needs a very good validation

P4 | 219 - “broader range of AI driven techniques adhering to the laws of nature” This is an unusual and not very scientific statement

P7 | Table 1 - It is not clear to which single method (or multiple methods?) OmniMol is compared in Table 1. There is just a single number for each endpoint. It should be made clear to the reader which method OmniMol is compared against. There is also a range of baseline methods in the molecular property prediction domains, e.g. Random Forest with Morgan binary Fingerprints with radius 2 and 2048 bits, that are often strong baselines and should be considered in every study presenting a new method.

P8 | 394 -Is this a significant difference? include standard deviations and significance test

Fig 4 -very important for understanding

(Remarks on code availability)

Version 1:

Reviewer comments:

Reviewer #1

(Remarks to the Author)

I want to thank the authors for their hard efforts. They addressed properly all my suggestions except for "comment 12". In this suggestion (The applicability domain of reported models should be reported and discussed), I asked the authors to provide some method to evaluate the range of properties and activities of the test set in comparison with the training set to assess if or not the predictions were reliable in terms of "the model indeed learned from presented data" or "if the prediction is just a guess". In this sense, it would be valuable to implement this tool so that users can evaluate the trustability of predictions for unseen data.

There are some papers that could guide on this task:

Gadaleta, D., Mangiatordi, G. F., Catto, M., Carotti, A., & Nicolotti, O. (2016). Applicability domain for QSAR models: where theory meets reality. *International journal of quantitative structure-property relationships (IJQSPR)*, 1(1), 45-63.

Sahigara, F., Mansouri, K., Ballabio, D., Mauri, A., Consonni, V., & Todeschini, R. (2012). Comparison of different approaches to define the applicability domain of QSAR models. *Molecules*, 17(5), 4791-4810.

Roy, K., Kar, S., & Ambure, P. (2015). On a simple approach for determining applicability domain of QSAR models. *Chemometrics and Intelligent Laboratory Systems*, 145, 22-29.

Dragos, H., Gilles, M., & Alexandre, V. (2009). Predicting the predictability: a unified approach to the applicability domain problem of QSAR models. *Journal of chemical information and modeling*, 49(7), 1762-1776.

Jaworska, J., Nikolova-Jeliazkova, N., & Aldenberg, T. (2005). QSAR applicability domain estimation by projection of the training set in descriptor space: a review. *Alternatives to laboratory animals*, 33(5), 445-459.

(Remarks on code availability)

Reviewer #2

(Remarks to the Author)

The revisions have not changed the overall critiques of the paper, despite the large number of words and figures that were thrown at the critiques.

The major issues are:

Comment 1:

Nothing in your response is about prospective application. I am not disagreeing that your attention analysis is suggestive,

but it is all retrospective. I especially find it laughable that you title response point 4 "Prospective Value" with things that you have not done prospectively or demonstrated any value.

Comment 2:
Statistical rigor

Burying one statistical analysis for one dataset deep in supplemental, which shows a p-value "approaches but does not reach the conventional significance threshold of 0.05," along with lots of metrics and discussion is classic p-hacking behavior. The comparison and the language around the method comparison remains quite suspect and revisions decidedly do not address the point raised.

Comment 3

The response attempts to justify why the complex geometry step is a reasonable thing to do. This complex thing you have done might be a good idea compared to the simpler one. But nothing has changed in that you have not provided evidence that this is true.

(Remarks on code availability)

Reviewer #3

(Remarks to the Author)
- comments are sufficiently addressed

(Remarks on code availability)

Version 2:

Reviewer comments:

Reviewer #1

(Remarks to the Author)
The manuscript could be accepted after the proposed revision.

(Remarks on code availability)

Reviewer #3

(Remarks to the Author)

The concept of the applicability domain (AD) is crucial in the field of quantitative structure-activity relationships (QSAR) and predictive modeling, particularly in the context of chemical and biological data. Netzeva et al. (2005) contributed significantly to this topic by defining applicability domain as the chemical space within which a predictive model can make reliable predictions. It essentially delineates the boundaries of the dataset that the model was trained on and indicates where the model's predictions can be considered valid.

The applicability domain ensures that when a new compound is assessed using the model, it falls within the characteristics of the training set used to develop that model. If a compound lies outside this domain, the prediction could be less reliable or even invalid. This aspect is especially important in regulatory contexts, where predictions made by models are often used to make decisions about safety and efficacy.

Netzeva et al. proposed a framework for assessing the applicability domain based on several key approaches:

Structural Similarity: This approach involves comparing the chemical structure of the new compound to those in the training set. Metrics such as Tanimoto similarity coefficients can be used to quantify how similar the new compound is to the compounds in the dataset.

Descriptor Space: This involves evaluating the space defined by the molecular descriptors (quantitative representations of chemical structure). The model's predictive capability is typically confined to a certain range of descriptor values. If a new compound lies outside this range, it is deemed outside the applicability domain.

Statistical Measures: Some methods employ statistical techniques to define the boundaries of the applicability domain. For instance, the use of confidence intervals around predictions or built-in confidence measures can help identify when a prediction is reliable. There are also frameworks like conformal prediction framework.

An applicability domain measure is considered efficient when it effectively identifies the boundaries within which a predictive model can reliably make predictions.

You decided for the descriptor space approach. I am wondering because your model is not trained on these descriptors you

only used them for the definition of the AD if I understood this correctly. If these are not the features your model is trained on how do you come to the conclusion that their distribution is a good approximation for the limits of your method? Or asking the other way around can you eliminate unreliable predictions  improve accuracy of the model when excluding compounds that are out of domain. Does your method makes more misclassifications due to outliers or close to the decision boundary (similar compounds but conflicting labels).

(Remarks on code availability)

Version 3:

Reviewer comments:

Reviewer #3

(Remarks to the Author)

Thank you very much for the adaptations.

I have no further comments and the manuscript can be accepted.

(Remarks on code availability)

README available and code installable.

Appendix B	Response to comments	3181
		3182
	We sincerely thank all reviewers for their thorough and constructive feedback, which has helped us significantly improve our manuscript. Below, we provide detailed point-by-point responses to each reviewer’s comments. We have substantially expanded our supplementary materials to include comprehensive analyses and additional results. We encourage reviewers to consult the supplementary section, which contains important supporting evidence and detailed explanations referenced in our responses.	3183 3184 3185 3186 3187 3188
		3189
1	Response to Reviewer #1:	3190
		3191
	1.1 Comment 1:	3192
	Since "imperfectly annotated dataset" is the core of the work, the authors should provide a formal definition and give examples to illustrate it. It is hard to understand properly the context and the importance of this concept for non-modellers scientists	3193 3194 3195
	Our response: We appreciate the reviewer’s insightful comment regarding the importance of clearly defining the concept of an "imperfectly annotated dataset." In response, we have enhanced the introduction of our work to include a more formal and mathematical definition of this problem. The revised definition is as follows:	3196 3197 3198 3199
	Let $\mathcal{M} = \{m_1, m_2, \dots, m_{ \mathcal{M} }\}$ and $\mathcal{E} = \{e_1, e_2, \dots, e_{ \mathcal{E} }\}$ be the set of all molecules and all properties of interest, correspondingly, with $ \mathcal{M} $ and $ \mathcal{E} $ denotes the total number of molecules and properties. Each property $e_i \in \mathcal{E}$ is associated with a subset of molecules $\mathcal{M}_{e_i} \subseteq \mathcal{M}$, indicating that only a portion of the molecules are labeled with property e_i . Formally, $\mathcal{M}_{e_i} = \{m_j \in \mathcal{M} : m_j \text{ is labeled with property } e_i\}, \forall e_i \in \mathcal{E}$. An imperfectly annotated dataset is characterized by $\exists e_i \in \mathcal{E}$ such that $\mathcal{M}_{e_i} \subsetneq \mathcal{M}$, i.e., there exists at least one property that is not labeled for all molecules.	3200 3201 3202 3203 3204 3205 3206
	This formal definition provides a rigorous mathematical framework for understanding the concept of an imperfectly annotated dataset.	3207 3208
		3209
	1.2 Comment 2:	3210
	Figure 1D: What do the arrows’ thickness differences mean?	3211
	Our response: The thickness of the arrows in Fig. 1D is proportional to the number of molecules associated with each property in the corresponding dataset. Arrows with greater thickness represent properties that have a larger number of labeled molecules, while thinner arrows indicate properties with fewer labeled molecules. This visual representation highlights the imbalanced nature of the imperfectly annotated dataset, where different properties may have significantly varying amounts of annotated molecular data.	3212 3213 3214 3215 3216 3217 3218 3219
		3220
	1.3 Comment 3:	3221
	Please, define OC20, PCQM4MV2, and QM9 (write the full names, give more context, etc) and provide links for the datasets.	3222 3223
	Our response: We have provided formal definitions and corresponding links for the datasets in our results section:	3224 3225
	Notable examples include the Open Catalyst 2020 (OC20) dataset [1], which contains over 264 million catalysis systems with fully annotated geometry, system energy and atomic force; the PubChemQC project dataset (PCQM4MV2) [4], consisting of 3.4 million molecules with DFT-calculated HOMO-LUMO gaps; and the QM9 dataset [5], which includes 134k stable small organic molecules with fully labeled geometries and properties.	3226 3227 3228 3229 3230 3231 3232 3233

3234 **1.4 Comment 4:**

3235 The authors stated that they adopted the same protocol from ADMET-Lab 2.0 to
3236 validate their models and that's completely fine to keep comparability between plat-
3237 forms. However, the individual datasets should be characterized according to data
3238 balance and other metrics as MCC should be included to avoid biased evaluation of
3239 quality metrics.

3240 **Our response:** In response, we have implemented the following improvements:

3241 **Comprehensive Metric Reporting:** We have included Matthews Correlation
3242 Coefficient (MCC), specificity, and sensitivity for all classification tasks. These com-
3243 prehensive metrics are now available in the supplementary material under "Full
3244 Evaluation Metrics." **Enhanced Comparative Analysis:** Table 1 in the main text
3245 has been updated to include MCC results, allowing for a more robust comparison with
3246 ADMETLab 2.0. It's important to note that the MCC results in Table 1 represent
3247 the best model performance after fine-tuning from the pretrained model using mixed
3248 datasets. In contrast, the MCC results in the supplementary table reflect the perfor-
3249 mance of the pretrained model without fine-tuning. **Data Balance Assessment:** We
3250 acknowledge the importance of characterizing datasets according to data balance. We
3251 have included the detailed dataset statistics in our supplementary material.

3253 **1.5 Comment 5:**

3254 In this same sense, regression models should be evaluated according to metrics present
3255 in Gramatica and Sangion, 2016 (<https://doi.org/10.1021/acs.jcim.6b00088>) work
3256 such as r^2_m , q^2_{F1} , q^2_{F2} , q^2_{F3} , CCC among others.

3258 **Our response:**

3259 We have made the following additions:

3260 **Extended Regression Metrics:** As recommended, we have incorporated addi-
3261 tional evaluation metrics for our regression models, including r^2_m , q^2_{F1} , q^2_{F2} , q^2_{F3} , and
3262 Concordance Correlation Coefficient (CCC), as suggested in Gramatica and Sangion
3263 (2016). These metrics are now included in our supplementary material, providing a
3264 more thorough evaluation of our regression models. **Main Results Update:** We have
3265 added the Root Mean Square Error (RMSE) metric for our method in Table 1 of the
3266 main text, enhancing the comparability of our results. **Comprehensive Evaluation**
3267 **in Supplementary Material:** A new table in the supplementary material now pro-
3268 vides a full evaluation of our mixed pretrained model using all the requested metrics.
3269 This comprehensive evaluation allows for a more in-depth assessment of our model's
3270 performance across various statistical measures.

3271

3272 **1.6 Comment 6:**

3273 Table 1: Provide the meaning for each acronym (e.g. HIA, EC, EI, HILL, etc).

3274 **Our response:** We have summarized all 52 endpoints in Supplementary Table 6
3275 of the supplementary material, providing the full names, meanings, and explanations
3276 of the output results for each endpoint.

3277

3278 **1.7 Comment 7:**

3280 Please, state the molecular target of Remdesivir instead of the disease.

3281 **Our response:** We employ the expression "only one enantiomeric form of Remde-
3282 sivir can inhibit the RNA-dependent RNA polymerase (RdRp) of coronaviruses,
3283 which further shows potential efficacy against SARS-CoV-2", replacing the previous
3284 statment.

3285

3286

1.8 Comment 8: 3287

In the following sentence "Additionally, the node-level attention also prevents the 3288
model from being biased towards the main part of the molecule or being influenced by 3289
the number of nodes (molecule weight) of the molecule, an issue that might occur with 3290
conventional sum-aggregated graph-level readout.", what did the authors mean with 3291
"the main part of the molecule"? Please provide examples to illustrate this problem. 3292

Our response: By describing "the main part of the molecule," we refer to the main 3293
fragments of a molecule, excluding the functional groups which are crucial to specific 3294
endpoint. We want to express that OmniMol can accurately identify key functional 3295
groups for specific tasks, regardless of the size and structure of the molecule. In other 3296
words, OmniMol focuses attention on these important functional groups rather than 3297
only assigning excessive attention values to other main fragments of one molecule. 3298

For example, In Fig. 4, OmniMol can accurately assign strong attention value to the 3299
tertiary amine group in both compounds **1a** and **1b**, despite the significant differences 3300
in molecular size and structure between **1a** and **1b**. In fact, tertiary amines (especially 3301
heterocyclic tertiary amines) are key substructures that contribute to hERG inhibition. 3302

To avoid ambiguity, we employ the expression "The node-level attention archi- 3303
tecture enables OmniMol to focus on critical functional groups essential for specific 3304
properties while avoiding excessive attention to other molecular fragments, address- 3305
ing the limitations of conventional sum-aggregated graph-level representations that 3306
may struggle with varying molecular sizes and complexities", replacing the previous 3307
statement. 3308

1.9 Comment 9: 3309

Please, provide a direct comparison of presented results with maps of atomic 3310
importance from other works. For instance, you can use other hERG inhibition soft- 3311
ware/servers and provide maps for the same molecules in terms of comparison. It 3312
would be interesting if you provide a comparison with as much software as possible 3313
and for the most number of endpoints as possible. Lastly, it would be important to 3314
compare with SAR studies. In this sense, you can introduce a layer of "validation" by 3315
comparing physicochemical interpretation for predictions. 3316

Our response: 3317

We conducted a detailed investigation and selected several specialized models for 3318
ADMET properties prediction that feature attention distribution (some models were 3319
excluded due to poor reproducibility or lack of an online demo). We then compared 3320
OmniMol with these models on eight key endpoints, ensuring that each endpoint 3321
included at least one specialized model with three instances. 3322

As illustrated in Fig. 4, on eight specific tasks (hERG, Ames, Carcinogenicity, 3323
H-HT/DILI, EC/EI, PPB, Acute toxicity, and Skin sensitivity), we presented the 3324
attention distribution graphs for OmniMol and the corresponding specialized models. 3325
By comparing the performance of OmniMol and the corresponding specialized mod- 3326
els on specific tasks, with a focus on the similarity of attention distribution on the 3327
same molecules, we roughly categorized the results into three levels: high similarity, 3328
moderate similarity, and slight similarity. It turns out that the similarity of atten- 3329
tion distribution between OmniMol and most existing specialized models falls at the 3330
high similarity and moderate similarity levels. In addition, OmniMol assigns reliable 3331
attention distributions and values to specific groups across nearly all eight endpoints. 3332
These indicates that OmniMol not only demonstrates performance comparable to 3333
other models, but also aligns with established prior knowledge. 3334

Then we conducted a series of structure-activity relationship (SAR) studies on four 3335
pivotal tasks (hERG, oral bioavailability, BBBP, Pgp-sub) to demonstrate and validate 3336
OmniMol exhibits robust performance in practical applications and is valuable for 3337

3340 SAR optimization (each task includes 3 group of instances). As illustrated in Fig. 5, in
3341 order to better demonstrate the practical applicability and generalization capability of
3342 OmniMol, the molecules selected in our SAR case studies do not include in the whole
3343 dataset, except for compounds **11a** and **12b**. To our delight, in the SAR study cases of
3344 all four tasks, OmniMol can accurately identify the key functional groups/atomic sites
3345 that influence the corresponding ADMET properties. More importantly, compared to
3346 the pre-optimized molecules, the attention values at the corresponding sites of the
3347 post-optimized molecules significantly decrease or disappear. In contrast, three other
3348 specialized models seem to be rigid as their attention distribution is almost identical
3349 before and after optimization. These changes in attention values aligns well with SAR
3350 study, greatly demonstrating the practical applicability and generalization capability
3351 of OmniMol.

3352

3353 **1.10 Comment 10**

3354

3355 The discussion is too shallow and must be drastically improved. For example, the
3356 requested comparison related to Table 2 explicability is useful. In the same sense,
3357 comparison with other models for the same endpoint could be interesting as well
3358 (please, describe the differences in dataset size, diversity, predictability metrics, etc).

3359 **Our response :** We thank the reviewer for this constructive suggestion. In the
3360 revised discussion, we have significantly expanded the comparison of OmniMol to other
3361 models. Specifically, we have included a detailed analysis of OmniMol’s performance
3362 across eight ADMET-P prediction tasks in comparison to existing specialized models.
3363 This comparison highlights OmniMol’s ability to achieve competitive or superior per-
3364 formance, as well as its capacity to generate interpretable attention distributions that
3365 align with established pharmaceutical knowledge.

3366 We also described the key differences between OmniMol and specialized models,
3367 such as the input data format (e.g., SMILES strings for OmniMol versus physico-
3368 chemical descriptors or substructure fingerprints for specialized models) and the
3369 practicability of OmniMol for real-world for SAR optimization applications. The
3370 discussion now provides a detailed analysis of SAR optimization cases. This compre-
3371 hensive comparison demonstrates OmniMol’s superior adaptability and generalization
3372 to unseen data, as well as its practical utility in real-world applications, such as SAR
3373 optimization in preclinical studies.

3374

3375 **1.11 Comment 11:**

3376 Authors must characterize the datasets with cheminformatics tools (calculation of
3377 fingerprints, similarity, physicochemical descriptors, etc).

3378 **Our response:** We used the RDKit package to conduct a detailed calculation
3379 of the similarity based on Morgan fingerprints and MACCS fingerprints (Supple-
3380 mentary Table 9), Additionally, we calculated seven key physicochemical descriptors
3381 (Supplementary Table 10) for each endpoint across the entire dataset.

3382 For calculation of similarity based on fingerprints (Supplementary Table 9), we
3383 computed the Tanimoto similarity between the training and testing sets, as well as
3384 between the training and validation sets for each endpoint. The similarity is defined
3385 as the average distance of all molecules in training set to their nearest molecules
3386 in the testing/validation set, respectively, which is calculated by three methods: 128
3387 and 2048 bits Morgan fingerprints, MACCS fingerprints. By analyzing the calculated
3388 results, we drawn several conclusions about the dataset. Such as the similarity cal-
3389 culated using MACCS fingerprints is significantly higher than that calculated using
3390 Morgan fingerprints, which indicates that the molecules within the same endpoint have
3391 strong associations with functional groups, which further highlights the importance
3392

of attention distribution (for more detailed information, please see the supplementary material). 3393
3394

For calculation of physicochemical descriptors (Supplementary Table 10), Molecular weight (MW), LogP, Hydrogen Bond Donors (HBD), Hydrogen Bond Acceptors (HBA), Topological Polar Surface Area (TPSA), Rotatable Bonds (RB) and Number of Rings (NR) were selected as they are important for virtual screening. The result showed that for each physicochemical descriptor, the training, testing, and validation sets within the same endpoint are very similar; however, there are significant differences between different endpoints. 3395
3396
3397
3398
3399
3400
3401
3402

1.12 Comment 12: 3403

The applicability domain of reported models should be reported and discussed. 3404

Our response: 3405

We appreciate the reviewer’s suggestion to discuss the applicability domain of OmniMol. In the revised discussion, we explicitly addressed the current scope (such as practical application in molecular optimization) and limitations of OmniMol’s applicability. Currently, OmniMol is designed for small drug-like molecule datasets, which represent the primary focus of our study. This limitation has been clearly acknowledged in the revised text. 3406
3407
3408
3409
3410
3411
3412

Furthermore, we elaborated on how OmniMol’s framework could potentially be extended to broader and more complex datasets in future work. We provided examples of potential applications, such as predicting HOMO-LUMO gaps in small molecules, adsorption energies in catalytic systems, and force field predictions for biological molecules. Additionally, we suggested that OmniMol could be scaled to macro-scale systems, such as industrial simulations for car or airfoil design. These discussions underline OmniMol’s potential for significant advancements in molecular and material sciences while transparently addressing its current domain limitations. 3413
3414
3415
3416
3417
3418
3419
3420
3421

1.13 Comment 13: 3422

X/Y-Randomization tests should be carried out to prove that models were not obtained by chance. 3423
3424

Our response: 3425

To validate that our model’s performance was not obtained by chance, we conducted Y-randomization tests on 8 datasets that included in our explainability result, for evaluating the statistical significance of our method. In this test, we randomly permuted the targeted properties (Y) while keeping the molecular input unchanged, which effectively destroys any true structure-activity relationships while maintaining the original representation distribution. This experiment directly tests whether the model has learned meaningful structure-property relationships or merely captured random correlations. 3426
3427
3428
3429
3430
3431
3432
3433

The Y-randomization results shown in Supplementary Table 17 (also discussed in Supplementary Sec. 16) provide strong evidence that our model’s performance is not due to chance. For classification tasks, the ROC-AUC scores on Y-randomized data consistently fell to near-random levels (0.448-0.658), showing marked deterioration from the original performance (0.794-0.942). This substantial drop in performance is particularly evident in tasks like F(20%) (ROC-AUC: 0.933 → 0.448) and hERG (ROC-AUC: 0.942 → 0.519). The accuracy metrics show similar degradation, with differences ranging from 15.5 to 38.7 percentage points compared to the original models. 3434
3435
3436
3437
3438
3439
3440
3441
3442

The regression task (LogP prediction) demonstrates even more dramatic evidence against chance correlation, with the R^2 value dropping from 0.964 to nearly zero (0.004) under Y-randomization, while the MAE increased almost six-fold from 0.223 3443
3444
3445

3446 **Supplementary Table 17** Y-randomization Test Results Comparing
3447 Original and Randomized Performance

Task Name	ROC_AUC		ACC		
	Original	Y-randomized	Original	Y-randomized	
F(20%)	0.933	0.448	0.880	0.740	
SkinSen	0.842	0.658	0.875	0.675	
Carcinogenicity	0.806	0.537	0.769	0.538	
hERG	0.942	0.519	0.887	0.500	
Ames	0.907	0.535	0.845	0.562	
H-HT	0.794	0.524	0.728	0.573	
DILI	0.932	0.563	0.915	0.574	
		R^2		MAE	
		Original	Y-randomized	Original	Y-randomized
LogP	0.964	0.004	0.223	1.322	

3450
3451
3452
3453
3454
3455
3456
3457
3458
3459
3460
3461 to 1.322. This severe performance degradation under Y-randomization is characteristic
3462 of models that have successfully captured true structure-property relationships rather
3463 than chance correlations.

3464 These results consistently demonstrate that across all tasks, the model’s perfor-
3465 mance collapses to near-random levels when the structure-property relationships are
3466 destroyed through Y-randomization, while maintaining strong predictive power on the
3467 original data. This pattern provides robust statistical evidence that our model has
3468 learned meaningful chemical structure-property relationships rather than achieving its
3469 performance by chance.

3470

3471 **1.14 Comment 14:**

3472 I strongly suggest to the authors that the datasets and other raw data (predictions,
3473 final conformers of compounds, etc) that they judge useful for the readership should
3474 be available for download.

3475 **Our response:** We acknowledge the importance of data availability for repro-
3476 ducibility and further research. We have made the following resources publicly available
3477 through our GitHub repository (<https://github.com/bowenwang77/OmniMol>):
3478

- 3479 • Complete training code and implementation details
- 3480 • Pre-configured conda environment with all dependencies
- 3481 • Pretrained model checkpoint for direct evaluation and transfer learning
- 3482 • Example datasets demonstrating the required data format and structure
- 3483 • Raw positional data of final molecular conformations discussed in Section 2.3
- 3484 • Attention values analyzed in Section 2.5

3485 To ensure full reproducibility, we provide detailed instructions for:
3486

- 3487 • Data preparation and processing
- 3488 • Model training and evaluation
- 3489 • Fine-tuning procedures for both classification and regression tasks

3490 While we cannot release the proprietary ADMET 2.0 dataset due to confidential-
3491 ity agreements with our data providers, we are developing an online platform that will
3492 allow researchers to utilize OmniMol for ADMET predictions. In the meantime, we
3493 recommend researchers use the publicly available HelixADMET dataset, which con-
3494 tains comprehensive ADMET property data suitable for validating our methodology.
3495 Our repository includes example scripts and tutorials demonstrating how to process
3496 and utilize such alternative datasets with our framework.
3497
3498

Additionally, we have included example molecules and their corresponding predictions, conformations, and attention analyses in the supplementary materials to help readers understand and verify our results. Researchers can readily reproduce these results using either our provided examples or their own datasets following our documented procedures.

2 Response to Reviewer #2:

2.1 Comment 1:

The work is a pure ML retrospective study almost entirely on one dataset collection where improvements on existing methods is small. This scope is more appropriate to an ML or cheminformatics journal until more convincing prospective work can be done.

Our response:

We respectfully disagree that this work is purely a retrospective ML study. While we demonstrate strong performance across 52 ADMET prediction tasks and chirality-awareness tasks, our work along with our revisions goes significantly beyond benchmark comparisons through several key contributions:

1. Practical Application in SAR Optimization We validate OmniMol’s practical application through structure-activity relationship (SAR) analysis using novel instances from published studies (Fig. 5). In all four endpoints, OmniMol assigns high attention values to endpoint-relevant groups in the pre-optimized molecules, while the attention at the corresponding sites in the post-optimized molecules diminishes or even disappears, which shows OmniMol is helpful in the time-consuming and labor-intensive SAR optimization stage as it can provide insights to researchers for optimizing druggability rapidly. Our SAR analysis spans multiple critical ADMET properties, showing valuable applications in real-world drug development.

2. Comprehensive Chemical Insights We demonstrate OmniMol’s ability to autonomously identify chemically relevant functional groups across multiple ADMET properties without explicit feature engineering. The model’s attention patterns align with established pharmaceutical knowledge and are even superior to the interpretations of specialized single-task models (Fig. 4). Unlike traditional descriptor-based approaches, OmniMol learns directly from 3D molecular structures computed by SMILES, enabling more flexible and comprehensive chemical feature recognition in untrained data and real-world applications.

3. Interpretability and Chemical Reasoning Through detailed attention analysis, we show how OmniMol recognizes specific structural features known to influence specific ADMET properties. The model aligns with established pharmaceutical knowledge and structural alert patterns (e.g., hERG channel binding sites, toxicophores) while maintaining flexibility to learn new structure-property relationships. This interpretability enables trust in the model’s predictions and provides actionable insights for molecular optimization.

4. Prospective Value Our method not only demonstrates state-of-the-art performance in traditional ADMET-P prediction tasks but also provides a framework for broader applications in molecular and material sciences. Future extensions of OmniMol could further enhance its impact, transforming it into a versatile tool for both research and practical applications in drug discovery and beyond.

These aspects collectively demonstrate that our work advances both the technical and practical aspects of molecular property prediction, making it suitable for *Nature Communications* broad readership interested in chemical science and drug discovery applications.

3552 **2.2 Comment 2:**

3553 The comparisons to existing methods are weak. **a.** I find it misleading that only one
3554 baseline was included in the main text where in fact other baselines seem to perform
3555 better. (see point 2b as well)

3556 **b.** Lack of any statistical rigor in the comparison of methods across multiple
3557 datasets. I recommend you look at methods such as [1] (later modified by [2]) or Pat
3558 Walter’s writing [3, 4]. Notably, I suspect HelixADMET and OmniMol would come
3559 out as equivalent in a more statistically rigorous comparison as they have equal num-
3560 ber of datasets where they outperform each other. I suspect your mean AUC difference
3561 is driven entirely by Rodent Acute Toxicity which is a notable outlier in the entire set
3562 of results.

3563 **Our response to 2a:**

3564 We appreciate the reviewer’s concern regarding the comprehensiveness of our com-
3565 parisons to existing methods. To address this, we have made several substantial
3566 improvements in our revision:

- 3568 • Regarding the comparison with HelixADMET and admetSAR 2.0, we acknowledge
3569 that direct inclusion in the main text was not appropriate due to dataset alignment
3570 issues. As these models used different datasets (detailed in Supplementary Sec. 1), we
3571 focused our main comparison on ADMETLab 2.0, where we could ensure identical
3572 dataset usage for fair evaluation.
- 3573 • We have expanded our comparative analysis to include traditional descriptor-
3574 based methods (Supplementary Sec. 13). Our comprehensive evaluation shows that
3575 OmniMol significantly outperforms both conventional and modern approaches:
 - 3576 – For classification tasks, OmniMol achieves a mean ROC-AUC of 0.905, surpassing
3577 ADMETlab 2.0 (0.863) and Random Forest with Morgan fingerprints (0.841)
 - 3578 – In regression tasks, OmniMol demonstrates superior performance with a mean
3579 R^2 of 0.844, compared to ADMETlab 2.0 (0.770) and Random Forest (0.621)
 - 3580 – This performance advantage is consistent across diverse physicochemical proper-
3581 ties and ADMET endpoints
- 3582 • To ensure rigorous comparison, we have conducted additional evaluations with
3583 HelixADMET using identical datasets and scaffold splitting settings. These results
3584 are detailed in our response to point 2b and Supplementary Sec. 17.

3586 **Our response to 2b:**

3587 **Response to 2b:**

3588 We have strengthened our evaluation framework by incorporating multiple addi-
3589 tional metrics:

- 3591 • Classification tasks: MCC, sensitivity, and specificity
- 3592 • Regression tasks: $RMSE$, CCC , Q_{F1}^2 , Q_{F2}^2
- 3593 • Statistical significance testing for chiral cliff predictions (Supplementary Sec. 14)

3594 We agree with the reviewer on the importance of incorporating statistical rigor
3595 in the comparison of methods across multiple datasets. In response, we have added
3596 statistical significance testing, detailed in Supplementary Sec. 14, focusing on the chiral
3597 cliff dataset where complete model access enabled fair comparisons.

3598 Regarding the comparison with HelixADMET, several important points warrant
3599 clarification. First, OmniMol and HelixADMET were developed on different datasets.
3600 We primarily compared against ADMETLab 2.0 because we could obtain their original
3601 training-evaluation-testing split, ensuring fair comparison conditions. Second, Helix-
3602 ADMET employs a three-stage training process (pretraining on unlabeled data, mixed
3603

3604

training on ADMET datasets, and task-specific finetuning) with significantly larger datasets, while our method corresponds only to their latter two phases.

We acknowledge the reviewer’s observation about Rodent Acute Toxicity performance differences. While the dataset sizes are comparable for this task (7,327 vs 7,646 samples), we noted that HelixADMET’s superior performance in the metabolism group, particularly for cytochromes P450 substrate datasets, corresponds to their significantly larger training data (9,236 instances versus our 3,344 instances from ADMETLab 2.0).

To ensure fair comparison under similar conditions, we conducted additional experiments focusing on metabolism datasets using scaffold splitting. The results in Supplementary Table 19 show that our method achieves superior performance on 7 out of 10 substrate datasets. Particularly noteworthy is the substantial improvement in substrate prediction tasks, with up to 11.2% AUC-ROC improvement over HelixADMET on the CYP3A4-sub dataset. Further analysis can be found at Supplementary 17.

Furthermore, our method achieved these results using only 10 datasets from the metabolism category, compared to HelixADMET’s utilization of over 40 datasets (663,004 instances including PCBA, or 225,075 excluding PCBA). Given established scaling laws in machine learning, where performance typically improves with larger datasets, these results suggest that OmniMol could achieve even better performance when trained on larger datasets comparable to HelixADMET’s full training set.

These findings demonstrate OmniMol’s robust capability in learning transferable representations across multiple ADMET tasks, with strong potential for further performance improvements when scaled to larger datasets. The results confirm the effectiveness of our approach, even when operating with more limited data resources compared to existing methods.

2.3 Comment 3

The geometry update process is much discussed and celebrated in the paper (such as line 794 “serves as the cornerstone”), but there is no evidence that it is useful.

a. Many methods already use a 3D conformers (or multiple conformers) as part of an ML model. What is novel here is the AlphaFold-like iterative geometry updating but from a corrupted true geometry. As far as I can tell from Section A.2, only the entire geometry module was removed, not specifically this unusual corrupted geometry part. The natural and simpler method is to just use the SE(3) encoder on the derived geometry (see technical point below on the source of the conformation).

b. The loss function is unusual with only supervision on the last step and is in contrast to e.g. diffusion models. I would not consider it essential to explore this for a first publication, but I suggest you explore alternatives.

Our response to 3a: The geometry update process in OmniMol is not merely an architectural choice but a carefully designed component supported by both theoretical foundations and empirical evidence:

- **Theoretical Foundations:** The corrupted geometry approach aligns with established principles in machine learning. Similar to Masked Language Models and Masked Auto-Encoders, denoising corrupted inputs serves as an effective self-supervised learning mechanism. This approach forces the model to learn robust molecular representations by reconstructing from imperfect geometries. Also, the process mimics real-world scenarios where initial molecular conformations may be imperfect or uncertain.
- **Empirical Evidence from Literature:** Multiple studies have demonstrated the effectiveness of geometry-based learning. Noisy Nodes [2] showed significant

3658 performance improvements through geometry noise addition. Uni-Mol [7] and Uni-
3659 Mol+ [3] validated the effectiveness of coordinate noise in molecular representation
3660 learning.

3661 • **Direct Evidence:** The effectiveness of our geometry update process is supported
3662 by ablation studies in DRFormer [6], which forms the backbone of our model,
3663 showed 2% MAE reduction with geometry supervision. Our analysis in Section 2.3
3664 demonstrates how this process effectively learns interatomic potentials.

3665 **Our response to 3b:**

3666 We acknowledge the reviewer’s observation regarding our loss function design. Our
3667 choice to supervise only the final step, similar to DRFormer [6], was a deliberate
3668 design decision balancing efficiency and effectiveness. While diffusion models indeed
3669 supervise all intermediate steps, our approach achieves strong performance while main-
3670 taining lower computational overhead. The final-step supervision strategy allows the
3671 model to flexibly discover optimal geometric pathways without being constrained by
3672 intermediate state requirements. This is particularly important in our recycling block
3673 architecture, where final-step supervision ensures geometric accuracy and prevents
3674 error accumulation throughout the iterative process. Nevertheless, we recognize the
3675 proven effectiveness of diffusion models in various domains. As part of our ongoing
3676 research, we are actively exploring the possibility of incorporating diffusion training
3677 pipelines within our framework as an alternative to the current iterative positional
3678 update approach. This could potentially offer new avenues for improvement while
3679 maintaining the core benefits of our current architecture.

3681 **2.4 Comment 4:**

3682
3683 The task embedding work is interesting. There is a long history of multi-task mod-
3684 els for cheminformatics e.g. [5, 6, 7] but I don’t know of ones that include dataset
3685 statistics as an input description of the task. The paper examines the embedding, but
3686 does not provide any evidence that anything is learned. In order to make the claim
3687 that something is learned, you need to compare the learned to the original features
3688 space. You could perhaps do this comparison with dimensionality reduction of the
3689 original feature space compared to the dimensionality of the learned embedding and
3690 demonstrate in some way that the learned embedding is better.

3691 **Our response:**

3692 We thank the reviewer for the constructive suggestion. We conducted a com-
3693 parative analysis between the initial and learned meta-embeddings, as illustrated in
3694 Supplementary Fig. 4 (further analyzed in Supplementary Sec. 15). The upper panel
3695 displays the original meta-embeddings derived from dataset statistics, while the lower
3696 panel shows the final learned meta-embeddings after model training. This comparison
3697 reveals several significant transformations that demonstrate the model’s capacity to
3698 learn meaningful task relationships:

3699
3700 1. Initially, regression and classification tasks within the same ADMET-P category
3701 were disparately positioned in the embedding space. For instance, regression-based
3702 toxicity tasks (BCF, LC₅₀) were distinctly separated from classification-based
3703 toxicity endpoints (DILI, hERG). Similarly, physicochemical property predictors
3704 (LogP, LogD, LogS) and absorption-related tasks were scattered across the space.
3705 In contrast, the learned embeddings exhibit clear organizational principles: tasks
3706 are clustered primarily by their ADMET-P category while maintaining a natural
3707 separation between regression and classification tasks. This organization suggests
3708 that the model has learned to recognize both the mechanistic similarities within
3709 ADMET-P categories and the fundamental differences in prediction types.
3710

UMAP visualization of OmniMol learned task relationship

Supplementary Fig. 4 Comparative analysis between the initial and learned meta-embeddingsg

- The learned embeddings reveal biologically meaningful relationships that were not encoded in the original feature space. A notable example is the positioning of DILI and H-HT tasks, which are known to be mechanistically related due to their involvement in drug-induced liver injury and hepatotoxicity. Similarly, eye corrosion

3711
3712
3713
3714
3715
3716
3717
3718
3719
3720
3721
3722
3723
3724
3725
3726
3727
3728
3729
3730
3731
3732
3733
3734
3735
3736
3737
3738
3739
3740
3741
3742
3743
3744
3745
3746
3747
3748
3749
3750
3751
3752
3753
3754
3755
3756
3757
3758
3759
3760
3761
3762
3763

3764 (EC) and eye irritation (EI) are closely related. While these tasks were initially
3765 embedded far apart, the learned representation positioned them in close proximity,
3766 despite no explicit encoding of this biological relationship in the input features. This
3767 demonstrates the model’s ability to capture intrinsic task relationships through the
3768 learning process.

3769 3. The spatial distribution characteristics of the embeddings underwent significant
3770 refinement. The initial embedding space showed heterogeneous clustering, particu-
3771 larly evident in the widely dispersed metabolism-related tasks from majority.
3772 The learned embeddings display a more uniform distribution while maintaining
3773 clear category boundaries, suggesting an optimization of the embedding space that
3774 balances task relationships with effective space utilization.

3775 These transformations from the initial to the final embedding space demonstrate
3776 that our model learns meaningful task representations beyond the original dataset
3777 statistics. The learned embeddings capture both explicit categorical relationships
3778 (ADMET-P groupings) and implicit biological connections (e.g., DILI and H-HT cor-
3779 relation), while maintaining a structured organization that reflects the fundamental
3780 nature of the prediction tasks (regression vs. classification). This provides strong evi-
3781 dence that the meta-learning process effectively discovers and encodes relevant task
3782 relationships, enhancing the model’s ability to transfer knowledge across related tasks.
3783

3784 2.5 Comment 5: 3785

3786 Section A.5: Judging by the style of writing and phrases such as “as demonstrated
3787 in your example” (where no example is in the text), I believe an LLM was used to
3788 generate this proof. As the proof is a scientific task and not just copy-editing, my
3789 understanding is that such a use needs to be acknowledged. Further, the proof you
3790 need is some statement about your final encoding values $\mathbf{CH}_l^{(m)}(\psi)$ and how they
3791 change with respect to reflections not simply the trivial cross product relationship
3792 covered in A.5
3793

3794 **Our response: 1. Clarification on the Use of Language Model:** We initially
3795 drafted the proof independently to ensure it correctly reflects our theoretical frame-
3796 work specific to chirality in molecular representations. To enhance clarity and precision
3797 in our mathematical notation and grammar, We acknowledge that we subsequently
3798 utilized a language model. This tool helped polish the mathematical expressions and
3799 language used in the proof to maintain academic rigor and readability. This process
3800 did not involve generating new content but rather refining the existing proof to meet
3801 high standards of scientific communication. **2. Revised Supplementary Proof on**
3802 **Encoding Values:** We have revised our supplementary material to include an analy-
3803 sis of how the circular harmonic encoding values $\mathbf{CH}_l^{(m)}(\psi)$ are affected by geometric
3804 reflections.
3805

3806 2.6 Comment 6: 3807

3808 I am unclear if you used the exact same data splits as used in AdmetLab2.0. If they
3809 are available, you should. I note that they report “validation and test sets by a ratio
3810 of 8:1:1, and stratified sampling was used when partitioning the data for classification
3811 to keep the ratio of the positive and negative instances in the three subsets balanced.”
3812 [8] and I see no mention of stratification in your paper. If they did stratification and
3813 you did not, it weakens the comparisons.

3814 **Our response:** We can confirm that the exact data splits used in ADMETLab
3815 2.0 were available to us, and we have indeed used the same splits in our experiments.
3816 This includes the 8:1:1 ratio for the training, validation, and test sets, as well as

the stratified sampling approach for partitioning the classification data to maintain a balanced ratio of positive and negative instances across the three subsets.

2.7 Comment 7:

It’s typical to do an alignment operation to find the minimum MAE or RMSE. If you didn’t do that here, you probably should.

Our response: We thank the reviewer for this valuable suggestion. While we acknowledge that alignment operations are commonly used to minimize MAE or RMSE, we must carefully consider their implementation to maintain the integrity of the evaluation process. We have conducted additional experiments with alignment operations, and these results are included in the Supplementary Table 18 (further analyzed in Supplementary Sec. 12) for reference.

Supplementary Table 18 Alignment for best MAE and RMSE

Task type	Task Name	MAE		RMSE	
		Without Alignment	With Alignment	Without Alignment	With Alignment
Physiochemical property	LogS	0.509	0.506	0.741	0.730
	LogD	0.290	0.289	0.371	0.370
	LogP	0.223	0.223	0.309	0.308
Absorption	Caco-2	0.195	0.193	0.275	0.273
	MDCK	0.179	0.179	0.251	0.249
Distribution	ppb	0.058	0.058	0.094	0.093
	VDss	0.347	0.343	0.660	0.610
	Fu	0.197	0.196	0.310	0.310
Toxicity	BCF	0.416	0.413	0.591	0.587
	IGC50	0.232	0.230	0.360	0.358
	LC50	0.532	0.529	0.782	0.776
	LC50DM	0.566	0.563	0.821	0.810
Mean		0.312	0.310	0.464	0.456

It is worth noting that while alignment operations can potentially improve performance metrics, our main results do not incorporate these adjustments to maintain evaluation rigor and prevent potential data leakage through alignment processes. The comparison in Supplementary Table 18 shows that alignment operations yield modest improvements in both MAE and RMSE across various ADMET prediction tasks.

2.8 Comment 8:

The generation of appropriate conformers is actually quite a tricky subject. See the documentation of Omega for example [9]. A single conformer almost certainly does not explain all properties of a molecule. At a minimum, the procedure you used to estimate the lowest energy conformer needs to be described. More broadly for future work you may want to consider conformers more holistically. **Our response:**

1. We would like to clarify that our model does not use only a single conformer per molecule. For each molecule in our dataset, we generate an ensemble of 11 conformers, consisting of 10 3D conformers and 1 2D conformer. During both training and testing, we randomly sample one conformer from this ensemble of 11 conformers for each batch. This allows our model to learn from a diverse set of physically plausible conformers for each molecule. The original code used for generating the conformers is provided under our public GitHub repository (<https://github.com/bowenwang77/OmniMol.git>) as (scripts/data_proc_SMILES2lmdb.py).
2. Our conformer generation procedure aims to efficiently sample low-energy conformers that capture the relevant conformational space of each molecule:

- 3870 • We start by generating an initial 3D structure for the molecule using a distance
 3871 geometry approach (implemented by RDKit’s `EmbedMolecule` function). This
 3872 embeds the molecular graph in 3D space, satisfying basic geometrical constraints.
 3873 We use different random seeds to generate multiple diverse initial 3D conformers.
 3874 • Each initial 3D conformer is then optimized using the Merck Molecular Force
 3875 Field (MMFF) to obtain a low-energy conformer (implemented by RDKit’s
 3876 `MMFFOptimizeMolecule` function). MMFF is a well-established classical force
 3877 field that provides a good balance between accuracy and efficiency for organic
 3878 molecules. The force field optimization drives the conformer towards a nearby
 3879 local minimum on the potential energy surface, yielding physically plausible
 3880 low-energy conformers.
 3881 • In rare cases where the 3D conformer generation or optimization fails, we fall
 3882 back to a 2D conformer generated by RDKit’s `Compute2DCoords` function. The
 3883 2D conformer provides a reasonable planar representation of the molecule when
 3884 3D conformer generation is not successful.
 3885 • To further augment the conformer ensemble, we also include one 2D conformer
 3886 for each molecule. 2D conformers can capture important structural information
 3887 for certain molecular properties and provide added diversity to the ensemble.
 3888
 3889 3. The following pseudo code outlines our conformer generation procedure:

3891 **Algorithm 1** Conformer Generation

3892 **Require:** SMILES string of molecule
 3893 **Ensure:** Ensemble of 11 conformers (10 3D + 1 2D)
 3894 1: `mol` \leftarrow `MolFromSmiles(SMILES)`
 3895 2: `mol` \leftarrow `AddHs(mol)` ▷ Add hydrogen atoms
 3896 3: `conformers` \leftarrow []
 3897 4: **for** $i \leftarrow 1$ **to** 10 **do**
 3898 5: `conformer` \leftarrow `EmbedMolecule(mol, randomSeed=i)`
 3899 6: **if** conformer generation successful **then**
 3900 7: `OptimizeMMFF(conformer)`
 3901 8: **else**
 3902 9: `conformer` \leftarrow `Compute2DCoords(mol)`
 3903 10: **end if**
 3904 11: `conformers.append(conformer)`
 3905 12: **end for**
 3906 13: `conformer2D` \leftarrow `Compute2DCoords(mol)`
 3907 14: `conformers.append(conformer2D)`
 3908 15: **return** `conformers`

3910 By generating an ensemble of conformers and randomly sampling from it during
 3911 training and testing, we aim to capture the relevant conformational diversity of
 3912 each molecule and improve the robustness of our model. The combination of 3D
 3913 conformers optimized by MMFF and 2D conformers provides a balance between
 3914 physical realism and computational efficiency.
 3915

3916 **2.9 Comment 9:**

3918 Line 854. You are missing a citation to what EGNN you are referring to. There are
 3919 many variants of equivariant networks out there.

3921 **Our response:** We thank the reviewer for pointing out this oversight. We have
 3922 now added a specific citation to the EGNN paper by Satorras et al. (2021) "E(n)

Equivariant Graph Neural Networks" which introduced the original EGNN architecture that we reference. We have also revised the text to more clearly specify the particular EGNN variant and mechanism we are discussing.	3923 3924 3925
The revised paragraph now explicitly connects to the specific EGNN formulation where the challenge of sum-based aggregation and atom count differences was first identified and discussed. This helps readers better understand the context of our discussion about prediction biases in molecular systems of varying sizes.	3926 3927 3928 3929
2.10 Comment 10:	3930 3931
I struggled to work through the math at the end of this section and it should be made more clear. You are using non-standard notation with i,j not referring to rows and cols of matrices, making it difficult to understand which vectors are row vector and which are column vectors. Also, whether 'diag' produces a vector or a matrix is unclear. Some intuition for the purpose of the inverse of \mathbf{A}_i could also help. Our response: We thank the reviewer for this important feedback regarding mathematical clarity. We have extensively revised this section to:	3932 3933 3934 3935 3936 3937 3938 3939
1. Clarify vector and matrix notation:	3940
• We now explicitly denote \mathbf{v}_{ij} as a column vector in \mathbb{R}^3	3941
• We specify that \mathbf{v}_{ij}^\top represents the transpose, resulting in a 1×3 row vector	3942
• We clarify that \mathbf{A}_i is a 3×3 symmetric positive definite matrix	3943 3944
2. Provide physical intuition:	3945
• \mathbf{A}_i represents the average outer product of direction vectors, capturing the spatial distribution of neighboring atoms	3946 3947 3948
• $(\mathbf{A}_i)^{-1}$ is used in deriving the analytical solution of the optimization problem, where \mathbf{A}_i is invertible when the neighborhood contains at least three non-coplanar atoms	3949 3950 3951
• The resulting \mathbf{c}_i represents the optimal center point that minimizes the distance to all projected vectors	3952 3953
3. Bridge the optimization problem to its solution:	3954
• We've added a brief explanation of how the optimization problem leads to the analytical solution through differentiation	3955 3956 3957
• We clarify that the solution is unique when \mathbf{A}_i is invertible (which is guaranteed by our graph construction)	3958 3959
We would be happy to provide additional mathematical details or further clarify any specific aspects that the reviewer finds unclear.	3960 3961 3962
2.11 Comment 11:	3963
Line 967 refers to w_+ and w_- that are neither defined nor used.	3964
Our response:	3965
w_+ and w_- are the weight corresponding to positive samples and negative samples. In the revised paper, we removed these two notations for simplicity.	3966 3967 3968
2.12 Comment 12:	3969
Citation 1 is good, but there are more recent studies you may be interested in, such as [10,11]	3970 3971 3972
Our response: We have carefully reviewed the more recent study and included them in the corresponding part in introduction.	3973 3974 3975

3976 **3 Response to Reviewer #3:**

3977 **3.1 Comment 1:**
3978

3979 The evaluation should be more comprehensive and clearly demonstrate which method-
3980 ological ideas or modules impact prediction performance. This would greatly enhance
3981 the value of the study. Currently, the results are not sufficiently differentiated. **Our**
3982 **response:**

3983 We appreciate the reviewer’s feedback on our model evaluation. We agree that a
3984 comprehensive evaluation demonstrating the impact of each methodological compo-
3985 nent is crucial for our study. In response to this comment, we have significantly revised
3986 and expanded our ablation study as follows:

3987 **Comprehensive Component Analysis.** We have conducted a more detailed
3988 discussion and analysis of each key component in our model, including the SE(3)-
3989 encoder, Meta Encoder, mixed training and Fine Tuning.

3990 **Revised Ablation Table** We have modified our ablation table to clearly illustrate
3991 the contribution of each component to the overall model performance. The new table
3992 provides a more granular view of how each module impacts prediction accuracy.

3993 **Performance Metrics** We have included additional performance metric0,s such
3994 as MCC and RMSE, to offer a more comprehensive understanding of our model
3995 performance.

3996 We believe these revisions provide a more comprehensive and clearly differentiated
3997 evaluation of our model’s components.

3998
3999 **3.2 Comment 2:**
4000

4001 The introduction could be more comprehensive, and the current state-of-the-art
4002 approaches should be mentioned and described.

4003 **Our response:** We have substantially enhanced our introduction, incorporating:
4004

- 4005 • A more extensive research background
- 4006 • A formal mathematical definition of the hypergraph view
- 4007 • A description of the current state-of-the-art model, ADMETLab 2.0

4008 These additions provide a more comprehensive context for our work and situate it
4009 within the current landscape of ADMET prediction research.

4011 **3.3 Comment 3:**
4012

4013 The authors should explicitly indicate what is new in their work and how they have
4014 built upon existing methods.

4015 **Our response:** In the revised manuscript, we have explicitly highlighted the
4016 novel aspects of OmniMol. Our method builds upon the foundation of Graphormer,
4017 incorporating several key innovations:

- 4019 1. A specialized encoder that converts task-related meta-information into task embed-
4020 dings.
- 4021 2. A task-routed mixture of experts (t-MOE) architecture.
- 4022 3. The DR-Label strategy and iterative geometry update mechanism.
- 4023 4. An SE(3)-encoder for enhanced chirality awareness.

4024 A comprehensive analysis of these components and their contributions can be found
4025 in the ablation study presented in the supplementary materials and in our response
4026 to Comment 1.

4028

3.4 Comment 4:	4029
A clear comparison to other methods is missing, and the results should systematically demonstrate the advantages of their method compared to alternative approaches. The effects and influence factors should be analyzed separately.	4030 4031 4032 4033 4034
Our response: In the revised manuscript, we have conducted a systematic comparison of OmniMol with state-of-the-art methods for ADMET prediction tasks. Specifically:	4035 4036 4037
 • We have compared OmniMol with ADMETLab 2.0, HelixADMET, and admeSAR 2.0 in both the results section and the supplementary materials, including Supplementary Table 4, Supplementary Sec. 1 and Sec. 17. • We have clearly delineated the differences in datasets used by various methods to ensure fair comparison. It is noteworthy that only ADMETLab 2.0 utilized the same dataset as OmniMol. • For chiral awareness datasets, we have benchmarked our method against the baseline methods described in the ChiRo paper in Supplementary Sec. 14. 	4038 4039 4040 4041 4042 4043 4044 4045
This comprehensive comparison systematically demonstrates the advantages of OmniMol over alternative approaches. The effects of individual components and influencing factors have been analyzed separately in our ablation studies Supplementary Sec. 3 , providing a clear understanding of each element’s contribution to the overall performance of OmniMol.	4046 4047 4048 4049 4050 4051
3.5 Comment 5:	4052
Additionally, the reported results should include standard deviations	4053 4054 4055
Our response: To ensure a fair comparison, we have reported the standard deviations(summarized over 6 different runs) for the model trained on the full dataset, corresponding to model D in our ablation study. These results are presented in the appendix under Supplementary Sec. 2	4056 4057 4058 4059 4060
3.6 Comment 6:	4061
It is customary to compare a new method to existing methods, such as Chemprop, Random Forest, KNN, etc. However, in this study, there seems to be only a comparison to a single method, which is also not directly named.	4062 4063 4064 4065
Our response:	4066
In the revised manuscript, we have expanded our comparative analysis to include several state-of-the-art methods:	4067 4068 4069
 • ADMETLab 2.0 • HelixADMET • admeSAR 2.0 	4070 4071 4072 4073
The results of this comparison are visually represented in a radar graph for clear interpretation.	4074 4075
It is worth noting that ADMETLab 2.0 employs a multi-task graph attention (MGA) framework. We refer to this work as ADMETLab 2.0, as it is more widely recognized under this name in the field.	4076 4077 4078
This comprehensive comparison provides a broader context for evaluating the performance of our proposed method against established approaches in the field of ADMET prediction.	4079 4080 4081

4082 For comparison with traditional method (Random Forest with Morgan binary Fin-
4083 gerprints with radius 2 and 2048 bits as suggested), we provided detailed result in our
4084 reply to Comment 16.

4085

4086 3.7 Comment 7:

4087

4088 It is also common practice to compare against baselines using different challenging
4089 splits, such as random splits, scaffold splits, or other cluster splits. In this study, it
4090 appears that only one split was used.

4091

Our response:

4092

4093

4094

4095

4096

4097

4098

Supplementary Table 19 Comparison of model performance on metabolism datasets using scaffold split and HelixADMET datasets. The table reports the AUC-ROC scores for OmniMol and HelixADMET (LiteGEM model). The best performance for each task is highlighted in bold.

4099

4100

4101

4102

4103

4104

4105

4106

4107

4108

4109

4110

4111

4112

4113

4114

4115

4116

4117

4118

4119

4120

4121

4122

4123

4124

4125

4126

4127

4128

4129

4130

4131

4132

4133

4134

We thank the reviewer for pointing out the importance of evaluating model performance under different data splitting strategies. To address this concern, we have conducted additional experiments using scaffold splitting, which is recognized as a more challenging and realistic assessment of a model’s generalization ability, particularly in the context of drug discovery.

In the revised manuscript, we have included Supplementary Sec. 17. In this section, we describe our experimental setup following the scaffold splitting procedure and present the results comparing OmniMol with HelixADMET under these conditions. Detailed description of experiment setting can be found in the corresponding section.

Our experimental result, displayed in Supplementary Table 19, show that OmniMol maintains strong performance under scaffold splitting, outperforming HelixADMET significantly on the substrate datasets and achieving better results on the inhibitor datasets. This indicates that OmniMol effectively generalizes to novel chemical scaffolds, which is crucial for predicting the properties of unseen compounds in practical applications.

By including these additional experiments, we have strengthened the evaluation of our model’s robustness across different data splitting strategies.

4130 3.8 Comment 8:

The discussion section is very brief and does not adequately highlight the limitations of the approach.

Our response:

We thank the reviewer for pointing out the need to highlight limitations. In the revised discussion, we explicitly outlined the primary limitation of OmniMol, which lies in its current focus on datasets of small drug-like molecules. While this focus allowed us to demonstrate OmniMol’s capabilities within a well-defined scope, we acknowledge that the method’s applicability to larger and more diverse systems remains to be fully explored.

We also discussed potential directions for future work to address these limitations. For example, we suggested extending OmniMol to diverse datasets, including macro-scale systems and other molecular property prediction tasks, such as HOMO-LUMO gap prediction, adsorption energy computation, and force field predictions. These extensions would broaden OmniMol’s applicability and enhance its value for a wider range of applications. By transparently discussing these limitations and proposing future directions, we aim to provide a balanced and comprehensive perspective on OmniMol’s current and potential capabilities.

3.9 Comment 9:

P2 73-79 – more literature information should be included. Very optimistic description although relevant endpoints are still very challenging.

Our response: We have conducted additional literature research and incorporated relevant discussions into the original text regarding the significance of ADMET prediction, representative studies, and the existing limitations and challenges. For more details, please see the first paragraph of Introduction.

3.10 Comment 10:

P2 93 - please differentiate more between the general problem and a concrete use case. Very short description of a complex relationships.

Our response:

We thank the reviewer for the constructive suggestion. We have thoroughly revised our manuscript to better differentiate between the general problem and its concrete application. In the second paragraph, we have formalized the general problem of imperfect annotation in molecular representation learning through mathematical notation, introducing sets \mathcal{M} and \mathcal{E} to represent molecules and properties, and formally defining imperfect annotation as $\exists e_i \in \mathcal{E}$ such that $\mathcal{M}_{e_i} \subsetneq \mathcal{M}$. This mathematical framework provides a rigorous foundation for understanding the general challenge of imperfectly annotated molecular datasets.

We then present ADMET-P prediction as a representative example of this general challenge. To strengthen the connection between the general problem and its concrete application, we have expanded the third paragraph of introduction. This includes an analysis of how imperfect annotation affects model development, incorporating specific examples such as ADMETlab 2.0’s multi-task graph attention framework, and discussing the computational complexity implications. We have also elaborated on the challenges in capturing property relationships and training synchronization issues in existing approaches. These revisions provide a more comprehensive treatment of the complex relationships between the general problem of imperfect annotation in molecular representation learning and its specific manifestation in ADMET-P prediction tasks.

3.11 Comment 11:

P4 1168 -"practical clinical data analysis" -> which clinical data and what kind of analysis?

4188 **Our response:** We thank the reviewer for pointing out this imprecise terminology.
4189 We sincerely apologize for the incorrect use of "clinical data analysis" to "preclinical
4190 data analysis" to more accurately reflect our intended meaning. Specifically, we refer
4191 to preclinical studies such as structure-activity relationship (SAR) analysis during the
4192 Hit-to-Lead stage of drug discovery, where researchers optimize the activity and drug-
4193 gability of lead compounds. In this context, "analysis" encompasses the application
4194 of medicinal chemistry knowledge for compound optimization, including the exam-
4195 ination of activity/toxicity-related structural features and properties that influence
4196 druggability.

4197 In our revision, we have clarified this by explicitly stating "when conducting
4198 practical preclinical data analysis of molecule properties (such as structure-activity
4199 relationship study to optimize activity and druggability of lead compounds during the
4200 Hit-to-Lead stage)..." This modification better connects to our main argument about
4201 the importance of model explainability in molecular representation learning, partic-
4202 ularly when the predictions may influence drug development decisions. We have also
4203 added relevant citations to strengthen this point.

4204

4205 **3.12 Comment 12:**

4206 P4 1173 -, "We" typo
4207

4208 **Our response:** We appreciate the reviewer for noting the typo. It has been
4209 corrected.
4210

4211

4212 **3.13 Comment 13:**

4213 P4 |209-220 - This paragraph should talk about the concrete results that came out
4214 of this study, for example, the improvement in prediction performance compared to
4215 strong baseline methods. Instead, this paragraph talks about the hypothetical benefits
4216 of the method's ideas, making it unclear if the method has an actual benefit.

4217 **Our response:** We have thoroughly revised the paragraph to focus on concrete
4218 empirical results. Specifically, we now highlight that OmniMol achieves state-of-the-
4219 art performance in 47 out of 52 ADMET-P prediction tasks and demonstrates superior
4220 performance in chirality awareness tasks. We also include experimental evidence of
4221 the model's explainability across multiple relationship scales: molecular interactions,
4222 molecule-property associations, and property correlations.

4223

4224 **3.14 Comment 14:**

4225

4226 P4 | 214 -" OmniMol lays the groundwork for applying generalized MRL across
4227 diverse datasets, thereby capturing fundamental physical mechanisms." -> very strong
4228 statement which needs a very good validation

4229 **Our response:** We appreciate the reviewer's astute observation regarding the
4230 strength of this statement. We acknowledge that such a claim requires substantial
4231 validation, and we thank the reviewer for the opportunity to clarify our position.

4232 The underlying premise of our statement is that a more generalized multi-task
4233 representation learning (MRL) method across diverse tasks can potentially indicate
4234 shared and general knowledge gained across these tasks, which may approximate fun-
4235 damental mechanisms at the physical level. As a step towards validating this concept,
4236 our explainability results demonstrate that OmniMol can function as a learning-based
4237 conformation relaxation method without explicit intermediate supervision, suggesting
4238 the acquisition of certain physical mechanisms.

4239 However, we concur that further research is necessary to fully substantiate this
4240 claim across a broader range of physical phenomena. In light of this, we have revised

the relevant section to more accurately reflect the current state and potential of our work. The revised text now emphasizes: 4241

1. OmniMol’s demonstrated performance in ADMET-P predictions and chirality awareness tasks. 4242
2. The model’s potential for capturing shared knowledge across diverse molecular tasks. 4243
3. The need for additional research to validate its broader applicability across diverse datasets and physical mechanisms. 4244

This revision aims to maintain the significance of our findings while acknowledging the need for continued investigation to fully realize the potential of our approach in capturing fundamental physical mechanisms. 4245

3.15 Comment 15: 4251

P4 | 219 - "broader range of AI driven techniques adhering to the laws of nature" This is an unusual and not very scientific statement 4252

Our response: We thank the reviewer for pointing out the imprecise nature of this statement. We agree that it lacks scientific specificity and may be open to misinterpretation. In our revision, we have removed this phrase and instead focused on the potential of OmniMol to contribute to the development of more comprehensive and physically informed predictive models in chemistry and materials science. This revision maintains the essence of our work’s potential impact while adhering to more precise scientific language. 4253

3.16 Comment 16: 4254

P7 | Table 1 - It is not clear to which single method (or multiple methods?) OmniMol is compared in Table 1. There is just a single number for each endpoint. It should be made clear to the reader which method OmniMol is compared against. There is also a range of baseline methods in the molecular property prediction domains, e.g. Random Forest with Morgan binary Fingerprints with radius 2 and 2048 bits, that are often strong baselines and should be considered in every study presenting a new method. 4255

Our response: 4256

To provide comprehensive performance benchmarking, we additionally compared OmniMol against both traditional machine learning approaches and state-of-the-art deep learning frameworks, as illustrated in Supplementary Table 20 and Supplementary Table 21. We implemented the widely-used baseline of Random Forest with Morgan fingerprints (radius=2, 2048 bits), which serves as a robust traditional machine learning benchmark. 4257

As shown in Supplementary Table 20 and Supplementary Table 21, OmniMol consistently outperforms both baseline methods across all metrics. For classification tasks, OmniMol achieves a mean ROC-AUC of 0.905, substantially surpassing both ADMETlab 2.0 (MGA framework) (0.863) and Random Forest (0.841). The advantage of OmniMol is even more evident in regression tasks, where it achieves a mean R^2 of 0.844, representing significant improvements over both ADMETlab 2.0 (0.770) and Random Forest (0.621). This superior performance is consistent across various physicochemical properties and ADMET endpoints. 4258

The traditional Random Forest approach, while providing a reasonable baseline, consistently underperforms compared to both deep learning methods. This performance gap is particularly evident in regression tasks, where Random Forest’s mean R^2 (0.621) is substantially lower than both modern approaches, highlighting the advantages of deep learning architectures in capturing complex structure-property relationships. 4259

Additional comparisons with other state-of-the-art methods, including HelixADMET’s LiteGEM framework, are provided in Supplementary Table 1. While dataset 4260

4294 **Supplementary Table 20** Compare with traditional methods

Task Name	ROC-AUC			ACC			MCC		
	RF	ADMETlab 2.0	OmniMol	RF	ADMETlab 2.0	OmniMol	RF	ADMETlab 2.0	OmniMol
4297 Pgp-inh	0.943	0.922	0.942	0.849	0.867	0.907	0.685	0.723	0.799
Pgp-sub	0.846	0.840	0.907	0.776	0.768	0.840	0.552	0.538	0.631
4298 HIA	0.883	0.866	0.940	0.907	0.924	0.949	0.491	0.687	0.697
F(20%)	0.723	0.833	0.933	0.790	0.750	0.880	0.366	0.414	0.617
4299 F(30%)	0.764	0.848	0.910	0.703	0.802	0.891	0.308	0.580	0.678
4300 BBBP	0.905	0.908	0.922	0.878	0.862	0.853	0.620	0.718	0.748
CYP1A2-inh	0.907	0.928	0.934	0.831	0.852	0.885	0.660	0.704	0.791
4301 CYP1A2-sub	0.822	0.737	0.976	0.784	0.649	0.892	0.569	0.298	0.618
CYP2C19-inh	0.889	0.913	0.924	0.814	0.839	0.857	0.627	0.679	0.773
4302 CYP2C19-sub	0.652	0.758	0.958	0.654	0.654	0.923	0.272	0.300	0.517
CYP2C9-inh	0.888	0.919	0.919	0.807	0.841	0.872	0.546	0.671	0.735
4303 CYP2C9-sub	0.763	0.725	0.902	0.707	0.707	0.866	0.378	0.386	0.639
4304 CYP2D6-inh	0.867	0.892	0.917	0.866	0.824	0.898	0.510	0.558	0.693
CYP2D6-sub	0.758	0.847	0.903	0.663	0.775	0.865	0.326	0.553	0.743
4305 CYP3A4-inh	0.885	0.921	0.922	0.793	0.832	0.848	0.568	0.659	0.721
4306 CYP3A4-sub	0.778	0.776	0.810	0.733	0.713	0.772	0.467	0.437	0.478
T12	0.784	0.801	0.851	0.738	0.727	0.771	0.435	0.478	0.525
4307 hERG	0.937	0.943	0.942	0.870	0.889	0.887	0.740	0.778	0.830
Hepatotoxicity	0.796	0.814	0.794	0.707	0.720	0.728	0.395	0.461	0.415
4308 DILI	0.912	0.924	0.932	0.851	0.894	0.915	0.702	0.793	0.807
Ames	0.894	0.902	0.907	0.811	0.807	0.845	0.615	0.606	0.707
4309 ROA	0.832	0.853	0.835	0.755	0.778	0.777	0.467	0.549	0.547
4310 FDAMDD	0.834	0.804	0.837	0.760	0.736	0.793	0.519	0.471	0.522
SkinSen	0.821	0.707	0.842	0.750	0.775	0.875	0.393	0.462	0.340
4311 Carcinogenicity	0.747	0.788	0.806	0.683	0.731	0.769	0.365	0.476	0.507
EC	0.992	0.983	0.996	0.943	0.957	0.978	0.883	0.908	0.980
4312 EI	0.974	0.982	0.982	0.931	0.952	0.962	0.815	0.876	0.955
Respiratory	0.820	0.828	0.876	0.757	0.764	0.821	0.485	0.514	0.639
4313 NR-AR	0.886	0.886	0.931	0.985	0.890	0.985	0.755	0.348	0.717
4314 NR-AR-LBD	0.847	0.915	0.934	0.978	0.936	0.983	0.603	0.472	0.720
4315 NR-AhR	0.919	0.943	0.952	0.907	0.862	0.937	0.426	0.573	0.731
NR-Aromatase	0.740	0.852	0.884	0.958	0.849	0.961	0.251	0.264	0.320
4316 NR-ER	0.783	0.771	0.837	0.891	0.815	0.921	0.210	0.320	0.498
NR-ER-LBD	0.827	0.850	0.915	0.964	0.903	0.966	0.482	0.364	0.590
4317 NR-PPAR-Y	0.821	0.893	0.896	0.968	0.896	0.979	-0.007	0.344	0.590
SR-ARE	0.826	0.863	0.888	0.864	0.827	0.901	0.332	0.469	0.606
4318 SR-ATAD5	0.819	0.874	0.867	0.971	0.919	0.975	0.394	0.361	0.365
4319 SR-HSE	0.830	0.907	0.912	0.945	0.868	0.959	0.182	0.393	0.598
SR-MMP	0.883	0.927	0.953	0.890	0.897	0.940	0.492	0.660	0.800
4320 SR-p53	0.835	0.881	0.911	0.936	0.841	0.960	0.240	0.365	0.599
4321 Mean	0.841	0.863	0.905	0.834	0.822	0.890	0.478	0.530	0.645

4323 **Supplementary Table 21** Compare with traditional methods

Task Name	R^2			MAE			RMSE		
	RF	ADMETlab 2.0	OmniMol	RF	ADMETlab 2.0	OmniMol	RF	ADMETlab 2.0	OmniMol
4325 LogS	0.715	0.854	0.878	0.866	0.588	0.509	1.186	0.850	0.741
4327 LogD	0.716	0.892	0.924	0.564	0.347	0.290	0.751	0.462	0.371
4328 LogP	0.789	0.957	0.964	0.578	0.256	0.223	0.787	0.357	0.309
Caco-2	0.543	0.746	0.886	0.298	0.222	0.195	0.543	0.307	0.275
4329 MDCK	0.633	0.731	0.801	0.236	0.199	0.179	0.339	0.291	0.251
ppb	0.582	0.733	0.856	0.117	0.083	0.058	0.169	0.135	0.094
4330 VDss	0.659	0.782	0.809	0.560	0.457	0.347	0.838	0.670	0.660
4331 Fu	0.505	0.763	0.848	0.400	0.263	0.197	0.530	0.367	0.310
BCF	0.586	0.786	0.800	0.648	0.435	0.416	0.837	0.603	0.591
4332 IGC50	0.550	0.723	0.858	0.477	0.335	0.232	0.632	0.496	0.360
4333 LC50	0.604	0.745	0.789	0.838	0.643	0.532	1.076	0.863	0.782
4334 LC50DM	0.572	0.524	0.716	0.608	0.692	0.566	0.837	0.994	0.821
4335 Mean	0.621	0.770	0.844	0.516	0.377	0.312	0.710	0.533	0.464

4336 differences prevent direct inclusion in the main results, these additional compar-
4337 isons further support the robust performance of our approach across diverse ADMET
4338 prediction tasks.

4341 3.17 Comment 17:

4342 P8 1394 -Is this a significant difference? include standard deviations and significance
4343 test

4344 **Our response:**

4345 We conducted the significance test between OmniMol and the best traditional
4346 baseline on the Chiral Cliff dataset. We ran both method 6 times. For each time, we

Supplementary Table 22 Performance comparison between OmniMol and the best traditional baseline (AP+BCD:GB) on chiral cliff dataset across multiple metrics.

Metric	Model	Mean	STD
Acc	OmniMol	0.8015	0.0041
	AP+BCD:GB	0.7694	0.0028
MCC	OmniMol	0.5974	0.0083
	AP+BCD:GB	0.5316	0.0059
F1	OmniMol	0.7747	0.0046
	AP+BCD:GB	0.7356	0.0045
McNemar Statistic		3.2550	1.0883
McNemar P value		0.0857	0.0526
Different Predictions Percentage		20.6418	0.9516

calculate the Acc, MCC, F1-score for each method. We also calculate the McNemar Statistic and McNemar P Value for the comparison of two different methods.

From the result, our comparative analysis of OmniMol and the best traditional baseline reveals consistent performance advantages for OmniMol across all standard evaluation metrics on the chiral cliff dataset. OmniMol achieved superior performance with an accuracy of 0.802 ± 0.004 compared to the baseline’s 0.769 ± 0.003 . Notably, the Matthews Correlation Coefficient (MCC), which is particularly valuable for imbalanced classification tasks, showed a more substantial difference: OmniMol attained an MCC of 0.597 ± 0.008 versus the baseline’s 0.531 ± 0.006 . The F1 score further confirmed this trend with OmniMol achieving 0.775 ± 0.005 compared to the baseline’s 0.736 ± 0.004 . Based on these comprehensive evaluations, OmniMol demonstrates superior predictive capabilities with statistically meaningful improvements across all performance metrics. The magnitude of improvement (approximately 3.2 percentage points in accuracy and 6.6 percentage points in MCC) represents a substantial advancement in predictive power for this specific classification task.

References

- [1] Lowik Chanussot, Abhishek Das, Siddharth Goyal, Thibaut Lavril, Muhammed Shuaibi, Morgane Riviere, Kevin Tran, Javier Heras-Domingo, Caleb Ho, Weihua Hu, et al. Open catalyst 2020 (oc20) dataset and community challenges. *ACS Catalysis*, 11(10):6059–6072, 2021.
- [2] Jonathan Godwin, Michael Schaarschmidt, Alexander Gaunt, Alvaro Sanchez-Gonzalez, Yulia Rubanova, Petar Veličković, James Kirkpatrick, and Peter Battaglia. Simple gnn regularisation for 3d molecular property prediction & beyond. 2022.
- [3] Shuqi Lu, Zhifeng Gao, Di He, Linfeng Zhang, and Guolin Ke. Highly accurate quantum chemical property prediction with uni-mol+, 2023.
- [4] Maho Nakata and Tomomi Shimazaki. Pubchemqc project: a large-scale first-principles electronic structure database for data-driven chemistry. *Journal of chemical information and modeling*, 57(6):1300–1308, 2017.
- [5] Raghunathan Ramakrishnan, Pavlo O Dral, Matthias Rupp, and O Anatole Von Lilienfeld. Quantum chemistry structures and properties of 134 kilo molecules. *Scientific data*, 1(1):1–7, 2014.

4400 [6] Bowen Wang, Chen Liang, Jiaze Wang, Furui Liu, Shaogang Hao, Dong Li, Jianye
4401 Hao, Guangyong Chen, Xiaolong Zou, and Pheng-Ann Heng. Dr-label: Improving
4402 gnn models for catalysis systems by label deconstruction and reconstruction. *arXiv*
4403 *preprint arXiv:2303.02875*, 2023.
4404
4405 [7] Gengmo Zhou, Zhifeng Gao, Qiankun Ding, Hang Zheng, Hongteng Xu, Zhewei
4406 Wei, Linfeng Zhang, and Guolin Ke. Uni-mol: A universal 3d molecular represen-
4407 tation learning framework. 2023.
4408
4409
4410
4411
4412
4413
4414
4415
4416
4417
4418
4419
4420
4421
4422
4423
4424
4425
4426
4427
4428
4429
4430
4431
4432
4433
4434
4435
4436
4437
4438
4439
4440
4441
4442
4443
4444
4445
4446
4447
4448
4449
4450
4451
4452

3446 Appendix B Response to comments

3447

3448 We sincerely thank all reviewers for their thorough and constructive feedback, which
3449 has helped us significantly improve our manuscript. Below, we provide detailed point-
3450 by-point responses to each reviewer’s comments. We have substantially expanded our
3451 supplementary materials to include comprehensive analyses and additional results. We
3452 encourage reviewers to consult the supplementary section, which contains important
3453 supporting evidence and detailed explanations referenced in our responses.

3454

3455 1 Response to Reviewer #1:

3456

3457 1.1 Comment:

3458 I want to thank the authors for their hard efforts. They addressed properly all my
3459 suggestions except for "comment 12". In this suggestion (The applicability domain of
3460 reported models should be reported and discussed), I asked the authors to provide
3461 some method to evaluate the range of properties and activities of the test set in
3462 comparison with the training set to assess if or not the predictions were reliable in
3463 terms of "the model indeed learned from presented data" or "if the prediction is just
3464 a guess". In this sense, it would be valuable to implement this tool so that users can
3465 evaluate the trustability of predictions for unseen data.

3466 Our response:

3467 We sincerely thank the reviewer for clarifying the concern and providing valuable
3468 references to guide our revision. We acknowledge that in our original submission, we
3469 misunderstood the concept of “Applicability Domain (AD)” and mistakenly discussed
3470 potential application scenarios instead of quantitatively evaluating the reliability of
3471 model predictions based on training and test data distribution comparisons.

3472 In response, we have now conducted a thorough quantitative analysis following
3473 the suggestions and references provided by the reviewer. Specifically, we have defined
3474 the AD using seven vital molecular descriptors: Molecular Weight (MolWt), LogP,
3475 Hydrogen Bond Donors (HBD), Hydrogen Bond Acceptors (HBA), Topological Polar
3476 Surface Area (TPSA), Rotatable Bonds (RB), and Ring Count (RC). We have clearly
3477 defined the criteria as follows:

3478

- 3479 • For MolWt and LogP:

- 3480 – “In domain”: values within the 1%-99% percentile range of the training set.

- 3481 – “Warning”: values within 0%-1% or 99%-100% percentiles.

- 3482 – “Out of domain”: values outside the minimum or maximum of the training set.

3483

- 3484 • For HBD, HBA, TPSA, RB, and RC:

- 3485 – “In domain”: values within the 0%-99% percentile range of the training set.

- 3486 – “Warning”: values within the 99%-100% percentile range.

- 3487 – “Out of domain”: values beyond the maximum observed in the training set.

3488

3489 To clearly demonstrate our model’s AD, we have provided:

3490 1. A detailed supplementary table (`Applicability_Domain.csv`) publicly avail-
3491 able in our GitHub repository ([https://github.com/bowenwang77/OmniMol/blob/
3492 master/Applicability_Domain.csv](https://github.com/bowenwang77/OmniMol/blob/master/Applicability_Domain.csv)), containing numerical boundaries (In domain,
3493 Warning, or Out of domain) for each descriptor and the percentage of test set
3494 molecules that fall in corresponding “In domain” region for the full dataset as well
3495 as individual endpoints.

3496 2. Supplementary Fig. 5 (included below) visually illustrates the AD boundaries, dis-
3497 tribution of the entire training set molecules, and clearly highlights the proportion
3498 of test set compounds falling within the “In domain” region.

Supplementary Fig. 5 Applicability Domain of OmniMol. Numerical boundaries of molecular descriptors used to define the AD and distribution of training set compounds. Numbers above bars represent boundary values; percentages below each bar indicate compounds of test set within “In domain” region. The values on either side of the percentages represent the minimum and maximum of the test set, respectively.

From our analysis:

- For the entire dataset (across all endpoints), 90.29% of the test set molecules fall within the “In domain” region considering all seven descriptors. Importantly, no compound from the test set was found in the “Out of domain” region, confirming that predictions are made primarily via interpolation rather than extrapolation.
- Endpoint-specific analyses further confirm that, for most individual ADMET-P tasks, more than 90% of test set molecules fall within the established “In domain” region, further validating the reliability of our model predictions. For example, as illustrated explicitly in our supplementary file, the Ames mutagenicity endpoint dataset has 93.74% of molecules within the “In domain” region.

We appreciate the reviewer’s valuable suggestion regarding providing users with a practical tool to assess AD prior to prediction. To further enhance the transparency, utility, and trustworthiness of our model, we will incorporate an easy-to-use script into our GitHub repository. This script will allow users to input a SMILES string of their molecule of interest, and it will automatically evaluate whether the molecule falls within the “In domain”, “Warning”, or “Out of domain” regions for all descriptors, both for the entire dataset and for specific endpoints. We anticipate this tool will assist users in making informed decisions about the reliability of predictions from our model.

3552 We sincerely thank the reviewer again for the insightful comment and helpful
3553 references, which have significantly improved the rigor and clarity of our manuscript.
3554

3555 **2 Response to Reviewer #2:**

3556

3557 **2.1 Comment 1:**

3558 Nothing in your response is about prospective application. I am not disagreeing that
3559 your attention analysis is suggestive, but it is all retrospective. I especially find it
3560 laughable that you title response point 4 "Prospective Value" with things that you
3561 have not done prospectively or demonstrated any value.

3562 **Our response:**

3563 We sincerely thank the reviewer for highlighting this important issue and fully
3564 acknowledge that our previous wording inadvertently caused misunderstanding.

3565 Upon carefully reviewing our manuscript and our previous responses, we con-
3566 firm that the phrase "*Prospective Value*" appeared only once in our earlier response
3567 to Reviewer #2 and does not occur in our manuscript itself. This phrase was
3568 intended from a machine learning model-development perspective, referring generally
3569 to potential future applications and extensions. We now fully realize that the term
3570 "prospective" has a precise and fundamentally different meaning in chemistry and drug
3571 discovery research, specifically referring to experimental validations conducted on pre-
3572 viously unexplored molecules or hypotheses. We sincerely apologize for the confusion
3573 caused by our oversight regarding this critical distinction between the machine-learning
3574 methodology perspective and the chemistry/drug-discovery experimental validation
3575 perspective.

3576 To prevent any further misunderstanding, we have thoroughly reviewed and revised
3577 our manuscript and ensured that no wording or phrasing implies prospective experi-
3578 mental validation. As requested by the editor, we have renamed the relevant part in
3579 our response to clearly reflect our intended meaning and avoid ambiguity. Further-
3580 more, we have explicitly stressed in our revised discussion section that our study is
3581 strictly retrospective and does not involve prospective experimental validations.

3582 Nevertheless, we respectfully emphasize that, while our study is retrospective, we
3583 have explicitly validated OmniMol using multiple independent, experimentally con-
3584 firmed SAR studies from previously published literature. Importantly, most of the
3585 molecules used in these validations were never presented or included during model
3586 development or training. This rigorous retrospective validation clearly demonstrates
3587 OmniMol's practical capability of independently identifying experimentally validated,
3588 optimization-relevant structural features and subtle atomic-level structural modifi-
3589 cations across diverse endpoints. Although these validations remain retrospective in
3590 nature, we believe they provide strong evidence of OmniMol's practical utility and
3591 reliability when applied to unseen chemical structures.

3592 We greatly appreciate the reviewer's insightful and constructive comments, which
3593 have significantly improved the clarity and precision of our manuscript.
3594

3595 **2.2 Comment 2:**

3596

3597 Statistical rigor: Burying one statistical analysis for one dataset deep in supplemental,
3598 which shows a p-value "approaches but does not reach the conventional significance
3599 threshold of 0.05," along with lots of metrics and discussion is classic p-hacking behav-
3600 ior. The comparison and the language around the method comparison remains quite
3601 suspect and revisions decidedly do not address the point raise.

3602 **Our response:**

3603 We thank Reviewer #2 for carefully highlighting this important concern. We fully
3604 understand the reviewer's point regarding statistical rigor and the possibility that

our previous wording and positioning of the statistical analysis might inadvertently suggest selective reporting (“p-hacking”).

We explicitly clarify that we did not engage in selective reporting, multiple testing without acknowledgment, or intentional manipulation of statistical analyses. The McNemar test was explicitly requested by another reviewer during the first-round revision (to include significance tests and standard deviations). We conducted the requested analysis openly and transparently, clearly reporting the results exactly as obtained, including the borderline McNemar p-value (0.086 ± 0.053). We placed this analysis in the supplementary section primarily because the main manuscript was already extensive, and we believed supplementary material would be the appropriate place for detailed statistical tests.

However, we fully acknowledge the reviewer’s and editor’s concerns that our previous wording (“approaches significance,” “substantial advancement,” etc.) was overly strong, given that the McNemar test did not reach conventional statistical significance ($p < 0.05$).

Following the reviewer’s and editor’s explicit guidance, we have carefully revised our supplementary text and main manuscript wording clearly and explicitly:

- We clearly state that at a statistical level, OmniMol’s results over chiral cliff are on par with the current state-of-the-art method, without statistically significantly outperforming it.
- We clearly and neutrally stated the numerical performance differences (e.g., accuracy, MCC, F1) between OmniMol and the baseline method, avoiding phrases like “significantly outperform”.
- We explicitly avoid any implication of statistically significant superiority, clearly acknowledging the limitations of our statistical results throughout the whole manuscript.

2.3 Comment 3:

The response attempts to justify why the complex geometry step is a reasonable thing to do. This complex thing you have done might be a good idea compared to the simpler one. But nothing has changed in that you have not provided evidence that this is true.

Our response:

We sincerely appreciate the reviewer’s valuable comment highlighting the necessity for explicit empirical justification of our proposed geometry-aware modeling strategy. To address this clearly, we conducted an additional targeted ablation study, specifically designed to evaluate the effectiveness of our geometry-related modules.

Experimental Setting and Background. To ensure computational feasibility, we selected 8 representative ADMET-P subtasks (Ames, CYP1A2-inh, CYP3A4-inh, Carcinogenicity, DILI, F(20%), F(30%), and H-HT) from our full dataset. These tasks were specifically chosen due to their importance in our subsequent analysis presented in Section 2.4 (see Figure 3c), where we investigate task relationships and adaptive modeling strategies.

The computational complexity of this additional ablation was significantly reduced compared to our original full-scale model training. Specifically, the following hyperparameters were adjusted:

- Number of Transformer blocks: 2 (reduced from 4)
- Embedding dimension: 384 (reduced from 768)
- Feed-forward embedding dimension: 384 (reduced from 768)
- Total updates: 30,000 (reduced from 500,000)
- Warm-up updates: 1,000 (reduced from 5,000)

3658 **Detailed Explanation of Geometry-related Modules.** Our geometry-aware
 3659 modeling strategy is motivated by the philosophy that accurate molecular prop-
 3660 erty prediction benefits when the model implicitly learns the underlying physics of
 3661 molecular conformational relaxation, i.e., the transition from perturbed (high-energy)
 3662 states to relaxed (equilibrium) states. To achieve this, we introduced three critical
 3663 components explicitly evaluated in this study:

- 3664 • **Geometry Loss:** Refers to supervising the model’s predictions of geometries (atom
 3665 positions and interatomic distances) during training. Specifically, we encourage the
 3666 model to gradually converge from perturbed to equilibrium geometries by applying
 3667 explicit loss functions on node-wise (atomic position) and edge-wise (interatomic
 3668 distance) predictions.
- 3669 • **Geometry Noise Scale:** Denotes the standard deviation of the Gaussian noise
 3670 added to the equilibrium geometry at the training stage. A noise scale of
 3671 0 \AA means no perturbation (original equilibrium geometry), 0.2 \AA represents mod-
 3672 erate perturbation reflecting physically plausible conformational fluctuations, and
 3673 20 \AA represents extremely large random perturbation, significantly distorting atomic
 3674 positions.
- 3675 • **Geometry Update:** Indicates whether our model explicitly updates atomic posi-
 3676 tions iteratively during forward propagation. \checkmark means the model incrementally
 3677 predicts intermediate geometries, gradually approximating the relaxed equilibrium
 3678 state. \times means the model directly predicts final equilibrium geometry without
 3679 iterative refinement.

3681 **Results and Detailed Discussion.** The results of this additional ablation study
 3682 are summarized in Supplementary Table 18.

3683
 3684 **Supplementary Table 18** Additional ablation study evaluating geometry-related modules on
 3685 selected subtasks.

Experiment	Geometry Loss	Geometry Noise Scale (\AA)	Geometry Update	Accuracy	ROC-AUC	Specificity	Sensitivity	MCC
A (Full Geometry Module)	\checkmark	0.2	\checkmark	0.7644	0.8413	0.7909	0.7537	0.5320
B	\times	0.2	\checkmark	0.7212	0.8099	0.7447	0.7212	0.4538
C	\times	0	\checkmark	0.7433	0.8292	0.7704	0.7322	0.4897
D	\times	0	\times	0.7354	0.8040	0.7451	0.7392	0.4713
E	\checkmark	0.2	\times	0.7188	0.7858	0.7177	0.7143	0.4238
F	\times	20	\checkmark	0.7002	0.7870	0.7318	0.6930	0.4159
G (Minimal Geometry Module)	\times	20	\times	0.7109	0.7811	0.7249	0.7095	0.4241

3693
 3694 Key observations are as follows:
 3695

3696 **1. Importance of Iterative Geometry Updates (Compare A vs. E; C vs. D):**

- 3697 • Comparing experiments A (full module) and E (no geometry update), we observe
 3698 a substantial performance drop (ROC-AUC from 0.8413 to 0.7858) when iterative
 3699 geometry updates are removed. This clearly highlights the crucial role of iterative
 3700 prediction steps in accurately capturing the molecular relaxation process.
- 3701 • Intuitively, gradually predicting intermediate states (iterative updates) is essen-
 3702 tial, just like numerically solving differential equations requires incremental steps
 3703 rather than a single jump.

3705 **2. Value of Geometry Supervision (Compare A vs. B; A vs. C):**

- 3706 • Removing geometry supervision (A vs. B) and removing geometry perturbation
 3707 (A vs. C) both degrade performance significantly (ROC-AUC drops from 0.8413
 3708 in A to 0.8099 in B and 0.8292 in C).

3710

• Geometry supervision helps the model explicitly learn representation towards physically realistic conformations. Without this supervision, the model must implicitly infer the relaxation process from noisy and distorted inputs, resulting in reduced accuracy.	3711 3712 3713 3714
3. Impact of Noise Magnitude (Compare G, F vs. A, B):	3715 3716
• Extremely large geometry perturbations (20 Å, experiments F and G) significantly distort structural information, leading to poor predictive performance regardless of geometry updates or supervision.	3717 3718 3719
• This result aligns with expectation—adding excessively large noise essentially destroys meaningful geometric information, making subsequent geometry updates ineffective. Indeed, comparing experiments G (large perturbation without updates) and F (large perturbation with updates), we see minimal performance differences, confirming that iterative updates alone cannot rescue severely corrupted input.	3720 3721 3722 3723 3724 3725
4. Intrinsic Predictive Value of Original Geometry (Compare D vs. E; B vs. C):	3726 3727 3728
• Comparing experiments D (original geometry, no updates, no supervision) and E (perturbed geometry, no updates, supervised), we observe that static original geometry delivers relatively strong baseline performance.	3729 3730 3731
• This observation clearly indicates the intrinsic predictive value contained in the accurate equilibrium geometry itself. Conversely, adding noise without supervision (experiment B vs. C) notably reduces performance, as the model struggles to recover meaningful structural cues from corrupted geometry. Thus, precise molecular geometry inherently encodes valuable structural and relational information crucial for accurate property prediction.	3732 3733 3734 3735 3736 3737
5. Necessity of Combining Iterative Updates with Geometry Supervision (Compare G vs. E vs. A):	3738 3739 3740
• Starting from the minimal setting (experiment G), adding only geometry supervision and moderate perturbation (experiment E) results in marginal improvement. However, adding iterative geometry updates on top of these components (experiment A) provides significant performance gains.	3741 3742 3743 3744
• Intuitively, this demonstrates that the relaxation process demands gradual, incremental geometric updates rather than single-step predictions. Just as incremental numerical methods (e.g., FEM simulations) require small iterative steps to accurately solve complex systems, our model benefits greatly from iterative geometry updates.	3745 3746 3747 3748 3749
This additional ablation complements our previous study (Supplementary Table 4) by explicitly isolating and rigorously evaluating geometry-related modules. While the original ablation investigated broader architectural components (Meta-Encoder, SE(3)-Encoder, Mixed Training, Fine-tuning), this additional study clearly demonstrates the standalone efficacy of geometry-aware modules. Collectively, these results provide robust, complementary empirical evidence supporting our proposed iterative geometry update, supervised geometry loss, and moderate perturbation strategy, thereby justifying our model’s complexity with significant performance gains in molecular property prediction.	3750 3751 3752 3753 3754 3755 3756 3757 3758
We thank the reviewer for prompting this valuable additional analysis, which enhances the clarity and rigor of our methodology.	3759 3760 3761 3762 3763

Appendix B	Response to comments	3605
		3606
	We sincerely thank all reviewers for their thorough and constructive feedback, which has helped us significantly improve our manuscript. Below, we provide detailed response to reviewer 3’s comments. We have also improved our supplementary materials to include more comprehensive analyses and additional results.	3607 3608 3609 3610
		3611
1	Response to Reviewer #3:	3612
		3613
1.1	Comment:	3614
	The concept of the applicability domain (AD) is crucial in the field of quantitative structure-activity relationships (QSAR) and predictive modeling, particularly in the context of chemical and biological data. Netzeva et al. (2005) contributed significantly to this topic by defining applicability domain as the chemical space within which a predictive model can make reliable predictions. It essentially delineates the boundaries of the dataset that the model was trained on and indicates where the model’s predictions can be considered valid. The applicability domain ensures that when a new compound is assessed using the model, it falls within the characteristics of the training set used to develop that model. If a compound lies outside this domain, the prediction could be less reliable or even invalid. This aspect is especially important in regulatory contexts, where predictions made by models are often used to make decisions about safety and efficacy. Netzeva et al. proposed a framework for assessing the applicability domain based on several key approaches:	3615 3616 3617 3618 3619 3620 3621 3622 3623 3624 3625 3626 3627
	Structural Similarity: This approach involves comparing the chemical structure of the new compound to those in the training set. Metrics such as Tanimoto similarity coefficients can be used to quantify how similar the new compound is to the compounds in the dataset.	3628 3629 3630 3631
	Descriptor Space: This involves evaluating the space defined by the molecular descriptors (quantitative representations of chemical structure). The model’s predictive capability is typically confined to a certain range of descriptor values. If a new compound lies outside this range, it is deemed outside the applicability domain.	3632 3633 3634 3635
	Statistical Measures: Some methods employ statistical techniques to define the boundaries of the applicability domain. For instance, the use of confidence intervals around predictions or build in confidence measures can help identify when a prediction is reliable. There are also frameworks like conformal prediction framework.	3636 3637 3638 3639
	An applicability domain measure is considered efficient when it effectively identifies the boundaries within which a predictive model can reliably make predictions.	3640 3641
	You decided for the descriptor space approach. I am wondering because you model is not trained on these descriptors you only used them for the definition of the AD if I understood this correctly. If these are not the features your model is trained on how do you come to the conclusion that their distribution is a good approximation for the limits of your method? Or asking the other way around can you eliminate unreliable predictions -> improve accuracy of the model when excluding compounds that are out of domain. Does your method makes more misclassifications due to outliers or close to the decision boundary (similar compounds but conflicting labels).	3642 3643 3644 3645 3646 3647 3648 3649
	Our Response to Reviewer #3:	3650
	We sincerely thank Reviewer #3 for providing detailed and insightful feedback regarding the definition and validation of the applicability domain (AD). We fully agree with the reviewer on the importance of rigorously defining and validating an effective AD to ensure trustworthy predictions in QSAR modeling, as clearly emphasized by foundational works such as Netzeva et al. (2005).	3651 3652 3653 3654 3655
	In our initial submission and previous response, we employed a simple descriptor-range-based approach to define the AD. Specifically, we selected seven well-known	3656 3657

3658 molecular descriptors—Molecular Weight (MolWt), LogP, Hydrogen Bond Donors
3659 (HBD), Hydrogen Bond Acceptors (HBA), Topological Polar Surface Area (TPSA),
3660 Rotatable Bonds (RB), and Ring Count (RC)—due to their established importance
3661 in QSAR modeling and drug discovery (e.g., Lipinski’s Rule of Five). We initially
3662 defined the "In domain" (ID), "Warning", and "Out-of-domain" (OOD) regions based
3663 on percentile thresholds (e.g., 1%-99% for MolWt and LogP, and 0%-99% for other
3664 descriptors).

3665 However, upon experimental evaluation of this percentile-based method, we
3666 observed significant limitations. Specifically, due to overly restrictive thresholds, the
3667 vast majority of test set compounds fell within the broadly defined "In domain" region.
3668 Consequently, very few or even zero test samples were identified as "Warning" or
3669 "Out-of-domain" across many endpoints. Such scarcity of samples in the latter cate-
3670 gories rendered performance metrics statistically unreliable and prevented meaningful
3671 validation of our AD definition.

3672 In response to this critical limitation, and guided by Reviewer #3’s insightful
3673 suggestions, we have substantially revised and improved our approach:

3674 **Expanded and Comprehensive Descriptor Set:** We have extended our
3675 descriptor set from 7 to 20 molecular descriptors, encompassing a wide range of
3676 physicochemical, topological, electronic, stereochemical, complexity, aromaticity, and
3677 atomic composition characteristics. Although our predictive models (OmniMol) are
3678 graph-based neural networks trained directly on molecular graphs rather than explicit
3679 descriptor vectors, these comprehensive descriptors collectively represent the molecu-
3680 lar chemical space in a detailed and interpretable manner. Importantly, the learned
3681 embeddings from graph-based message passing in our models implicitly capture molec-
3682 ular features closely related to these explicit descriptors. Therefore, employing these
3683 descriptors provides a rational approximation of the chemical space boundaries within
3684 which our model embeddings are valid and reliable.

3685 **Isolation Forest-based Applicability Domain (AD) Definition:** To objec-
3686 tively and rigorously define the AD, we adopted the Isolation Forest anomaly detection
3687 algorithm, a well-established statistical method for identifying structural outliers and
3688 anomalies in high-dimensional descriptor spaces [1–3]. Specifically:

- 3689 • An Isolation Forest model was trained for each endpoint using the 20-dimensional
3690 descriptor vectors from the training compounds.
3691 • Each compound (training and test) obtained an anomaly score from the trained
3692 model, quantifying its structural similarity to the training dataset core.
3693 • A threshold was empirically determined such that approximately 75% of the training
3694 set was considered “In-Domain” (ID), and the remaining 25% as potential “Out-of-
3695 Domain” (OOD) samples, providing statistically meaningful subsets for subsequent
3696 validation.
3697

3698 **Empirical Validation of AD Utility:** To directly address your second question–
3699 whether excluding OOD compounds truly improves the model’s predictive reliability–
3700 we thoroughly evaluated predictive performance differences between ID and OOD
3701 subsets across all endpoints:

3702 **Classification Tasks (39 endpoints):**

- 3703 • Average proportion of ID test samples: 73.18%.
3704 • ID samples consistently showed superior predictive metrics compared to OOD
3705 samples:
3706 – Accuracy improvement (ID vs. OOD): +0.0324 (macro-average)
3707 – F1-score improvement (ID vs. OOD): +0.0297 (macro-average)

3709
3710

• Out of 39 endpoints, 27 endpoints exhibited higher performance for ID samples compared to OOD samples in accuracy and F1-score.	3711
	3712
For example, the SR-ARE endpoint showed an accuracy improvement of 0.070 (ID: 0.912 vs. OOD: 0.842) and corresponding improvement in F1-score (0.066).	3713
	3714
Regression Tasks (12 endpoints):	3715
	3716
• Average proportion of ID test samples: 73.45%.	3717
• ID samples consistently showed significantly better predictive metrics compared to OOD samples:	3718
– MAE reduction (ID vs. OOD): -0.1489 (macro-average improvement)	3719
– R ² increase (ID vs. OOD): +0.1064 (macro-average improvement)	3720
	3721
• Out of 12 endpoints, 10 showed lower MAE and 8 showed higher R ² for ID samples than OOD samples.	3722
	3723
Detailed result is shown in Supplementary Fig. 6 and Supplementary Fig. 7 .	3724
	3725
	3726
	3727
	3728
	3729
	3730
	3731
	3732
	3733
	3734
	3735
	3736
	3737
	3738
	3739
	3740
	3741
	3742
	3743
	3744
	3745
	3746
	3747
	3748
	3749
	3750
	3751
	3752
	3753
	3754
	3755
	3756
	3757
	3758
	3759
	3760
	3761
	3762
	3763

3764
 3765
 3766
 3767
 3768
 3769
 3770
 3771
 3772
 3773
 3774
 3775
 3776
 3777
 3778
 3779
 3780
 3781
 3782
 3783
 3784
 3785
 3786
 3787
 3788
 3789
 3790
 3791
 3792
 3793
 3794
 3795
 3796
 3797
 3798
 3799
 3800
 3801
 3802
 3803
 3804
 3805
 3806
 3807
 3808
 3809
 3810
 3811
 3812
 3813
 3814
 3815
 3816

Supplementary Fig. 6 Classification task performance difference heatmap of AD

Supplementary Fig. 7 Regression task performance difference heatmap of AD.

These comprehensive validation results quantitatively confirm that our revised Isolation Forest-based AD effectively identifies structurally atypical compounds associated with higher prediction errors. Thus, the AD definition we implemented robustly addresses your concerns by empirically demonstrating improved accuracy and reliability when excluding OOD compounds.

Finally, to enhance transparency and facilitate practical use, we have provided detailed instructions and tools for AD assessment in our revised supplementary materials and in our public GitHub repository (https://github.com/bowenwang77/OmniMol/blob/master/check_applicability_domain.py). Users can easily input molecules (as SMILES strings) and select the target endpoint, after which the provided script will automatically compute descriptors, apply the trained Isolation Forest model, and classify each compound as either ID or OOD. Moreover, users can flexibly adjust thresholds according to their specific tolerance for prediction uncertainty.

We hope this substantially revised AD analysis and validation approach comprehensively addresses your valuable suggestions and significantly strengthens the reliability and practical applicability of our predictive models.

References

- [1] Fei Tony Liu, Kai Ming Ting, and Zhi-Hua Zhou. Isolation forest. In *2008 eighth IEEE international conference on data mining*, pages 413–422. IEEE, 2008.
- [2] Robert P Sheridan. Three useful dimensions for domain applicability in qsar models using random forest. *Journal of chemical information and modeling*, 52(3):814–823, 2012.
- [3] Cindy Trinh, Silvia Lasala, Olivier Herbinet, and Dimitrios Meimaroglou. On the development of descriptor-based machine learning models for thermodynamic properties: Part 2—applicability domain and outliers. *Algorithms*, 16(12):573,

3870 2023.
3871
3872
3873
3874
3875
3876
3877
3878
3879
3880
3881
3882
3883
3884
3885
3886
3887
3888
3889
3890
3891
3892
3893
3894
3895
3896
3897
3898
3899
3900
3901
3902
3903
3904
3905
3906
3907
3908
3909
3910
3911
3912
3913
3914
3915
3916
3917
3918
3919
3920
3921
3922